# Optimal paths and dynamical symmetry breaking in the current fluctuations of driven diffusive media

Pablo I. Hurtado[1,2,$]

**1** Departamento de Electromagnetismo y Física de la Materia, Universidad de Granada, Granada 18071, Spain
**2** Institute Carlos I for Theoretical and Computational Physics, Universidad de Granada, Granada 18071, Spain

[$]phurtado@onsager.ugr.es

February 27, 2025

Large deviation theory provides a framework to understand macroscopic fluctuations and collective phenomena in many-body nonequilibrium systems in terms of microscopic dynamics. In these lecture notes we discuss the large deviation statistics of the current, a central observable out of equilibrium, using mostly macroscopic fluctuation theory (MFT) but also microscopic spectral methods. Special emphasis is put on describing the optimal path leading to a rare fluctuation, as well as on different dynamical symmetry breaking phenomena that appear at the fluctuating level. We start with a brief overview of the statistics of trajectories in driven diffusive systems as described by MFT. We then discuss the additivity principle, a simplifying conjecture to compute the current distribution in many one-dimensional ($1d$) nonequilibrium systems, and extend this idea to generic $d$-dimensional driven diffusive media. Crucially, we derive a fundamental relation which strongly constrains the architecture of the optimal vector current field in $d$ dimensions, making manifest the spatiotemporal nonlocality of current fluctuations. Next we discuss the intriguing phenomenon of dynamical phase transitions (DPTs) in current fluctuations, i.e. possibility of dynamical symmetry breaking events in the trajectory statistics associated to atypical values of the current. We first analyze a discrete particle-hole symmetry-breaking DPT in the transport fluctuations of open channels, working out a Landau-like theory for this DPT as well as the joint statistics of the current and an appropriate order parameter for the transition. Interestingly, Maxwell-like violations of additivity are observed in the non-convex regimes of the joint large deviation function. We then move on to discuss time-translation symmetry breaking DPTs in periodic systems, in which the system of interest self-organizes into a coherent traveling wave that facilitates the current deviation by gathering particles/energy in a localized condensate. We also shed light on the microscopic spectral mechanism leading to these and other symmetry breaking DPTs, which is linked to an emerging degeneracy of the ground state of the associated microscopic generator, with all symmetry-breaking features encoded in the subleading eigenvectors of this degenerate subspace. The introduction of an order parameter space of lower dimensionality allows to confirm quantitatively these spectral fingerprints of DPTs. Using this spectral view on DPTs, we uncover the signatures of the recently discovered time-crystal phase of matter in the traveling-wave DPT found in many periodic diffusive systems. Using Doob's transform to understand the underlying physics, we propose a packing-field mechanism to build programmable time-crystal phases in driven diffusive systems. We end up these lecture notes discussing some open challenges and future applications in this exciting research field.

*Before I came here I was confused about this subject. Having listened to your lecture I am still confused. But on a higher level.* **Enrico Fermi**

*The most exciting phrase to hear in science, the one that heralds the most discoveries, is not* "Eureka!" *but* "That's funny..." **Isaac Asimov**

## Contents

# 1  Introduction

Nonequilibrium phenomena appear ubiquitously in nature. Despite their widespread presence and significance, a comprehensive bottom-up approach that predicts nonequilibrium macroscopic behavior from microscopic physics, similar to what equilibrium statistical mechanics provides, is still missing. This gap exists because of the inherent complexity in merging statistical methods with dynamical processes, which are always crucial in nonequilibrium scenarios [1–6]. The absence of such a framework significantly limits our ability to manipulate, control, and engineer many natural and artificial systems which typically operate under nonequilibrium conditions. It is nowadays recognized that macroscopic fluctuations, along with their statistics and associated structures, are key to understanding nonequilibrium physics [1,2,6,7]. The theory of large deviations [8] has emerged as the natural framework of this new understanding, with large deviation functions (LDFs) measuring the probability of fluctuations and the optimal paths sustaining these rare events as central objects in the theory. Out of equilibrium, LDFs play a role similar to that of thermodynamic potentials in equilibrium systems [1,2], though these LDFs inherit the intricacies of nonequilibrium behavior in the form of long-range correlations, non-local behaviors, etc. Thus, the long-sough general theory of nonequilibrium phenomena is currently envisaged as a theory of macroscopic fluctuations, making the study and understanding of LDFs and the associated optimal paths a central focus in theoretical physics. This paradigm has led to a number of groundbreaking results valid arbitrarily far from equilibrium, many of them in the form of fluctuation theorems [9–14] or nonequilibrium equalities [15–19], but also Clausius-like inequalities [20, 21], thermodynamic uncertaintly relations [22–25], etc.

A crucial aspect of this emerging paradigm is identifying the macroscopic observables that characterize nonequilibrium behavior. The system of interest often conserves locally certain quantities (such as particle density, energy, momentum, charge, etc.), and the key nonequilibrium observable is hence the current or flux the system develops in response to an external driving as e.g. boundary-induced gradients or external fields. In this way, the understanding of current statistics in terms of microscopic or mesoscopic dynamics has become one of the central goals of nonequilibrium statistical physics, sparking an extensive research effort that has led to some remarkable results [1, 2, 26–58].

The typical problem we will be interested in is that of a channel of length $L$ connecting two particle reservoirs at different densities (or chemical potentials), see Fig 1 for a sketch of this setting. Particles might be also driven in some preferential direction by an external field $E$. In this situation one expects the appearance of a particle current flowing typically from the high density reservoir to the low density one [1, 2, 5]. If we monitor the total current $Q_\tau$ flowing during a time $\tau$, we expect a behavior as the one depicted in the bottom left panel of Fig. 1, with the cumulative current increasing approximately linearly in time with unavoidable random fluctuations due to the stochastic/chaotic nature of the particle dynamics. However the net current (or total current per unit time), $q = Q_\tau/\tau$, will converge as time increases to a well defined value, $\lim_{\tau\to\infty} q = \langle q \rangle$, given by the well known Fick's law of diffusion, i.e proportional to the density gradient in the channel and the external driving field, if any [1]. If we perform many realizations of this experiment for a long but finite $\tau$, each of them will give a different result for the cumulative current $Q_\tau$ due to the aforementioned fluctuations, see bottom-left panel in Fig. 1, thus leading to a distribution of the time-averaged current $q$ captured by a probability density function (pdf) $P_\tau(q)$, see bottom-right panel in Fig. 1. In most cases of interest, this probability can be shown to obey a *large-deviation principle* for long times $\tau$ and large system sizes $L$ [1, 2, 44], i.e. $P_\tau(q)$ scales in this limit as

$$P_\tau(q) \asymp e^{+\tau L G(q)}, \tag{1}$$

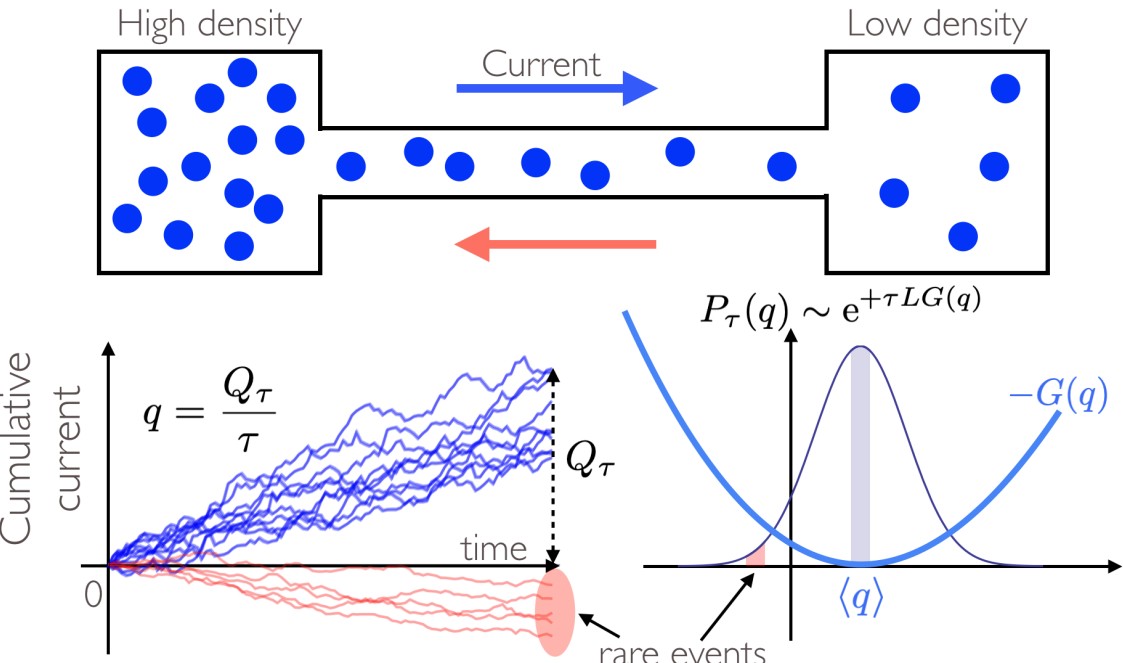

Figure 1: **An interesting problem.** Top panel: A channel of length $L$ connects two reservoirs at different densities. Particles might be also driven in some preferential direction by an external field $E$. Due to the density gradient, a particle current ensues. Bottom left panel: Different realizations of the experiment for long but finite time $\tau$ lead to different values of the cumulative current $Q_\tau$ and hence to a distribution of the time-averaged current $q = Q_\tau/\tau$, as captured by the probability density function (pdf) $P_\tau(q)$. Bottom right panel: $P_\tau(q)$ obeys a large-deviation principle, scaling exponentially with time and the system size, and the current LDF $G(q)$ captures the probability of both typical and rare current fluctuations.

where the symbol "$\asymp$" represents asymptotic logarithmic equality, i.e.

$$\lim_{\tau \to \infty} \frac{1}{\tau} \ln P_\tau(q) = G(q). \tag{2}$$

The function $G(q) \le 0$ defines the current large deviation function (LDF), and its typical shape is shown in the bottom-right panel in Fig. 1. This LDF is a measure of the (exponential) rate at which the probability of observing an empirical current $q$ –appreciably different from its steady-state value $\langle q \rangle$– decays as both $\tau$ and $L$ increase, and captures the concentration of the probability measure $P_\tau(q)$ around $\langle q \rangle$ as both $\tau, L \to \infty$. Note that this implies that $G(\langle q \rangle) = 0$. Most realizations of our long (but finite) time experiment will yield values of $q$ around the average value $\langle q \rangle$, i.e. close to the peak of $P_\tau(q)$. However, if we perform many measurements, few times we may find *surprising* results, as e.g. realizations where after a long time $\tau$ the cumulative current is negative, i.e. where a net flow of particles from the low density reservoir to the high density one has been maintained during a long time. These *rare events* populate the tails of the distribution $P_\tau(q)$, and despite the popular belief, they do not violate any fundamental physical principle (as e.g. the second law of thermodynamics). They are simply very, very unlikely, the more the longer the observation time $\tau$ is (and/or the larger the system size $L$ is).

These rare events codify interesting physics, important to understand nonequilibrium behavior. A particularly interesting observable is the typical or dominant path in phase space

responsible of a given current fluctuation. For instance, these optimal paths reflect and inherit the symmetries of microscopic physics at the mesoscopic level, the main example being microscopic time-reversibility, which implies the invariance of optimal paths against current inversions $q \leftrightarrow -q$. These (and other) symmetry properties of optimal paths can be then used to derive general fluctuation theorems [9–13, 46, 59–65].

Other times, when large enough fluctuations come about, the associated optimal path or trajectory may change drastically, in a singular manner, reflecting an underlying dynamic phase transition (DPT) at the fluctuating level [14,29,43,44,52,53,66–71]. Interestingly, some dynamical phases may display emergent order and collective rearrangements in their trajectories, including symmetry-breaking phenomena [14, 29, 43, 44], while the LDFs controlling the statistics of these fluctuations exhibit nonanalyticities and Lee-Yang singularities [72–79] at the DPT reminiscent of standard critical behavior. In addition to their conceptual importance, DPTs play also a key role in understanding the physics of different systems, from glass formers [69, 80–86] to superconducting transistors and micromasers [87, 88]. There have been also recent applications of DPTs to design quantum thermal switches [89–91], i.e. quantum devices where the heat current flowing between hot and cold reservoirs can be completely blocked, modulated or turned on at will. Remarkably, rare events can be made typical using Doob's h-transform [92–95] or external fields with optimal dissipation [2]. This can be then used to exploit existing DPTs to engineer and control complex systems with a desired statistics *on demand* [96], for instance to build time crystal phases in nonequilibrium matter [97–103].

These research efforts have been aided by the emergence of powerful tools capable of delving into the steady-state and fluctuating behavior of many-body nonequilibrium systems. These tools, which include advanced rare-event simulation techniques [31, 104, 105], spectral methods for microscopic dynamical analysis [8, 80, 81, 90, 93, 101, 102, 106, 107], and a compelling macroscopic fluctuation theory [1,2,26,28,32], among others, are opening new avenues of research into nonequilibrium physics. The combined use of these tools allow, among other things, to estimate with precision the probability of both typical and rare current fluctuations, offering also a complete picture of the optimal (or most likely) path that the system of interest follows in order to maintain one of these current fluctuations, including the possible emergence of DPTs, and the path symmetries that follow from microscopic dynamics. These lecture notes focus on understanding the properties and spatiotemporal structure of these optimal paths.

In order to achieve this general objective, we will first provide a brief overview of the statistical physics of trajectories and macroscopic fluctuation theory [2]. We will next describe a simplifying conjecture to compute the current LDF and the associated optimal paths in one-dimensional ($1d$) nonequilibrium systems, known as the additivity principle [27, 36]. We will then explain in detail how to generalize this conjecture to higher-dimensional systems [50, 53]. In particular, we will show how the system of interest uses the extra degrees of freedom at hand to optimize the probability of a rare event, in particular by developing a nontrivial structure in its hydrodynamic fields. We will then move on to study spontaneous symmetry-breaking at the fluctuating level [108]. We will do so first in open nonequilibrium systems connected to boundary reservoirs, possibly at different densities. We will show how, under certain symmetry conditions, a dynamical phase transition appears in the current statistics of some open systems [52,71,102,109], accompanied by a discrete $\mathbb{Z}_2$-symmetry breaking phenomenon. Next we will discuss DPTs in periodic structures [14, 43, 102, 110], where traveling density waves appear as optimal paths for large enough current fluctuations, breaking continuous time-translation symmetry. The previous results will be discussed at the hydrodynamic level, using macroscopic fluctuation theory and local stability analysis as main tools. We will provide a microscopic, spectral point of view on these symmetry-breaking DPTs in the next section, where we will describe the general spectral mechanism giving rise to continuous

DPTs in many body Markov systems, showing in detail how the different dynamical phases emerge from the specific structure of the leading eigenvectors of a deformed generator. This microscopic, spectral description provides a picture complementary to the field-theoretic MFT scenario. Finally, and as a proof of concept, we will combine these microscopic, spectral methods with hydrodynamic and MFT tools to show how to build continuous time crystal phases in many-body systems from existing DPTs in their current statistics.

The material covered in these lecture notes is mostly published in different papers elsewhere, both by the author and collaborators as well as main scientist working in the field. In this sense, the main references we will be inspired by are [1, 2, 14, 27, 36, 38, 43, 44, 50, 52, 53, 61, 71, 101–103, 109, 110] However, these lecture notes will make a special effort to convey all main ideas and methods in a self-contained, pedagogic manner, skipping as fewer details as possible without boring the reader, and trying to connect in a comprehensive way the different results to obtain a broader picture of this fascinating corner of knowledge.

## 2  Statistics of trajectories and macroscopic fluctuation theory

In this section we will review the statistical physics of an ensemble of trajectories conditioned to a given total current $Q_\tau$ flowing through the system of interest over a long time $\tau$, as described in the typical example of previous section. In order not to clutter our notation and explain the approach in full detail, we particularize our discussion to one-dimensional (1$d$) systems, though extensions to arbitrary dimension and vector currents are straightforward and will be discussed later on.

At the mesoscopic level, most of the systems whose dynamical fluctuations we are interested in can be described by a locally-conserved density field $\rho(x, t)$, representing a density of particles, energy, etc., which evolves in time according to a fluctuating hydrodynamic equation. This can be seen as a continuity equation

$$\partial_t \rho + \partial_x j = 0 \,, \tag{3}$$

coupling the time variation of the local density field $\rho(x, t)$ with the gradient of a fluctuating local current $j(x, t)$. Space and time variables has been already scaled diffusively, i.e. $x = \tilde{x}/L \in [0, 1]$ and $t = \tilde{t}/L^2$, where $\tilde{x}$ and $\tilde{t}$ are the microscopic space and time variables, respectively, and $L$ is the system size. This current field typically obeys Fick's (or Fourier's) law [1, 2, 5],

$$j(x, t) = -D(\rho)\partial_x \rho(x, t) + \sigma(\rho)E + \xi(x, t) \,, \tag{4}$$

where $D(\rho)$ and $\sigma(\rho)$ are the diffusivity and mobility transport coefficients, respectively, and $E$ is a possible external field applied on the system of interest. These transport coefficients, which are typically nonlinear functions of the local density field depending on the microscopic details of the model at hand, can be calculated in many cases using simplifying approximations such as the local equilibrium hypothesis [111–113]. They are related via a local Einstein relation $D(\rho) = f_0''(\rho)\sigma(\rho)$, with $f_0(\rho)$ the equilibrium free energy of the system at hand. The stochastic field $\xi(x, t)$ in Eq. (4) is a Gaussian white noise, with zero average $\langle \xi \rangle = 0$ and delta-correlated in space and time,

$$\langle \xi(x, t)\xi(x', t') \rangle = \frac{\sigma(\rho)}{L}\delta(x - x')\delta(t - t') \,. \tag{5}$$

This noise term captures all the fast degrees of freedom which have been integrated out in the coarse-graining procedure going from microscopic physics to the mesoscopic hydrodynamic description of Eqs. (3)-(4). The system size $L$ is typically a large parameter, $L \gg 1$, so the noise

term will be weak, causing the selection of an optimal path among all possible trajectories compatible with a given fluctuation (see below). To end the description, these mesoscopic evolution equations must be supplemented by appropriate boundary conditions, which can be either periodic or open, depending on the particular problem of interest. For the time being, and having in mind the typical problem described in the introduction for the sake of clarity, we will consider open boundary conditions as corresponding to a system coupled to boundary reservoirs, so $\rho(0, t) = \rho_0$ and $\rho(1, t) = \rho_1 \ \forall t \in [0, \tau]$. Later on we will consider other geometries.

At this mesoscopic level, a system trajectory is specified by the values of the density and current fields at all points of space during a given period of time $[0, \tau]$, i.e. $\{\rho(x, t), j(x, t)\}_0^\tau$. The probability $P(\{\rho, j\}_0^\tau)$ of any trajectory can be computed using a path integral formalism [1, 2, 44], and scales in the large-size limit as

$$P(\{\rho, j\}_0^\tau) \asymp e^{+L \mathcal{I}_\tau[\rho, j]}, \tag{6}$$

where $\mathcal{I}_\tau[\rho, j]$ is the well-known macroscopic fluctuation theory (MFT) action [2]

$$\mathcal{I}_\tau[\rho, j] = -\int_0^\tau dt \int_0^{1^*} dx \frac{\left(j + D(\rho)\partial_x \rho - \sigma(\rho)E\right)^2}{2\sigma(\rho)}. \tag{7}$$

The probability measure (6)-(7) represents the ensemble of space-time trajectories at this mesoscopic level of description, and it is nothing but the Gaussian cost of the white noise field responsible of the excess current fluctuation, $\xi(x, t) = j(x, t) + D(\rho)\partial_x \rho(x, t) - \sigma(\rho)E$, see Eq. (4). The asterisk $^*$ in the MFT action (7) means that the fields $\rho(x, t)$ and $j(x, t)$ are of course not independent, as they are coupled via the continuity equation constraint, $\partial_t \rho + \partial_x j = 0$, see Eq. (3), so $\mathcal{I}_\tau[\rho, j] \to -\infty$ for trajectories for which this constraint is not satisfied. We will drop off this $^*$ eventually and assume the continuity constraint implicitly (or impose it explicitly).

Interestingly, for each fixed trajectory $\{\rho(x, t), j(x, t)\}_0^\tau$, we can compute an associated empirical space&time-averaged current

$$q = \frac{1}{\tau} \int_0^\tau dt \int_0^1 dx \, j(x, t). \tag{8}$$

The probability of a given current $q$ can be now obtained by *summing* over all trajectories compatible with the prescribed current and the known constraints. This follows from the path integral

$$P_\tau(q) = \int \mathcal{D}\rho \, \mathcal{D}j \, P(\{\rho, j\}_0^\tau) \, \delta\left(\partial_t \rho + \partial_x j\right) \delta\left[q - \tau^{-1} \int_0^\tau dt \int_0^1 dx \, j(x, t)\right] \tag{9}$$

$$= \int \mathcal{D}\rho \, \mathcal{D}j \, e^{-L \int_0^\tau dt \int_0^1 dx \frac{(j + D(\rho)\partial_x \rho - \sigma(\rho)E)^2}{2\sigma(\rho)}} \, \delta\left(\partial_t \rho + \partial_x j\right) \delta\left[q - \tau^{-1} \int_0^\tau dt \int_0^1 dx \, j(x, t)\right],$$

where the Dirac delta-functionals impose explicitly the different constraints. As stated above, for long $\tau$ and large $L$ this probability obeys a *large-deviation principle* of the form

$$P_\tau(q) \asymp e^{+\tau L G(q)}. \tag{10}$$

The current LDF $G(q)$ can be thus obtained from the MFT action in a saddle-point calculation for large $L$ and $\tau$, i.e. by minimizing the action functional (7) over all trajectories sustaining such current,

$$G(q) = -\lim_{\tau \to \infty} \frac{1}{\tau} \min_{\{\rho, j\}_0^\tau}^* \int_0^\tau dt \int_0^1 dx \frac{\left(j + D(\rho)\partial_x \rho - \sigma(\rho)E\right)^2}{2\sigma(\rho)}, \tag{11}$$

where the $^*$ means again that the minimization procedure must be compatible with the pre-scribed constraints, i.e. the continuity equation $\partial_t \rho + \partial_x j = 0 \; \forall x \in [0,1]$ and $\forall t \in [0,\tau]$, the empirical current $q = \tau^{-1} \int_0^\tau dt \int_0^1 j(x,t)$, boundary conditions, etc. The optimal trajec-tory $\{\rho_q(x,t), j_q(x,t)\}_0^\tau$ solution of this variational problem defines the optimal path that the system of interest follows in mesoscopic phase space to sustain a current fluctuation $q$.

This MFT calculation leads to a hard variational problem whose general solution remains challenging in most cases. Therefore a main research path has been to explore different solu-tion schemes and simplifying hypotheses. This is the case for instance of the additivity prin-ciple, to be described in next subsection, which can be justified under certain conditions and used within MFT to obtain explicit predictions for the current LDF and the associated optimal path. As usual in physics, it is as important to formulate a reasonable conjecture as to know its range of validity. As we will see, the additivity principle may be eventually violated for large enough fluctuations, but quite remarkably this additivity breakdown, which is well character-ized within MFT, proceeds via a dynamical phase transition at the fluctuating level involving a symmetry breaking event. More on this issue below.

## 2.1 An additivity principle for current fluctuations in $1d$

The additivity principle is a conjecture, first proposed by T. Bodineau and B. Derrida [27], that enables the explicit calculation of the current statistics in $1d$ diffusive systems in contact with two boundary reservoirs at arbitrary densities/temperatures. Its name refers to an additivity property of current fluctuations in $1d$ systems when slicing the system into different small subsystems [27, 38, 114]: the probability of a current fluctuation factorizes, once maximized over the contact density between slices, and this implies a variational additivity property for the current LDF [27, 38].

Within the context of MFT [2, 44], the additivity principle can be better understood as a special feature of the optimal path to a current fluctuation, well supported on physical grounds. Indeed, in many cases, the optimal path to a current fluctuation turns out to be (mostly) *time-independent*, except for some short initial and final transients. This time-independence con-jecture can be shown to be mathematically equivalent to the additivity principle of Bodineau and Derrida [27]. In this way, under this assumption the optimal fields may have at most only spatial structure, i.e. $\rho_q(x)$ and $j_q(x)$. Now, because the continuity equation constraint, $\partial_t \rho_q + \partial_x j_q = 0$, it is immediate to show that under this time-independence hypothesis the optimal current field $j_q(x)$ is just a constant across the whole $1d$ system. And due to the con-straint on the space&time-averaged current, $q = \int_0^1 j_q(x)\, dx$, this constant is just the empirical current,

$$j_q(x) = q. \tag{12}$$

Note that this is just a property of $1d$ systems. As soon as we move to $d > 1$, the time-independence of optimal profiles together with the continuity constraint will lead to divergence-free (but not necessarily constant) optimal current fields. We will show how $d$-dimensional many-body systems *use* this extra freedom to maximize the probability of a rare fluctuation.

The physical picture associated to the additivity principle is that of a many-body system that, after a short transient time at the beginning of the large deviation event (negligible for the probability measure of the long-time path), settles into a time-independent state with a structured density field, which can be different from the stationary one, and a spatially uniform current field equal to $q$. This particular solution is expected to minimize the cost of a current fluctuation at least for small and moderate deviations from the average behavior. Under this assumption, the variational problem for the current LDF $G(q)$ in Eq. (11) reduces to

$$G(q) = -\min_{\rho(x)} \int_0^1 dx \; \frac{[q + D(\rho)\partial_x \rho - \sigma(\rho)E]^2}{2\sigma(\rho)}, \tag{13}$$

or equivalently $G(q) = -\min_{\rho(x)} \mathcal{F}_q(\rho)$, where $\mathcal{F}_q(\rho)$ is the integral in the right-hand side (rhs) of the previous equation. To derive a differential equation for the optimal density profile $\rho_q(x)$ that solves the above variational problem, we need to perform the functional derivative of the above expression and solve for $\delta \mathcal{F}_q(\rho)/\delta \rho = 0$. For that we perturb the density field, $\rho \to \rho + \delta \rho$, and write the resulting functional as

$$\mathcal{F}_q(\rho + \delta\rho) \approx \mathcal{F}_q(\rho) + \int_0^1 \left( \frac{\delta \mathcal{F}_q(\rho)}{\delta \rho} \right) \delta\rho(x) dx. \tag{14}$$

We leave this functional derivative as an excercise to the interested (and/or unfamiliarized) reader, just note that (i) it is typically easier to perform the functional derivative after expanding the $[\dots]^2$ in Eq. (13), and (ii) integration by parts is needed for the terms with $\partial_x \delta\rho$ to reach an expression with the structure of Eq. (14). After some simplifications, the differential equation for the optimal density profile $\rho_q(x)$ is

$$\left( \frac{D(\rho)^2}{\sigma(\rho)} \right) \partial_x^2 \rho + \left( \frac{D(\rho)^2}{2\sigma(\rho)} \right)' (\partial_x \rho)^2 - q^2 \left( \frac{1}{2\sigma(\rho)} \right)' - \frac{E^2}{2} \sigma'(\rho) = 0, \tag{15}$$

where $A'(\rho)$ means derivative with respect to the argument, and we are abusing language somewhat when identifying $\partial_x \rho = d\rho(x)/dx = \rho'(x)$. Multiplying all terms in the previous equation by $\partial_x \rho$, we have

$$\left( \frac{D^2}{\sigma} \right) (\partial_x \rho)(\partial_x^2 \rho) + \left( \frac{D^2}{2\sigma} \right)' (\partial_x \rho)^3 - q^2 \left( \frac{1}{2\sigma} \right)' \partial_x \rho - \frac{E^2}{2} \sigma' \partial_x \rho = 0. \tag{16}$$

The first two terms in the lhs just correspond to $\partial_x [\frac{D^2}{2\sigma}(\partial_x \rho)^2]$. Noting also that $(\frac{1}{2\sigma})' \partial_x \rho = \partial_x (\frac{1}{2\sigma})$ and $\sigma' \partial_x \rho = \partial_x \sigma(\rho)$, we arrive at

$$\partial_x \left[ \frac{D^2}{2\sigma} (\partial_x \rho)^2 - \frac{q^2}{2\sigma} - \frac{E^2}{2} \sigma \right] = 0, \tag{17}$$

which can be integrated once to obtain

$$D^2(\rho) \left( \frac{d\rho(x)}{dx} \right)^2 = q^2 + 2\sigma(\rho) K(q^2, E^2) + E^2 \sigma^2(\rho), \tag{18}$$

where $K(q^2, E^2)$ is an integration constant which is fixed by one of the boundary conditions for the density field, $\rho(0) = \rho_0$ or $\rho(1) = \rho_1$, the other one being used as boundary condition for the first-order ODE (18). In what follows we assume $\rho_0 > \rho_1$ without loss of generality. In this way, the solution to Eq. (18) is the optimal (time-independent) density profile $\rho_q(x)$ that the system adopts to sustain a current fluctuation $q$ under the additivity conjecture [27].

A first observation is that this optimal profile is independent of the sign of the current and/or the external field, i.e. $\rho_q(x)$ is invariant under the changes $q \to -q$ and $E \to -E$. This is a rather counter-intuitive result at first, which is ultimately a reflection of the time reversibility of microscopic dynamics at the mesoscopic, fluctuating level. Indeed, for each microscopic forward path compatible with a fluctuation $q$, there exists a backward path, directly related to the forward path by time reversibility, leading to a current $-q$. Note however that, although $\rho_q(x) = \rho_{-q}(x)$, the probability of these two different current fluctuations is typically (exponentially) different, $P_\tau(q) \neq P_\tau(-q)$, see Eqs. (10) and (13), though they are directly related. In fact, we can now use the invariance of the optimal path under changes in the current (and/or external field) signs to formulate a generalized fluctuation theorem [9–13, 59–61]. Extending our previous notation to write the current LDF under an external field $E$ as $G_E(q)$

and the optimal density profile responsible of this fluctuation, solution of Eq. (18), as $\rho_{q,E}(x)$, we can write

$$G_E(q) = -\int_0^1 dx \, \frac{\left[q + D(\rho_{q,E})\partial_x\rho_{q,E} - \sigma(\rho_{q,E})E\right]^2}{2\sigma(\rho_{q,E})}\,, \tag{19}$$

see Eq. (13). Introducing now two indices $\alpha, \beta = \pm 1$, and using the optimal profile invariance, $\rho_{q,E}(x) = \rho_{\beta q, \alpha E}(x)$, it is easy to show from Eq. (19) that

$$G_E(q) - G_{\alpha E}(\beta q) = (1-\alpha)\gamma E - (1-\beta)\epsilon q + (1-\alpha\beta)Eq\,, \tag{20}$$

where we have defined two constants

$$\gamma = \int_{\rho_0}^{\rho_1} D(\rho)d\rho\,, \qquad \epsilon = \int_{\rho_0}^{\rho_1} \frac{D(\rho)}{\sigma(\rho)}d\rho\,. \tag{21}$$

In particular, for $\alpha = +1$ and $\beta = -1$ we recover the standard Gallavotti-Cohen fluctuation theorem $G_E(q) - G_E(-q) = 2(E - \epsilon)q$, relating the probability of a current fluctuation $q$ and the reversed event $-q$ [9–12, 59].

Equation (19), together with (18), completely determines the current distribution, which is in general non-Gaussian except for small current fluctuations around the average $\langle q \rangle$, as dictated by the central limit theorem. We can now analyze the structure of the optimal path by looking at its governing differential equation (18). In the simplest case, when the integration constant $K = K(q^2, E^2)$ is large enough for the rhs of Eq. (18) not to vanish (something that happens for currents close enough to the average), the optimal profile $\rho_q(x)$ will be strictly monotone, with a negative slope $\forall x \in [0, 1]$ (since we are assuming $\rho_0 > \rho_1$), so

$$\frac{d\rho(x)}{dx} = -\frac{1}{D(\rho)}\sqrt{q^2 + 2K\sigma(\rho) + E^2\sigma^2(\rho)}\,. \tag{22}$$

We can integrate this equation to obtain an implicit equation for the constat $K$ in this regime. Indeed, writing (22) as a differential form

$$dx = \frac{-D(\rho)}{\sqrt{q^2 + 2K\sigma(\rho) + E^2\sigma^2(\rho)}}d\rho\,, \tag{23}$$

and integrating this expresion in the whole interval $x \in [0, 1]$ we find

$$1 = \int_{\rho_1}^{\rho_0} \frac{D(\rho)}{\sqrt{q^2 + 2K\sigma(\rho) + E^2\sigma^2(\rho)}}d\rho\,, \tag{24}$$

which just gives $K$ in terms of $q^2$ and $E^2$ in this monotone regime. Using the differential form (23) to change variables from $x$ to $\rho$ in the integral (19) for the current LDF, and using Eq. (22) to write explicitly $\partial_x\rho_{q,E}$ in this regime, we find the following expression for the current LDF in the regime of monotonous optimal profiles

$$G_E(q) = \int_{\rho_1}^{\rho_0} \frac{D(\rho)}{\sigma(\rho)}\left[(q-E) - \frac{q^2 + K\sigma(\rho) + E^2\sigma^2(\rho) - Eq\sigma(\rho)}{\sqrt{q^2 + 2K\sigma(\rho) + E^2\sigma^2(\rho)}}\right]d\rho\,. \tag{25}$$

On the other hand, for $K < 0$ the optimal density profile may exhibit some extrema, i.e. the rhs of Eq. (18) may vanish at some points $x^*$ where $\partial_x\rho|_{x^*} = 0$, i.e.

$$q^2 + 2K\sigma(\rho^*) + E^2\sigma^2(\rho^*) = 0\,, \tag{26}$$

where $\rho^* = \rho(x^*)$. To further advance and analyze the shape of the non-monotonous optimal density profiles, the number of extrema, etc. we would need to specify the particular form

of the diffusivity and mobility transport coefficients which characterize the particular model at hand. This endeavor lies outside the scope of the present lecture notes, as it is model-dependent, but we refer the interested reader to more specialized literature, as e.g. Refs. [1, 2, 27, 36, 38, 40, 44, 115–118] and references therein, for detailed discussions of this issue in particular models. In all cases, the predictions based on the additivity principle have been confirmed (within its range of validity) against detailed simulations of current statistics in these $1d$ microscopic transport models.

As a final remark, note that the current LDF obtained from these calculations in different models of transport typically exhibits linear tails for large enough $|q|$, i.e. $G_E(q) \propto -|q|$ for $|q - \langle q \rangle| \gg 1$. This happens e.g. for the Kipnis-Marchioro-Presutti (KMP) model of heat transport [38, 114, 119], the weakly-asymmetric simple exclusion process (WASEP) [43, 120–122], etc. These linear tails mean that large current fluctuations are far more probable than anticipated from the central limit theorem prediction. Indeed, the system adapts its density profile to maximize the probability of these rare events.

# 3 What about $d > 1$? Vector current statistics and a weak additivity principle

One-dimensional systems constitute the ideal ground where to start exploring complex questions in many-body physics, but eventually the interest shifts back to more realistic high dimensional systems, $d > 1$, where the interplay between many-body cooperative phenomena and the higher dimensionality can lead to intricate phenomenology.

This is the case of the problem of interest in these lecture notes, i.e. the understanding of current statistics in driven diffusive systems. In particular, we want to generalize the Aditivity Principle of the previous section to $d > 1$. For that, we consider a broad class of $d$-dimensional driven diffusive systems characterized by a density field $\rho(\mathbf{r}, t)$, with $\mathbf{r} \in \Lambda \equiv [0, 1]^d$ and $t \in [0, \tau]$, which evolves in time according to the fluctuating hydrodynamics equation [1,2,44]

$$\partial_t \rho(\mathbf{r}, t) + \boldsymbol{\nabla} \cdot \left( -D(\rho) \boldsymbol{\nabla} \rho(\mathbf{r}, t) + \sigma(\rho) \mathbf{E} + \boldsymbol{\xi}(\mathbf{r}, t) \right) = 0. \tag{27}$$

The vector field $\mathbf{j}(\mathbf{r}, t) \equiv -D(\rho) \boldsymbol{\nabla} \rho(\mathbf{r}, t) + \sigma(\rho) \mathbf{E} + \boldsymbol{\xi}(\mathbf{r}, t)$ is a fluctuating current, with $\mathbf{E}$ an external driving field, so Eq. (27) is just a $d$-dimensional continuity equation equivalent to Eq. (3) in $1d$. The deterministic part of this current field $\mathbf{j}(\mathbf{r}, t)$ is given by the $d$-dimensional version of Fick's law under external driving. As before, the stochastic vector field $\boldsymbol{\xi}(\mathbf{r}, t)$ is a Gaussian white noise term with zero average, $\langle \boldsymbol{\xi}(\mathbf{r}, t) \rangle = 0$, and variance

$$\langle \xi_\alpha(\mathbf{r}, t) \xi_\beta(\mathbf{r}', t') \rangle = \frac{\sigma(\rho)}{L^d} \delta_{\alpha\beta} \delta(\mathbf{r} - \mathbf{r}') \delta(t - t'),$$

with $L$ the system size in natural units, and $\alpha, \beta \in [1, d]$. Before continuing, let us mention that the our treatment here can be generalized to *anisotropic* driven diffusive systems by promoting the diffusivity and mobility transport coefficients from scalars to matrices, i.e. $\hat{D}(\rho) \equiv D(\rho) \hat{A}$ and $\hat{\sigma}(\rho) = \sigma(\rho) \hat{A}$, with $\hat{A}$ a diagonal anisotropy matrix with components $\hat{A}_{\alpha\beta} = a_\alpha \delta_{\alpha\beta}$, $\alpha, \beta \in [1, d]$. However, in order to keep our discussion simple, we will stick to isotropic systems for which the diffusivity and mobility are just simple scalar functions of the local density, and refer the interested reader to Refs. [50, 53] for a more technical discussion on the role of anisotropy (see also [56]).

The probability of observing a particular trajectory $\{\rho(\mathbf{r}, t), \mathbf{j}(\mathbf{r}, t)\}_0^\tau$ of duration $\tau$ for the density and current fields can we written starting from the Fokker-Planck description of the Langevin equation (27). This probability scales [1,2,44] in a large deviation form $P(\{\rho, \mathbf{j}\}_0^\tau) \asymp$

$\exp(+L^d I_\tau[\rho,\mathbf{j}])$, with a MFT action equivalent to that of $1d$ systems, see Eq. (7),

$$I_\tau[\rho,\mathbf{j}] = -\int_0^\tau dt \int_\Lambda^* d\mathbf{r} \frac{\left[\mathbf{j} + D(\rho)\boldsymbol{\nabla}\rho - \sigma(\rho)\mathbf{E}\right]^2}{2\sigma(\rho)}. \tag{28}$$

Here $\mathcal{J}(\mathbf{r},t) \equiv \mathbf{j}(\mathbf{r},t) + D(\rho)\boldsymbol{\nabla}\rho(\mathbf{r},t) - \sigma(\rho)\mathbf{E}$ is just the *excess current*, i.e. the departure of the current vector field $\mathbf{j}(\mathbf{r},t)$ from its constitutive form $-D(\rho)\boldsymbol{\nabla}\rho + \sigma(\rho)\mathbf{E}$. As before, the asterisk $^*$ in the MFT action (28) represents the unavoidable constraints, i.e. that the fields $\rho(\mathbf{r},t)$ and $\mathbf{j}(\mathbf{r},t)$ must be coupled via the continuity equation at every point of space and time, $\partial_t\rho + \boldsymbol{\nabla}\cdot\mathbf{j} = 0$, as well as the appropriate boundary conditions. Indeed, for trajectories $\{\rho,\mathbf{j}\}_0^\tau$ not obeying these constraints $I_\tau[\rho,\mathbf{j}] \to -\infty$.

The probability $P_\tau(\mathbf{q})$ of observing a space&time-averaged empirical vector current

$$\mathbf{q} = \frac{1}{\tau}\int_0^\tau dt \int_\Lambda d\mathbf{r}\,\mathbf{j}(\mathbf{r},t), \tag{29}$$

is obtained by summing up the probability of all trajectories $\{\rho,\mathbf{j}\}_0^\tau$ compatible with a fixed $\mathbf{q}$, i.e.

$$P_\tau(\mathbf{q}) = \int \mathcal{D}\rho\,\mathcal{D}\mathbf{j}\,\mathrm{P}\left(\{\rho,\mathbf{j}\}_0^\tau\right)\delta\left(\partial_t\rho + \boldsymbol{\nabla}\cdot\mathbf{j}\right)\delta\left(\mathbf{q} - \tau^{-1}\int_0^\tau dt \int_\Lambda d\mathbf{r}\,\mathbf{j}\right), \tag{30}$$

The Dirac $\delta$-functionals guaranteeing the contraints can be implemented explicitly via their Fourier-Laplace representation [53], namely

$$\delta\left(\mathbf{q} - \tau^{-1}\int_0^\tau dt \int_\Lambda d\mathbf{r}\,\mathbf{j}\right) = \int d\boldsymbol{\lambda}\,e^{-L^d\boldsymbol{\lambda}\cdot[\tau\mathbf{q} - \int_0^\tau dt \int_\Lambda d\mathbf{r}\,\mathbf{j}(\mathbf{r},t)]}, \tag{31}$$

$$\delta\left(\partial_t\rho + \boldsymbol{\nabla}\cdot\mathbf{j}\right) = \int \mathcal{D}\psi\,e^{-L^d\int_0^\tau dt \int_\Lambda d\mathbf{r}\,\psi(\mathbf{r},t)(\partial_t\rho+\boldsymbol{\nabla}\cdot\mathbf{j})},$$

where $\psi(\mathbf{r},t)$ and $\boldsymbol{\lambda}$ act as auxiliary fields that allow to implement the constraints (as e.g. Lagrange multipliers). In this way the vector current statistics can be written as an extended path integral

$$P_\tau(\mathbf{q}) = \int \mathcal{D}\rho\,\mathcal{D}\mathbf{j}\,\mathcal{D}\psi\,d\boldsymbol{\lambda}\,\exp\left(+L^d\mathcal{I}_\tau[\rho,\mathbf{j},\psi,\boldsymbol{\lambda}]\right) \tag{32}$$

where the modified MFT action reads

$$\mathcal{I}_\tau[\rho,\mathbf{j},\psi,\boldsymbol{\lambda}] = -\int_0^\tau dt \int_\Lambda d\mathbf{r}\left(\frac{\left[\mathbf{j} + D(\rho)\boldsymbol{\nabla}\rho - \sigma(\rho)\mathbf{E}\right]^2}{2\sigma(\rho)} + \psi\left(\partial_t\rho + \boldsymbol{\nabla}\cdot\mathbf{j}\right) + \boldsymbol{\lambda}\cdot[\mathbf{q} - \mathbf{j}]\right). \tag{33}$$

For long times and large system sizes, the probability density function $P_\tau(\mathbf{q})$ obeys a large deviation principle $P_\tau(\mathbf{q}) \asymp \exp[+\tau L^d G(\mathbf{q})]$. A steepest descent (or weak noise, Laplace method, etc.) calculation then leads to the following variational problem for the current LDF,

$$G(\mathbf{q}) = \lim_{\tau\to\infty}\frac{1}{\tau}\max_{\{\rho,\mathbf{j},\psi,\boldsymbol{\lambda}\}_0^\tau}\mathcal{I}_\tau[\rho,\mathbf{j},\psi,\boldsymbol{\lambda}]. \tag{34}$$

## 3.1 Structure of the optimal path in $d > 1$

The set $\left(\rho_{\mathbf{q}},\mathbf{j}_{\mathbf{q}},\psi_{\mathbf{q}},\boldsymbol{\lambda}_{\mathbf{q}}\right)$ of optimal fields which solve the variational problem (34) define *the most probable path* leading to a vector current fluctuation $\mathbf{q}$. Equations for these optimal fields can be derived now by functional differentiating the modified action (33) with respect to the

different fields. For instance, by varying over the density field, $\rho(\mathbf{r}, t) \rightarrow \rho(\mathbf{r}, t) + \delta\rho(\mathbf{r}, t)$, and expanding the resulting functional as in Eq. (14) above, we arrive at the following partial differential equation (note that, as in $1d$, integration by parts is needed to get rid of the terms with $\boldsymbol{\nabla}\delta\rho(\mathbf{r}, t)$)

$$\partial_t \psi_{\mathbf{q}} = -\left(\frac{D_{\mathbf{q}}^2}{2\sigma_{\mathbf{q}}}\right)' (\boldsymbol{\nabla}\rho_{\mathbf{q}})^2 - \left(\frac{D_{\mathbf{q}}^2}{\sigma_{\mathbf{q}}}\right)\boldsymbol{\nabla}^2\rho_{\mathbf{q}} - \left(\frac{D_{\mathbf{q}}}{\sigma_{\mathbf{q}}}\right)\boldsymbol{\nabla}\cdot\mathbf{j}_{\mathbf{q}} - \frac{\sigma_{\mathbf{q}}'}{2\sigma_{\mathbf{q}}^2}\mathbf{j}_{\mathbf{q}}^2 + \frac{\sigma_{\mathbf{q}}'}{2}\mathbf{E}^2 , \qquad (35)$$

where we have defined $D_{\mathbf{q}} = D(\rho_{\mathbf{q}})$ and $\sigma_{\mathbf{q}} = \sigma(\rho_{\mathbf{q}})$. Functional differentiating Eq. (33) with respect to the current field, i.e. perturbing $\mathbf{j}(\mathbf{r}, t) \rightarrow \mathbf{j}(\mathbf{r}, t) + \delta\mathbf{j}(\mathbf{r}, t)$, leads to

$$\mathbf{j}_{\mathbf{q}} + D_{\mathbf{q}}\boldsymbol{\nabla}\rho_{\mathbf{q}} - \sigma_{\mathbf{q}}\mathbf{E} = \sigma_{\mathbf{q}}\left(\boldsymbol{\lambda}_{\mathbf{q}} + \boldsymbol{\nabla}\psi_{\mathbf{q}}\right) \qquad (36)$$

where the lhs is just the optimal excess current $\mathcal{J}_{\mathbf{q}}$ defined above. Finally, variations over the auxiliary field $\psi(\mathbf{r}, t) \rightarrow \psi(\mathbf{r}, t) + \delta\psi(\mathbf{r}, t)$ impose the continuity constraint $\partial_t\rho_{\mathbf{q}} + \boldsymbol{\nabla}\cdot\mathbf{j}_{\mathbf{q}} = 0$, while variations ove $\boldsymbol{\lambda}$ lead to the constraint on the empirical current $\mathbf{q} = \tau^{-1}\int_0^\tau dt \int_\Lambda d\mathbf{r}\mathbf{j}_{\mathbf{q}}(\mathbf{r}, t)$.

The above equations can be interpreted in simple terms to understand the physical meaning of the optimal auxiliary fields $\boldsymbol{\lambda}_{\mathbf{q}}$ and $\psi_{\mathbf{q}}$ solution of this variational problem. Indeed, we can use the local Einstein formula $D(\rho) = \sigma(\rho)f_0''(\rho)$, relating the diffusivity and mobility transport coefficients via the system equilibrium free-energy density $f_0(\rho)$, to write Fick's law under external driving for the optimal fields as

$$-D_{\mathbf{q}}\boldsymbol{\nabla}\rho_{\mathbf{q}} + \sigma_{\mathbf{q}}\mathbf{E} = \sigma_{\mathbf{q}}\left[\mathbf{E} - \boldsymbol{\nabla}\left(\frac{\delta\mathcal{F}_0}{\delta\rho_{\mathbf{q}}}\right)\right], \qquad (37)$$

where $\mathcal{F}_0(\rho) = \int_\Lambda d\mathbf{r} f_0(\rho)$ is the equilibrium free energy functional of the system of interest. Using this in Eq. (36) we thus obtain that the optimal current field can be written as

$$\mathbf{j}_{\mathbf{q}} = \sigma_{\mathbf{q}}\left[(\mathbf{E} + \boldsymbol{\lambda}_{\mathbf{q}}) - \boldsymbol{\nabla}\left(\frac{\delta\mathcal{F}_0}{\delta\rho_{\mathbf{q}}} - \psi_{\mathbf{q}}\right)\right]. \qquad (38)$$

In this way, comparing this expression with Eq. (37), it becomes clear that $\boldsymbol{\lambda}_{\mathbf{q}}$ and $\psi_{\mathbf{q}}(\mathbf{r}, t)$ can be interpreted respectively as the additional constant field (added to $\mathbf{E}$) and structured driving (supplementing $-\delta\mathcal{F}_0/\delta\rho_{\mathbf{q}}$) necessary to obtain the current field $\mathbf{j}_{\mathbf{q}}(\mathbf{r}, t)$ within Fick's constitutive law. Alternatively, $\psi_{\mathbf{q}}$ can be seen as the (optimal) Legendre multiplier field selecting those noise realizations compatible with Fick's law and the local conservation law represented by the continuity equation. This can be better seen in the Hamiltonian formulation of the problem [52, 123] where $\psi$ plays the role of the conjugate moment to the density.

Interestingly Eq. (36), or equivalently Eq. (38), sets strong conditions on the structure of the optimal current vector field. In particular, defining now the reduced (optimal) excess current

$$\boldsymbol{\chi}_{\mathbf{q}}(\mathbf{r}, t) \equiv \frac{\mathbf{j}_{\mathbf{q}}(\mathbf{r}, t) + D_{\mathbf{q}}\boldsymbol{\nabla}\rho_{\mathbf{q}}(\mathbf{r}, t) - \sigma_{\mathbf{q}}\mathbf{E}}{\sigma_{\mathbf{q}}}, \qquad (39)$$

we have from Eq. (36) that $\boldsymbol{\chi}_{\mathbf{q}}(\mathbf{r}, t) = \boldsymbol{\lambda}_{\mathbf{q}} + \boldsymbol{\nabla}\psi_{\mathbf{q}}$, with $\boldsymbol{\lambda}_{\mathbf{q}}$ a constant vector. Hence taking now the Jacobian matrix $\boldsymbol{\nabla}\boldsymbol{\chi}_{\mathbf{q}}$, with components $(\boldsymbol{\nabla}\boldsymbol{\chi}_{\mathbf{q}})_{\alpha\beta} = \partial_\alpha\chi_{\mathbf{q},\beta}$, we have that $\boldsymbol{\nabla}\boldsymbol{\chi}_{\mathbf{q}} = \boldsymbol{\nabla}\boldsymbol{\nabla}\psi_{\mathbf{q}}$, or equivalently

$$\partial_\alpha\chi_{\mathbf{q},\beta} = \partial_\alpha\partial_\beta\psi_{\mathbf{q}}. \qquad (40)$$

In words, this means that the Jacobian matrix of the reduced (optimal) excess vector current $\boldsymbol{\chi}_{\mathbf{q}}$ corresponds to the Hessian of the optimal driving field $\psi_{\mathbf{q}}$ associated to the continuity equation. This apparently innocent observation leads however to a strong result. Indeed, if the optimal response function $\psi_{\mathbf{q}} : \Lambda^d \times [0, \tau] \rightarrow \mathbb{R}$ is sufficiently smooth, i.e. it is twice

continuously differentiable in its spatial domain (a $C^2$-class function of spatial coordinates), then by virtue of Schwarz's theorem [124] its Hessian matrix $\boldsymbol{\nabla}\boldsymbol{\nabla}\psi_{\mathbf{q}}$ is symmetric, so

$$\partial_\alpha \partial_\beta \psi_{\mathbf{q}} = \partial_\beta \partial_\alpha \psi_{\mathbf{q}}, \tag{41}$$

with $\forall \alpha, \beta \in [1, d]$. This immediately implies, via Eq. (40), that the Jacobian of the reduced (optimal) excess current is itself a symmetric matrix, i.e. $\partial_\alpha \chi_{\mathbf{q},\beta} = \partial_\beta \chi_{\mathbf{q},\alpha}$. From this symmetry, and using the definition of $\boldsymbol{\chi}_{\mathbf{q}}$ in Eq. (39), and the symmetry relation

$$\partial_\alpha \left(\frac{D_{\mathbf{q}}}{\sigma_{\mathbf{q}}}\right)\partial_\beta \rho_{\mathbf{q}} = \partial_\beta \left(\frac{D_{\mathbf{q}}}{\sigma_{\mathbf{q}}}\right)\partial_\alpha \rho_{\mathbf{q}} = \left(\frac{D_{\mathbf{q}}}{\sigma_{\mathbf{q}}}\right)' \partial_\alpha \rho_{\mathbf{q}} \partial_\beta \rho_{\mathbf{q}}, \tag{42}$$

we immediately arrive at the following fundamental property for the optimal current field responsible of a given current fluctuation,

$$\partial_\beta \left(\frac{j_{\alpha,\mathbf{q}}(\mathbf{r},t)}{\sigma_{\mathbf{q}}}\right) = \partial_\alpha \left(\frac{j_{\beta,\mathbf{q}}(\mathbf{r},t)}{\sigma_{\mathbf{q}}}\right), \qquad \forall (\mathbf{r},t) \in \Lambda^d \times [0, \tau]. \tag{43}$$

In the presence of anisotropy, this relation is modified to $\partial_\beta [j_{\alpha,\mathbf{q}}/(a_\alpha \sigma_{\mathbf{q}})] = \partial_\alpha [j_{\beta,\mathbf{q}}/(a_\beta \sigma_{\mathbf{q}})]$, where $a_\alpha$ are the (constant) components of the anisotropy matrix $\hat{A}$ defined above [53]. Note that the $C^2$-differentiability of the response function $\psi_{\mathbf{q}}$ is a *natural* requirement for most physical solutions to the variational problem (34), though we cannot discard the possible existence of singular, non-differentiable solutions for $\psi_{\mathbf{q}}$ which would violate (43) at singular points. Note also that a weaker condition for $\psi_{\mathbf{q}}$ which nevertheless suffices to ensure the symmetry of its Hessian matrix is that all partial derivatives are themselves differentiable.

To better understand the tight constraints that Eq. (43) imposes on the optimal current vector field $\mathbf{j}_{\mathbf{q}}(\mathbf{r},t)$ in $d > 1$, we note that the dominant paths responsible of a current fluctuation $\mathbf{q}$ in most high-dimensional problems of interest typically exhibit structure (if any) along a *principal direction*, that we denote as $x_\parallel$ [41, 44, 49–51, 53]. In particular this means that

$$\rho_{\mathbf{q}}(\mathbf{r},t) = \rho_{\mathbf{q}}(x_\parallel, t), \qquad \mathbf{j}_{\mathbf{q}}(\mathbf{r},t) = \mathbf{j}_{\mathbf{q}}(x_\parallel, t), \tag{44}$$

where we have decomposed space as $\mathbf{r} = (x_\parallel, \mathbf{x}_\perp)$ along the parallel $x_\parallel$ and all orthogonal $\mathbf{x}_\perp$ directions. Examples of problems with a well-defined principal direction include open systems subject to a boundary gradient along the $x_\parallel$-direction [50], see e.g. Fig. 1 above, or closed diffusive systems with periodic boundary conditions driven by a constant external field $\mathbf{E}$, for which different dynamic phase transitions appear to current regimes characterized by traveling waves with structure along one of the principal axes of the system of interest [56]. In all these cases, condition (43) simplifies to

$$\partial_\parallel \left(\frac{j_{\beta,\mathbf{q}}}{\sigma_{\mathbf{q}}}\right) = 0 \qquad \forall \beta \neq \parallel,$$

since $\partial_\parallel (j_{\beta,\mathbf{q}}/\sigma_{\mathbf{q}}) = \partial_\beta (j_{\parallel,\mathbf{q}}/\sigma_{\mathbf{q}})$ and $j_{\parallel,\mathbf{q}}(x_\parallel, t)$ depends exclusively on $x_\parallel$ and not on all orthogonal coordinates $\beta \neq \parallel$. This immediately implies that $j_{\beta,\mathbf{q}}(x_\parallel, t) = k_\beta \sigma[\rho_{\mathbf{q}}(x_\parallel, t)] \, \forall \beta \neq \parallel$, with $k_\beta$ a direction-dependent constant which follows from the constraint on the empirical current $\mathbf{q} = \tau^{-1} \int_0^\tau dt \int_\Lambda d\mathbf{r}\, \mathbf{j}_{\mathbf{q}}(x_\parallel, t)$. Imposing this constraint we arrive at

$$j_{\beta,\mathbf{q}}(x_\parallel, t) = q_\beta \frac{\tau \sigma[\rho_{\mathbf{q}}(x_\parallel, t)]}{\int_0^\tau ds \int_0^1 dy\, \sigma[\rho_{\mathbf{q}}(y, s)]} \qquad \forall \beta \neq \parallel, \tag{45}$$

where $q_\beta$ is the $\beta$-coordinate of the empirical current vector $\mathbf{q}$, orthogonal ($\beta \neq \|$) to the principal direction. In this way the relation between the Jacobian matrix for $\boldsymbol{\chi_q}$ and the Hessian matrix of the driving field $\psi_\mathbf{q}$, together with a natural analyticity condition for the latter, force the optimal current vector field $\mathbf{j_q}$ to exhibit a *non-trivial structure* along the dominant direction $\|$ in all its orthogonal components $\beta \neq \|$. Moreover, this structure is coupled to the optimal density field $\rho_\mathbf{q}$ via the mobility transport coefficient $\sigma(\rho_\mathbf{q})$.

Interestingly, this result makes explicit the *spatiotemporal nonlocality* of the current LDF (34) and the associated optimal trajectories. Indeed, the optimal current vector field $\mathbf{j_q}(\mathbf{r}, t)$ at a given point of space and time depends explicitly on the space&time integral of the mobility of the optimal density field, see the denominator in Eq. (45). Note also that for $1d$ systems conditions (43) and (45) become empty, so structureless optimal current fields are still possible, as in the $1d$ additivity principle of section §2.1 [27, 36, 38]. This evidences the richness of the fluctuation landscape for $d > 1$ driven diffusive systems when compared with their $1d$ counterparts.

## 3.2   Time-independent optimal path and weak additivity principle

The main result in the previous section, Eq. (43), is very general since it assumes only a natural analiticity condition for the optimal driving field $\psi_\mathbf{q}$ in generic $d$-dimensional systems. When supplemented with the additional observation that optimal paths typically exhibit structure along a single principal direction, this results leads to a more detailed, but still very general condition (45). In both cases we have not assumed anything on the *temporal structure* of the optimal path leading to a current fluctuation. In this section we explore the implications of these results whenever the optimal path is time-independent. This discussion will thus allow us to find the natural generalization of the additivity principle of section §2 to general $d > 1$ driven diffusive media, a conjecture called weak additivity principle (wAP) [50, 53] to distinguish it from a strong version of the additivity principle in $d$ dimensions that will be discussed below.

We will mostly consider here the case of open systems under an external gradient along an arbitrary direction $x_\|$. Hence we fix the boundary densities to $\rho(\mathbf{r}, t)|_{x_\|=0,1} = \rho_{0,1}$, which drive the system out of equilibrium as soon as $\rho_0 \neq \rho_1$, setting periodic boundary conditions for all other directions of space, see Fig. 1. Current fluctuations in this class of systems have been broadly studied during the last years, both in $1d$ [27, 36, 38, 40, 44] and $d > 1$ [41, 49–51, 53, 125]. Assuming now that the most probable trajectory to a current fluctuation (i) exhibits structure only along the principal direction $x_\|$, and (ii) is time-independent (apart from some initial and final transients of negligible weight for the current LDF), we thus have that $\rho_\mathbf{q} = \rho_\mathbf{q}(x_\|)$ and $\mathbf{j_q} = \mathbf{j_q}(x_\|)$. In this case, the continuity constraint $\partial_t \rho_\mathbf{q} + \boldsymbol{\nabla} \cdot \mathbf{j_q} = 0$ implies a divergence-free optimal current vector field, $\boldsymbol{\nabla} \cdot \mathbf{j_q}(x_\|) = \partial_\| j_{\|,\mathbf{q}}(x_\|) = 0$. These observations, together with our general condition (45) and the constraint on the empirical current $\mathbf{q}$, lead to an optimal current vector field $\mathbf{j_q}(x_\|) = (q_\|, \mathbf{j}_{\perp,\mathbf{q}}(x_\|))$ which is constant ($q_\|$) along the principal direction, and structured ($\mathbf{j}_{\perp,\mathbf{q}}(x_\|)$) along all orthogonal directions, with

$$\mathbf{j}_{\perp,\mathbf{q}}(x_\|) = \mathbf{q}_\perp \frac{\sigma[\rho_\mathbf{q}(x_\|)]}{\displaystyle\int_0^1 dy\, \sigma[\rho_\mathbf{q}(y)]} , \tag{46}$$

and where we have decomposed $\mathbf{q} = (q_\|, \mathbf{q}_\perp)$ along the gradient ($\|$) and all other, ($d-1$) directions ($\perp$) [50, 53]. Our general theorem (43) allows now to understand this structure as a direct consequence of the symmetry of the Jacobian matrix associated to the reduced excess vector current field.

The equation for the optimal density profile $\rho_{\mathbf{q}}(x_{\parallel})$ now follows from the partial differential equation (35) assuming time-independent solutions with structure only along the principal direction $x_{\parallel}$. In particular, since $\partial_t \psi_{\mathbf{q}} = 0 = \mathbf{\nabla} \cdot \mathbf{j}_{\mathbf{q}}$ under these assumptions, we have

$$q_{\parallel}^2 \left( \frac{1}{2\sigma_{\mathbf{q}}} \right)' - \mathbf{q}_{\perp}^2 \frac{\sigma_{\mathbf{q}}'}{2 \left( \int_0^1 dy\, \sigma[\rho_{\mathbf{q}}(y)] \right)^2} + \frac{\mathbf{E}^2}{2} \sigma_{\mathbf{q}}' - \left( \frac{D_{\mathbf{q}}^2}{2\sigma_{\mathbf{q}}} \right)' (\partial_{\parallel}\rho_{\mathbf{q}})^2 - \left( \frac{D_{\mathbf{q}}^2}{\sigma_{\mathbf{q}}} \right) \partial_{\parallel}^2 \rho_{\mathbf{q}} = 0, \quad (47)$$

where we have used that $-\frac{\sigma_{\mathbf{q}}'}{2\sigma_{\mathbf{q}}^2} = (\frac{1}{2\sigma_{\mathbf{q}}})'$ and $\mathbf{j}_{\mathbf{q}}^2(x_{\parallel}) = q_{\parallel}^2 + \mathbf{q}_{\perp}^2 \sigma_{\mathbf{q}}^2 / (\int_0^1 dy\, \sigma[\rho_{\mathbf{q}}(y)])^2$, see Eq. (46). Multiplying this equation by $\partial_{\parallel}\rho_{\mathbf{q}}$, and noting that

$$(\frac{D_{\mathbf{q}}^2}{2\sigma_{\mathbf{q}}})'(\partial_{\parallel}\rho_{\mathbf{q}})^3 + (\frac{D_{\mathbf{q}}^2}{\sigma_{\mathbf{q}}})(\partial_{\parallel}^2\rho_{\mathbf{q}})(\partial_{\parallel}\rho_{\mathbf{q}}) = \partial_{\parallel}[(\partial_{\parallel}\rho_{\mathbf{q}})^2 \frac{D_{\mathbf{q}}^2}{2\sigma_{\mathbf{q}}}],$$

we find

$$\partial_{\parallel} \left[ \frac{q_{\parallel}^2}{2\sigma_{\mathbf{q}}} - \frac{\mathbf{q}_{\perp}^2 \sigma_{\mathbf{q}}}{2\left( \int_0^1 dy\, \sigma[\rho_{\mathbf{q}}(y)] \right)^2} + \frac{\mathbf{E}^2}{2}\sigma_{\mathbf{q}} - (\partial_{\parallel}\rho_{\mathbf{q}})^2 \frac{D_{\mathbf{q}}^2}{2\sigma_{\mathbf{q}}} \right] = 0,$$

so the expression in brackets is just a constant $K = K(q_{\parallel}^2, \mathbf{q}_{\perp}^2, \mathbf{E}^2)$ which depends exclusively on $q_{\parallel}^2$, $\mathbf{q}_{\perp}^2$ and $\mathbf{E}^2$. Therefore the final differential equation for the optimal density profile under the weak additivity principle hypotheses is

$$D(\rho_{\mathbf{q}})^2(\partial_{\parallel}\rho_{\mathbf{q}})^2 = q_{\parallel}^2 - \mathbf{q}_{\perp}^2 \left( \frac{\sigma(\rho_{\mathbf{q}})}{\int_0^1 dy\, \sigma[\rho_{\mathbf{q}}(y)]} \right)^2 + 2K\sigma(\rho_{\mathbf{q}}) + \mathbf{E}^2\sigma(\rho_{\mathbf{q}})^2, \quad (48)$$

where we have recovered the original notation $D(\rho_{\mathbf{q}}) = D_{\mathbf{q}}$ and $\sigma(\rho_{\mathbf{q}}) = \sigma_{\mathbf{q}}$. This equation for the optimal density profile responsible of a vector current fluctuation $\mathbf{q}$ should be compared with Eq. (18) for $1d$ systems under the standard additivity principle to better grasp the effect of dimensionality on current statistics.

The current large deviation function (34) under the wAP conjecture, denoted here as $G_{\text{wAP}}(\mathbf{q})$, can be thus written as

$$G_{\text{wAP}}(\mathbf{q}) = -\int_0^1 dx_{\parallel} \frac{1}{2\sigma_{\mathbf{q}}} \left[ \left( q_{\parallel} + D_{\mathbf{q}}\partial_{\parallel}\rho_{\mathbf{q}}^{\text{wAP}} - \sigma_{\mathbf{q}}E_{\parallel} \right)^2 + \sigma_{\mathbf{q}}^2 \left( \frac{\mathbf{q}_{\perp}}{\int_0^1 dy\, \sigma[\rho_{\mathbf{q}}^{\text{wAP}}(y)]} - \mathbf{E}_{\perp} \right)^2 \right],$$
$$(49)$$

where we have decomposed $\mathbf{E} = (E_{\parallel}, \mathbf{E}_{\perp})$, $\mathbf{j}_{\mathbf{q}}(x_{\parallel}) = (q_{\parallel}, \mathbf{j}_{\perp,\mathbf{q}}(x_{\parallel}))$ with $\mathbf{j}_{\perp,\mathbf{q}}(x_{\parallel})$ given in Eq. (46), and with $\rho_{\mathbf{q}}^{\text{wAP}}(x_{\parallel})$ the solution to Eq. (48) above with boundary conditions $\rho_{\mathbf{q}}^{\text{wAP}}(0) = \rho_0$ and $\rho_{\mathbf{q}}^{\text{wAP}}(1) = \rho_1$. In this way, the current LDF can be written as $G_{\text{wAP}}(\mathbf{q}) = \mathcal{F}_{\text{wAP}}(\rho_{\mathbf{q}}^{\text{wAP}}; \mathbf{q})$, where $\mathcal{F}_{\text{wAP}}$ is the functional of Eq. (49).

## 3.3   A strong version of the additivity principle?

The results for the current LDF and the optimal path in the previous subsection should be compared with the *straightforward* extension of the $1d$ additivity principle to $d > 1$, dubbed here *strong additivity principle* or sAP in short [50]. This amounts to assume (i) a time-independent optimal path to a current fluctuation, (ii) structure only along a principal direction, and crucially (iii) a *structureless* optimal current vector field, so $\mathbf{j}_{\mathbf{q}}(x_{\parallel}) = \mathbf{q}$ due to the constraint on the empirical current. This last assumption is the key difference with the weak version of the

additivity principle (wAP) discussed above. This strong additivity conjecture thus leads to a different (and simpler) expression for the current LDF,

$$G_{\text{sAP}}(\mathbf{q}) = -\int_0^1 dx_\parallel \frac{1}{2\sigma_{\mathbf{q}}} \left[ \left( q_\parallel + D_{\mathbf{q}} \partial_\parallel \rho_{\mathbf{q}}^{\text{sAP}} - \sigma_{\mathbf{q}} E_\parallel \right)^2 + \left( \mathbf{q}_\perp - \sigma_{\mathbf{q}} \mathbf{E}_\perp \right)^2 \right] \equiv \mathcal{F}_{\text{sAP}}(\rho_{\mathbf{q}}^{\text{sAP}}; \mathbf{q}), \quad (50)$$

where now $D_{\mathbf{q}} = D[\rho_{\mathbf{q}}^{\text{sAP}}(x_\parallel)]$ and $\sigma_{\mathbf{q}} = \sigma[\rho_{\mathbf{q}}^{\text{sAP}}(x_\parallel)]$, with $\rho_{\mathbf{q}}^{\text{sAP}}(x_\parallel)$ the solution to the following differential equation

$$D(\rho_{\mathbf{q}})^2 (\partial_\parallel \rho_{\mathbf{q}})^2 = \mathbf{q}^2 + 2K\sigma(\rho_{\mathbf{q}}) + \sigma(\rho_{\mathbf{q}})^2 \mathbf{E}^2, \quad (51)$$

with $K = K(\mathbf{q}^2, \mathbf{E}^2)$ another integration constant, fixed by imposing one of the boundary conditions for the optimal density profile. Note that, for $\mathbf{q}$ fixed, we expect $G_{\text{sAP}}(\mathbf{q}) \neq G_{\text{wAP}}(\mathbf{q})$ and $\rho_{\mathbf{q}}^{\text{sAP}}(x_\parallel) \neq \rho_{\mathbf{q}}^{\text{wAP}}(x_\parallel)$ in general, and we could question which hypothesis (wAP or sAP) yields a maximal current LDF, see Eq. (34). Intuition suggests that the wAP should offer a better solution as it disposes of additional degrees of freedom (a possibly structured but divergence-free optimal current field) that the system of interest can employ to improve its rate function. This argument can be confirmed rigorously by noting first that $\rho_{\mathbf{q}}^{\text{wAP}}(x_\parallel)$ is the maximizer of the wAP action $\mathcal{F}_{\text{wAP}}(\rho_{\mathbf{q}}^{\text{wAP}}; \mathbf{q})$, and therefore $\mathcal{F}_{\text{wAP}}(\rho_{\mathbf{q}}^{\text{wAP}}; \mathbf{q}) \geq \mathcal{F}_{\text{wAP}}(\varphi; \mathbf{q}) \,\forall \varphi(x_\parallel) \neq \rho_{\mathbf{q}}^{\text{wAP}}(x_\parallel)$. We can now compare both functionals, $\mathcal{F}_{\text{wAP}}$ and $\mathcal{F}_{\text{sAP}}$, applied to *the same* density profile $\rho_{\mathbf{q}}^{\text{sAP}}$ at fixed current $\mathbf{q}$. For that we define the excess observable $\Delta \mathcal{F}_{\mathbf{q}} \equiv \mathcal{F}_{\text{wAP}}(\rho_{\mathbf{q}}^{\text{sAP}}; \mathbf{q}) - \mathcal{F}_{\text{sAP}}(\rho_{\mathbf{q}}^{\text{sAP}}; \mathbf{q})$ and find

$$\Delta \mathcal{F}_{\mathbf{q}} = \frac{\mathbf{q}_\perp^2}{2} \left[ \left( \int_0^1 dx_\parallel \frac{1}{\sigma[\rho_{\mathbf{q}}^{\text{sAP}}(x_\parallel)]} \right) - \left( \frac{1}{\int_0^1 dx_\parallel \, \sigma[\rho_{\mathbf{q}}^{\text{sAP}}(x_\parallel)]} \right) \right]. \quad (52)$$

Notice that, due to the reverse Hölder's inequality [126]

$$\int_0^1 dx_\parallel \frac{1}{\sigma[\rho_{\mathbf{q}}^{\text{sAP}}(x_\parallel)]} \geq \frac{1}{\int_0^1 dx_\parallel \, \sigma[\rho_{\mathbf{q}}^{\text{sAP}}(x_\parallel)]}, \quad (53)$$

and therefore $\Delta \mathcal{F}_{\mathbf{q}} \geq 0$ and hence $G_{\text{wAP}}(\mathbf{q}) \geq G_{\text{sAP}}(\mathbf{q}) \,\forall \mathbf{q} \in \mathbb{R}^d$. This proves that, when compared to the strong AP, the weak AP always yields a better minimizer of the MFT action for currents, see Eq. (34). This result therefore singles out the wAP as the relevant simplifying hypothesis to study current statistics in general $d$-dimensional systems. Interestingly, the excess action $\Delta \mathcal{F}_{\mathbf{q}}$ is proportional to $\mathbf{q}_\perp^2$, so both the sAP and wAP yield the same results for current fluctuations parallel to the gradient direction, $\mathbf{q} = (q_\parallel, \mathbf{q}_\perp = 0)$. This observation helps in making sense of previous, seemingly contradictory results regarding the validity of the strong additivity conjecture in $d$-dimensional driven diffusive systems [41,49,51,125]. Note also that the sAP and wAP also yield the same results for all vector currents $\mathbf{q}$ in the case of constant mobility transport coefficient, $\sigma(\rho) = \sigma$.

The predictions for current statistics obtained within the weak additivity principle scenario have been tested in detail against both numerical simulations of rare events and microscopic exact calculations of various paradigmatic models of diffusive transport in $d = 2$ [50]. These include the widely-studied Zero Range Process (ZRP) [127, 128], a model of interacting particles amenable to exact computations due to a factorization property of its stationary measure, and characterized at the hydrodynamic level (for constant particle hopping rates) by a diffusivity $D(\rho) = 1/[2(1 + \rho)^2]$ and a mobility $\sigma(\rho) = \rho/(1 + \rho)$ [50]. Another model where the predictions of the wAP conjecture have been confirmed with high precission is the Kipnis-Marchioro-Presutti model of heat transport [119], a diffusive lattice process including random

energy exchanges between nearest neighbors, and described at the macroscopic level by a fluctuating hydrodynamic equation (27) obeying Fourier's law with a constant conductivity $D(\rho) = 1/2$ and a mobility $\sigma(\rho) = \rho^2$. Finally, another model tested against the wAP predictions is a fluid of random walkers in $2d$, characterized macroscopically by a constant diffusivity $D(\rho) = 1/2$ and a linear mobility $\sigma(\rho) = \rho$. In all cases the results clearly demonstrate that the weak additivity principle yields the correct predictions for the current statistics of a broad class of $d$-dimensional interacting particle systems.

## 4    Dynamical symmetry breaking in the current fluctuations of open systems

Driven diffusive systems may undergo intriguing transitions to sustain rare fluctuations of a trajectory-dependent observable. These so-called dynamical phase transitions (DPTs) have gained attention in the last two decades [28, 29, 108, 109, 129–132], and can lead in some cases to symmetry-broken spatiotemporal trajectories which enhance the probability of rare fluctuations. These drastic changes in the structure of the optimal trajectories responsible for a fluctuation, triggered at DPTs, are accompanied by non-analyticities and Lee-Yang singularities in the associated large deviation functions, which play the role of thermodynamic potentials for nonequilibrium settings [8]. This is very much reminiscent of standard critical phenomena in condensed matter, but at the trajectory (rather than configurational) level.

Investigating the physics of DPTs is a fruitful endeavor. On one hand, the mere existence of DPTs and spontaneous symmetry-breaking phenomena at the fluctuation level draws the attention of any curious physicist. On the other hand, the appearance of DPTs make some rare events far more likely than anticipated, due to the emergence of self-organized structures in the optimal paths responsible of these fluctuations. Thus studying DPTs may help in understanding why some naturally-occurring rare events (as e.g. rogue waves [133]) appear with higher probability than expected. Finally, DPTs can be engineered to build interesting phases in nonequilibrium matter, as e.g. time crystals [101]. We will explore this possibility later on in these lecture notes. This interest has led to the discovery of multiple emergent phenomena associated with DPTs, including symmetry-breaking density profiles [52, 71, 109], localization effects [134], condensation transitions [57] or traveling waves [14, 43, 56] displaying time-crystalline order [101]. Interestingly, DPTs have been shown to play a key role in understanding the physics of different systems, as e.g. glass formers [69, 80–86]. Moreover, DPTs have been also predicted and observed in active media [135–145], where constituent particles can consume free-energy to produce directed motion, as well as in many different open quantum systems [77, 87, 88, 146–151], leading e.g. to applications such as DPT-based quantum thermal switches [89–91].

From a macroscopic perspective, the existence of DPTs in driven diffusive media is governed by the MFT action functional (7), and depends on the particular form of the transport coefficients characterizing the system, namely the diffusivity and the mobility [28, 29, 52, 129]. A different, complementary path to investigate the physics of DPTs consists in analyzing them in terms of the microscopic dynamics, governed by the corresponding stochastic generator [107, 147]. We will initially focus here on a macroscopic perspective, to explore later on the microscopic view using spectral theory [102].

In this section we will review a family of DPTs in the current statistics of some open systems, i.e. systems in contact with boundary reservoirs (possibly at different densities) [52, 71, 109]. In order to simplify the discussion and focus on the main mechanism behind these DPTs, we now go back to one-dimensional (1$d$) systems, though many of the results and techniques here discussed can be extended to high-dimensional systems, taking into account some of

the caveats already discussed in the previous section. Our theoretical framework is again macroscopic fluctuation theory for $1d$ driven diffusive media described by a fluctuating hydrodynamic equation $\partial_t \rho + \partial_x j = 0$ for a density field $\rho(x, t)$, with a fluctuating current $j(x, t) = -D(\rho)\partial_x \rho(x, t) + \sigma(\rho)E + \xi(x, t)$, see Eqs. (3)-(5) above [2]. The probability of a trajectory $\{\rho, j\}_0^\tau$ of duration $\tau$ scales for large system sizes $L$ as $P(\{\rho, j\}_0^\tau) \asymp \exp(+L \mathcal{I}_\tau[\rho, j])$, see Eq. (6), with the MFT action [2]

$$\mathcal{I}_\tau[\rho, j] = -\int_0^\tau dt \int_0^{1^*} dx \frac{\left(j + D(\rho)\partial_x \rho - \sigma(\rho)E\right)^2}{2\sigma(\rho)}, \tag{54}$$

see Eq. (7), where the $^*$ means that the integral is restricted to fields $\rho$ and $j$ obeying the continuity constraint (otherwise $\mathcal{I}_\tau[\rho, j] \to -\infty$). As discussed in section §2, the probability of an empirical current fluctuation $q = \tau^{-1}\int_0^\tau dt \int_0^1 j(x, t)$ scales according to the large deviation principle $P_\tau(q) \asymp \exp[+\tau L G(q)]$. The current large deviation function $G(q)$ can be obtained from a variational problem involving the MFT action (54), see Eq. (11) above. Under the additivity principle [27], which amounts to assume that the optimal trajectory solution of the variational problem (11) is time-independent (except maybe for some short, negligible initial and final transients), the current LDF can be thus written as

$$G(q) = -\min_{\rho(x)} \int_0^1 dx \frac{[q + D(\rho)\partial_x \rho - \sigma(\rho)E]^2}{2\sigma(\rho)}, \tag{55}$$

see also Eq. (13) above and the associated discussion. The optimal density profile responsible of a given current fluctuation now follows from the following ordinary differential equation (18), i.e.

$$D^2(\rho)\left(\frac{d\rho(x)}{dx}\right)^2 = q^2 + 2K\sigma(\rho) + E^2\sigma^2(\rho), \tag{56}$$

where $K(q^2, E^2)$ is an integration constant which is fixed by one of the boundary conditions for the density field, $\rho(0) = \rho_0$ or $\rho(1) = \rho_1$, the other one being used as boundary condition for this first-order ODE, see also Eq. (18).

Obtaining general predictions for the optimal profile and the current LDF is typically difficult. To make further progress, we consider now open diffusive systems with a discrete *particle-hole (PH) symmetry*. A system has a particle-hole symmetry when its dynamics is invariant under the transformation

$$x \to \tilde{x} = 1 - x, \qquad \rho \to \tilde{\rho} = 1 - \rho, \tag{57}$$

i.e. after exchanging particles and holes and reflecting space. By looking at the fluctuating hydrodynamics equation governing the system evolution, $\partial_t \rho + \partial_x [-D(\rho)\partial_x \rho + \sigma(\rho)E + \xi] = 0$, and noting that the PH transformation (57) implies that $\partial_x \to -\partial_{\tilde{x}}$, as well as $\partial_t \rho \to -\partial_t \tilde{\rho}$ and $\partial_x \rho \to +\partial_{\tilde{x}}\tilde{\rho}$, it is clear that the PH-symmetry holds at the dynamical level only if the diffusivity and mobility transport coefficients remain invariant under the transformation, i.e.

$$D(\rho) = D(1 - \rho), \qquad \sigma(\rho) = \sigma(1 - \rho). \tag{58}$$

Moreover, in order for the dynamics to be fully invariant under the PH transformation, we need the boundary conditions for the density field to obey the relation

$$\rho_0 = 1 - \rho_1, \tag{59}$$

so that $\tilde{\rho}_0 = \tilde{\rho}(\tilde{x} = 0) = 1 - \rho(x = 1) = 1 - \rho_1 = \rho_0$ and similarly $\tilde{\rho}_1 = \rho_1$. If these two conditions (58)-(59) hold, the system will be PH-symmetric and its dynamics remains invariant

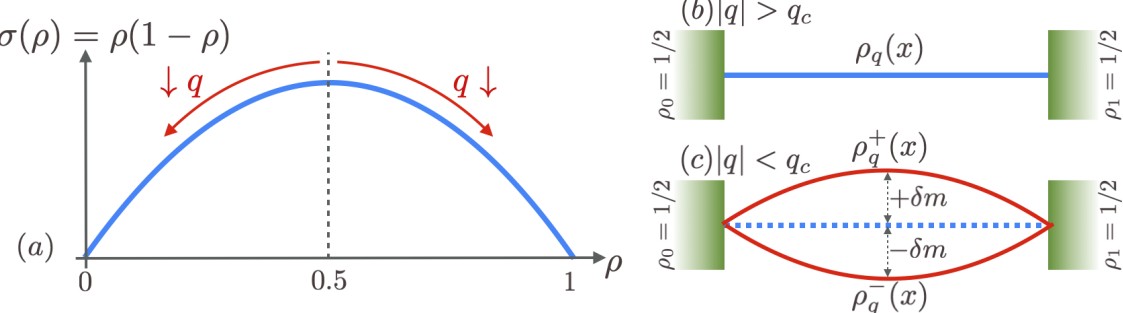

Figure 2: **Dynamical symmetry breaking in open systems.** (a) Mobility $\sigma(\rho) = \rho(1-\rho)$ for the WASEP model of particle diffusion under exclusion interactions. Note the particle-hole symmetry $\sigma(\rho) = \sigma(1-\rho)$ of this transport coefficient. For equal boundary densities, $\rho_0 = \frac{1}{2} = \rho_1$, the optimal density profile $\rho_q(x)$ for mild current fluctuations around the average current $\langle q \rangle$ is just homogeneous, $\rho_q(x) = \frac{1}{2}$, see panel (b), and is symmetric under the PH transformation (57). The current field is proportional to the mobility, and for $\rho = \frac{1}{2}$ this current can decrease by either increasing or decreasing the density, see red arrows in panel (a). In this way, for currents below a critical threshold $q_c$, see panel (c), two different but equally likely optimal profiles $\rho_q^\pm(x)$ emerge, with an excess mass $\pm\delta m$ when compared to the flat profile, breaking the PH symmetry of the governing action. These two different optimal profiles map onto each other under the PH transformation (57).

under the PH transformation (57). Moreover, it is easy to show that the MFT action (54) for a path $\{\rho, j\}_0^\tau$ also remains invariant under the PH transformation whenever these two conditions (58)-(59) hold, so $\mathcal{I}_\tau[\rho, j] = \mathcal{I}_\tau[\tilde{\rho}, \tilde{j}]$ under the PH transformation (57) (with $\tilde{j} = j$ due to the current symmetry).

The optimal trajectory responsible of a given current fluctuation, $\{\rho_q(x, t), j_q(x, t)\}_0^\tau$ in general or $\{\rho_q(x), q\}_0^\tau$ under the additivity principle, will *typically* inherit the PH symmetry of the governing MFT action, mapping onto itself under the PH transformation. For the additivity case

$$\rho_q(x) \xrightarrow[\tilde{\rho}=1-\rho]{\tilde{x}=1-x} \tilde{\rho}_q(\tilde{x}) = \rho_q(x). \tag{60}$$

However, there exists the possibility that the optimal profile $\rho_q(x)$ (which plays here the role of a *ground state*) exhibits fewer symmetries than the governing action, very much like in in many other spontaneous symmetry-breaking phenomena and phase transitions. This spontaneous symmetry breaking phenomenon at the fluctuation level is the focus of this section.

To understand phenomenologically the mechanism behind this symmetry-breaking phenomenon, we now focus on a particular example taken from the works of Y. Baek, Y. Kafri and V. Lecomte [52, 71]: current fluctuations in the $1d$ open weakly asymmetric simple exclusion process (WASEP) [109]. The WASEP is a model of particle diffusion under exclusion interactions, characterized at the hydrodynamic level by a constant diffusivity $D(\rho) = 1/2$ and a quadratic mobility $\sigma(\rho) = \rho(1-\rho)$, see Fig. 2.a. In this way the WASEP transport coefficients fulfill condition (58), and for appropriate boundary conditions, see Eq. (59), this model hence exhibits a PH symmetry as defined above. For simplicity, we consider now equal density boundary reservoirs, $\rho_0 = 1/2 = \rho_1$, for which condition (59) holds trivially, though boundary density gradients fulfilling this condition also fit in the same picture, see below. In this particular case the stationary density profile is just homogeneous, $\langle\rho(x)\rangle = 1/2 \equiv \bar{\rho}$, and the average current is simply $\langle q \rangle = \sigma(\bar{\rho})E = \bar{\rho}(1-\bar{\rho})E$. To understand the optimal way to sustain a fluctuation of the empirical current $q = \tau^{-1}\int_0^\tau dt \int_0^1 j(x, t)$ we now look at the current

field $j(x,t) = -D(\rho)\partial_x\rho + \sigma(\rho)E + \xi$ in the fluctuating hydrodynamic equation governing the system evolution. The diffusion term $-D(\rho)\partial_x\rho$ favors the flattening of any density field modulation, and cancels out for the homogeneous stationary profile. We thus expect that for small current fluctuations around the average flux, $|q - \langle q \rangle| \ll 1$, the optimal density profile will still be the homogeneous, stationary one, and the current statistics will be thus Gaussian, see Fig. 2.b. For quasi-homogeneous density profiles the local current field is dominated by the mobility $\sigma(\rho)$, so obtaining low enough currents $q$ is easier if $\sigma(\rho)$ is small. For $\bar{\rho} = \frac{1}{2}$, this can be achieved by either increasing or decreasing the density, see red arrows in Fig. 2.a. In this way, for low enough currents $q$ (indeed for currents below a critical threshold $q_c$), two different but equally likely optimal profiles $\rho_q^\pm(x)$ may emerge, with an excess mass $\pm\delta m$ when compared to the flat profile,

$$\pm\delta m = \left(\int_0^1 \rho_q^\pm(x)\,dx\right) - \bar{\rho}\,, \tag{61}$$

breaking the PH symmetry of the governing action. These two different optimal profiles map onto each other under the PH transformation (57),

$$\rho_q^\pm(x) \xrightarrow[\rho \to 1-\rho]{x \to 1-x} 1 - \rho_q^\mp(1-x)\,, \tag{62}$$

restoring the symmetry of the original action. This sort of dynamical PH-symmetry breaking makes indeed physical sense, since in order to sustain a low-current fluctuation the system can react by either crowding with particles hence hindering motion, or rather emptying the lattice to minimize particle flow. Both tendencies break the PH symmetry of the MFT action, eventually triggering a DPT.

## 4.1 Computing the critical current $q_c$: A local stability analysis

In this section we will compute the critical threshold $q_c$ to observe the above spontaneous symmetry breaking phenomenon in the current statistics of PH-symmetric open diffusive media. To simplify the calculation and get an overview of the mechanism, we will assume the simplest case of equal and PH-symmetric boundary densities, $\rho_0 = 1/2 = \rho_1$ in a generic system with diffusivity $D(\rho)$ and mobility $\sigma(\rho)$ obeying the PH symmetry condition (58). In this case the system steady state is homogeneous, with $\langle\rho(x)\rangle = 1/2 \equiv \bar{\rho}$ and average current $\langle q \rangle = \sigma(\bar{\rho})E$. The current LDF $G(q)$ now follows from the variational problem defined in Eq. (55) under the additivity conjecture, and the optimal profile $\rho_q(x)$ is just the solution of the ordinary differential equation (56), or its *mother* equation (17). Clearly the flat profile $\rho_q(x) = \bar{\rho}$ is always a solution of Eq. (17). As argued above, for small current fluctuations around the average, $|q - \langle q \rangle| \ll 1$, the optimal density profile will still be homogeneous, and the current LDF will be quadratic (corresponding to Gaussian current statistics in this regime), i.e.

$$G(q) = -\frac{(q - \sigma(\bar{\rho})E)^2}{2\sigma(\bar{\rho})} \equiv G_{\text{flat}}(q)\,, \tag{63}$$

see Eq. (55). The flat profile $\rho_q(x) = \bar{\rho}$ remains obviously invariant under the PH transformation (57), thus having the same symmetries that the MFT path action (54).

We now consider the stability of the flat optimal density profile against small perturbations. In particular, we assume

$$\rho_q(x) = \bar{\rho} + \varphi(x) \tag{64}$$

with $|\varphi(x)| \ll 1$ a spatially-structured perturbation with $\varphi(0) = 0 = \varphi(1)$ to comply with boundary conditions. Our aim is to study the effect of this perturbation on the current LDF (55)

to determine whether it *improves* the flat density profile prediction (63), i.e. whether the perturbed profile yields a larger $G(q) \geq G_{\text{flat}}(q)$. We hence write

$$G(q) = -\int_0^1 dx \, \frac{[q + D(\bar{\rho} + \varphi)\partial_x \varphi - \sigma(\bar{\rho} + \varphi)E]^2}{2\sigma(\bar{\rho} + \varphi)} \underset{|\varphi| \ll 1}{\approx} G_{\text{flat}}(q) + \Delta G(q), \qquad (65)$$

and we will conclude that the flat profile becomes *unstable* (i.e. not the optimal solution) whenever $\Delta G(q) \geq 0$ as defined above. In order to proceed, we expand the transport coefficients

$$D(\bar{\rho} + \varphi) \underset{|\varphi| \ll 1}{\approx} \bar{D} + \bar{D}'\varphi(x) + \frac{\bar{D}''}{2}\varphi^2(x) = \bar{D} + \frac{\bar{D}''}{2}\varphi^2(x), \qquad (66)$$

$$\sigma(\bar{\rho} + \varphi) \underset{|\varphi| \ll 1}{\approx} \bar{\sigma} + \bar{\sigma}'\varphi(x) + \frac{\bar{\sigma}''}{2}\varphi^2 = \bar{\sigma} + \frac{\bar{\sigma}''}{2}\varphi^2(x), \qquad (67)$$

where we have defined $\bar{D} = D(\bar{\rho})$ and $\bar{\sigma} = \sigma(\bar{\rho})$ (similarly for the derivatives), and we have used the symmetry of $D(\rho)$ and $\sigma(\rho)$ around $\bar{\rho} = 1/2$, see Eq. (58), to cancel the first-order terms in the expansions. Indeed all odd derivatives of the transport coefficients obeying condition (58) are zero around $\bar{\rho} = \frac{1}{2}$, i.e. $D^{(2n+1)}(\bar{\rho}) = 0 = \sigma^{(2n+1)}(\bar{\rho}) \; \forall n \geq 0$, due to the PH symmetry. Using these expansions in the integral kernel in Eq. (65) and expanding the resulting functional to the lowest order we find

$$
\begin{aligned}
G(q) & \underset{|\varphi| \ll 1}{\approx} -\int_0^1 dx \, \frac{\left[q + \left(\bar{D} + \frac{\bar{D}''}{2}\varphi^2(x)\right)\partial_x \varphi - \left(\bar{\sigma} + \frac{\bar{\sigma}''}{2}\varphi^2(x)\right)E\right]^2}{2\left(\bar{\sigma} + \frac{\bar{\sigma}''}{2}\varphi^2(x)\right)} \\
& \approx \; G_{\text{flat}}(q) - \int_0^1 dx \left[\frac{\bar{D}^2}{2\bar{\sigma}}(\partial_x \varphi)^2 - \frac{\bar{\sigma}''}{4\bar{\sigma}^2}\left(q^2 - \bar{\sigma}^2 E^2\right)\varphi^2 + \frac{\bar{D}}{\bar{\sigma}}(q - \bar{\sigma}E)\partial_x \varphi\right].
\end{aligned}
\qquad (68)
$$

We next expand $\varphi(x)$ in Fourier series, compatible with the boundary conditions,

$$\varphi(x) = \sum_{n \geq 1} \varphi_n \sin(n\pi x), \qquad (69)$$

with $\varphi_n$ some (small) amplitudes, so that $\partial_x \varphi = \pi \sum_{n \geq 1} n\varphi_n \cos(n\pi x)$ and

$$
\begin{aligned}
\varphi^2(x) & = \sum_{n \geq 1} \varphi_n^2 \sin^2(n\pi x) + \sum_{\substack{n,m \geq 1 \\ n \neq m}} \varphi_n \varphi_m \sin(n\pi x)\sin(m\pi x), \\
(\partial_x \varphi)^2 & = \pi^2 \left[\sum_{n \geq 1} n^2 \varphi_n^2 \cos^2(n\pi x) + \sum_{\substack{n,m \geq 1 \\ n \neq m}} nm\varphi_n \varphi_m \cos(n\pi x)\cos(m\pi x)\right].
\end{aligned}
$$

In this way, the integrals appearing in the perturbative term $\Delta G(q)$ defined by the second line of Eq. (68) can be trivially solved, in particular $\int_0^1 dx \, \sin(n\pi x)\sin(m\pi x) = \frac{1}{2}\delta_{nm} = \int_0^1 dx \, \cos(n\pi x)\cos(m\pi x)$, so we arrive at

$$\Delta G(q) = \sum_{n \geq 1} \frac{\varphi_n^2}{8\bar{\sigma}^2}\left[\bar{\sigma}''(q^2 - \bar{\sigma}^2 E^2) - 2\pi^2 \bar{D}^2 \bar{\sigma} n^2\right]. \qquad (70)$$

The terms in brackets in the previous expression define some $n$- and $q$-dependent coefficients $A_n(q) \equiv \bar{\sigma}''(q^2 - \bar{\sigma}^2 E^2) - 2\pi^2 \bar{D}^2 \bar{\sigma} n^2$. It turns out that $A_n(q) - A_{n+1}(q) = 2\pi^2 \bar{D}^2 \bar{\sigma}(2n + 1) \geq 0$ because the mobility (and the diffusivity) are always positive, so these coefficients decrease

with increasing $n$, i.e. $A_n(q) \geq A_{n+1}(q) \; \forall n \geq 1$. Therefore, if $A_1(q) \leq 0$ then $A_n(q) \leq 0 \; \forall n \geq 1$ and $\Delta G(q) \leq 0$, meaning that the flat density profile remains the optimal one while $A_1(q) \leq 0$. The first amplitude that can become positive, triggering the instability of the flat profile, is hence $A_1(q)$, and the condition defining the critical current $q_c$ for this instability to happen is thus $A_1(q_c) = 0$, which yields a critical threshold

$$q_c = \pm\sqrt{\bar{\sigma}^2 E^2 + 2\pi^2 \frac{\bar{D}^2 \bar{\sigma}}{\bar{\sigma}''}}\,. \tag{71}$$

At this critical current a dynamical phase transition to a PH-symmetry-broken regime appears, as we will see below, which dominates as far as $A_1(q) \geq 0$.

For systems with a mobility transport coefficient with positive curvature $\bar{\sigma}'' > 0$ at density $\bar{\rho}$ (or equivalently with $\sigma(\rho)$ upward concave at $\bar{\rho}$), the amplitude $A_1(q) \geq 0$ for currents *beyond* the critical current (71), i.e. $|q| \geq |q_c|$ or equivalently in the current interval $q \in (-\infty, -q_c] \cup [q_c, +\infty)$. This means that the PH-symmetry-broken dynamical phase dominates the for current fluctuations $|q| \geq |q_c|$, while the flat optimal profile prevails for $|q| < |q_c|$. Moreover, for $\bar{\sigma}'' > 0$ the terms inside the square-root sign in Eq. (71) are all positive so there exists a critical current for all possible external fields $E$ (including the undriven case $E = 0$).

On the other hand, for systems with a mobility with negative curvature $\bar{\sigma}'' < 0$ at density $\bar{\rho}$ (or equivalently with $\sigma(\rho)$ upward convex at $\bar{\rho}$), the amplitude $A_1(q)$ is positive for currents *below* the critical threshold (71), i.e. for $|q| \leq |q_c|$, so the PH-symmetry-broken regime now appears for $q \in [-q_c, q_c]$. In addition, since $\bar{\sigma}'' < 0$, there exist the possibility that the radicand within the square-root defining the critical current $q_c$ in Eq. (71) becomes negative, leading to a unphysical $q_c$. This results in a critical external field $E_c$ such that for $|E| \leq E_c$ no PH-symmetry-breaking dynamical phase transition is possible. The critical external field can be evaluated from the radicand zeros, leading to

$$E_c = \sqrt{\frac{2\pi^2 \bar{D}^2}{\bar{\sigma}|\bar{\sigma}''|}} \qquad \text{for } \bar{\sigma}'' < 0\,. \tag{72}$$

An example of this case is the WASEP model of particle diffusion under exclusion interactions, where $D(\rho) = \frac{1}{2}$ and $\sigma(\rho) = \rho(1-\rho)$ so that $\sigma''(\rho) = -2 < 0$, leading to a critical field $E_c = \pi$ such that no symmetry-breaking DPT can appear for any current $q$ when $|E| < E_c$.

In both cases, $\bar{\sigma}'' > 0$ or $\bar{\sigma}'' < 0$, as soon as we cross the critical current $q_c$ and enter infinitesimally into the symmetry-broken regime, the density profile

$$\rho_q^+(x) = \bar{\rho} + |\varphi_1| \sin(\pi x) \tag{73}$$

yields a better minimizer of the current LDF action (55) than just the flat profile. This optimal density field has an associated total mass

$$m_q^+ = \int_0^1 \rho_q^+(x)\,dx = \bar{\rho} + \frac{2|\varphi_1|}{\pi}\,. \tag{74}$$

Since $\Delta G(q)$ just depends on the squared amplitude $\varphi_1^2$, see Eq. (70), it is then clear that

$$\rho_q^-(x) = \bar{\rho} - |\varphi_1| \sin(\pi x) \tag{75}$$

is also a minimizer of the current LDF action, leading to the same current LDF $G(q)$ that $\rho_q^+(x)$, but with a different total mass $m_q^- = \int_0^1 \rho_q^-(x)\,dx = \bar{\rho} - (2|\varphi_1|/\pi) < m_q^+$. This is a clear signature of a spontaneous $\mathbb{Z}_2$-symmetry breaking transition in the current fluctuations of open driven diffusive media.

In this way, while in the homogeneous density regime the optimal density profile $\rho_q(x) = \bar{\rho}$ trivially inherits the PH symmetry of the original action, mapping onto itself under the PH transformation (57), for currents *beyond* a critical threshold (i.e. $|q| \geq |q_c|$ for systems with $\bar{\sigma}'' > 0$ or rather $|q| \leq |q_c|$ and $|E| > E_c$ for systems with $\bar{\sigma}'' < 0$), two different (but equally likely) optimal profiles $\rho_q^{\pm}(x)$ appear mapping onto each other under the PH transformation, i.e. such that

$$\rho_q^{\pm}(x) \to 1 - \rho_q^{\mp}(1-x), \tag{76}$$

and giving rise to a second-order singularity in the current LDF $G(q)$. This spontaneous PH symmetry breaking phenomenon admits a simple interpretation [52, 71]. Think for instance in systems with $\bar{\sigma}'' < 0$: in order to sustain a low-current fluctuation, the system can react by either crowding with particles (resulting in a larger total mass $m_q^+$) hence hindering motion, or rather emptying the lattice (resulting in a smaller total mass $m_q^-$) to minimize particle flow. Both tendencies break the PH-symmetry of the MFT action, eventually triggering a DPT.

## 4.2   A Landau-like theory for the dynamical phase transition

The previous stability analysis has allowed us to locate the critical current $q_c$ beyond which the instability is triggered. To analyze the structure of the current LDF $G(q)$ near the instability, we need to extend this perturbative treatment to include higher-order corrections, inspired by Landau's seminal analysis of symmetry-breaking phase transitions [152]. In particular, nearby $q_c$ the optimal density profile will be perturbatively close to the homogeneous one ($\bar{\rho}$) [52, 71, 109], admitting a series expansion

$$\rho_q(x) \underset{q \approx q_c}{\approx} \bar{\rho} + \sum_{l=1}^{4} \delta \mathrm{m}^l \varphi_l(x) + \mathcal{O}(\delta m^5), \tag{77}$$

in terms of the excess mass $\delta m = m_q - \bar{\rho}$, which plays the role of order parameter for the transition as it measures the *distance* to the homogeneous density phase. We truncate the expansion at order 4 as all higher orders turn out to be irrelevant near the critical point (see below). We will work close to the transition point, in the limit $|\delta m| \ll 1$, so the modulations over the homogeneous profile solution are small, $|\rho_q(x) - \bar{\rho}| \ll 1$. The functions $\varphi_l(x)$ must obey the boundary conditions $\varphi_l(0) = 0 = \varphi_l(1)$ $\forall l \in [1,4]$, and will be determined below. Proceeding as in Eq. (65) above, the current LDF can be expanded as $G(q) \approx G_{\mathrm{flat}}(q) + \Delta G(q)$, with $G_{\mathrm{flat}}(q) = -(q - \sigma(\bar{\rho})E)^2/[2\sigma(\bar{\rho})]$. The function $\Delta G(q)$ is obtained from the least action principle (55) and the expansion (77) for the optimal profile, and can be written as

$$\Delta G(q) = -\min_{\delta m} G(\delta m | q), \tag{78}$$

where $G(\delta m | q)$ is a function of the order parameter $\delta m$ and the current $q$ whose shape near $q_c$ we want to analyze next. In order to do so, we need to determine the functions $\varphi_l(x)$ in (77) by analyzing perturbatively the ODE (56) for the saddle-point profile

$$D^2(\rho)\rho'(x)^2 = q^2 + 2K\sigma(\rho) + E^2\sigma^2(\rho). \tag{79}$$

If $F(\rho)$ is a general function of the local density, symmetric around $\bar{\rho}$ so that all odd derivatives around $\bar{\rho}$ are zero (in particular $F'(\bar{\rho}) = F^{(3)}(\bar{\rho}) = 0$), we can expand

$$F\left[\bar{\rho} + \sum_{l=1}^{4} \delta \mathrm{m}^l \varphi_l(x)\right] \approx \bar{F} + \frac{1}{2}\bar{F}''\varphi_1(x)^2\delta m^2 + \bar{F}''\varphi_1(x)\varphi_2(x)\delta m^3$$
$$+ \frac{1}{2}\left[\bar{F}''\left(2\varphi_1(x)\varphi_3(x) + \varphi_2(x)^2\right) + \frac{1}{12}\bar{F}^{(4)}\varphi_1(x)^4\right]\delta m^4 + \mathcal{O}(\delta m^5), \tag{80}$$

with $\bar{F} = F(\bar{\rho})$ and similar definitions for the different derivatives. Using this expansion for both the diffusivity and mobility transport coefficients, we can write the lhs of Eq. (79) as

$$
\begin{aligned}
D^2(\rho)\rho'(x)^2 \quad \approx \quad & \bar{D}^2 \varphi_1'(x)^2 \delta m^2 + 2\bar{D}^2 \varphi_1'(x)\varphi_2'(x)\delta m^3 \\
& + \ \left[\bar{D}^2\left(\varphi_2'(x)^2 + 2\varphi_1'(x)\varphi_3'(x)\right) + \bar{D}\bar{D}''\varphi_1(x)^2\varphi_1'(x)^2\right]\delta m^4 .
\end{aligned} \tag{81}
$$

The perturbative expansion (77) for the optimal density profile is valid only in a small interval of currents fluctuations around $q_c$. Hence we assume now $q \approx q_c + q_2\delta m^2 + q_4\delta m^4$, with only even powers of the excess mass due to symmetry reasons, see section §4.1. Similarly, the constant $K = K(q^2, E^2)$ admits a series expansion $K = K_0 + K_2\delta m^2 + K_4\delta m^4$. Using these series expansions in Eq. (79), and gathering terms of the same order in $\delta m$ together, we obtain an equation of the form

$$
A_0 + A_2(x)\delta m^2 + A_3(x)\delta m^3 + A_4(x)\delta m^4 = 0, \tag{82}
$$

with $A_i(x)$ certain amplitudes depending on the different expansion coefficients. As this equation must hold for arbitrary (but small) values of $\delta m$, each order must be zero independently. Thus, at order $\delta m^0$, Eq. (79) implies

$$
q_c^2 + \bar{\sigma}(2K_0 + \bar{\sigma}E^2) = 0 \quad \Rightarrow \quad K_0 = -\frac{q_c^2 + \bar{\sigma}^2 E^2}{2\bar{\sigma}} = -\left(\bar{\sigma}E^2 + \frac{\bar{D}^2\pi^2}{\bar{\sigma}''}\right), \tag{83}
$$

where we have used that $q_c^2 = \bar{\sigma}^2 E^2 + 2\pi^2\frac{\bar{D}^2\bar{\sigma}}{\bar{\sigma}''}$, see Eq. (71). On the other hand, at order $\delta m^2$, we have

$$
\bar{D}^2\varphi_1'(x)^2 - \bar{\sigma}''(K_0 + \bar{\sigma}E^2)\varphi_1(x)^2 = 2(q_2 q_c + K_2\bar{\sigma}). \tag{84}
$$

The rhs of this equation is just a constant, while the lhs depends in principle on $x$ via the unknown function $\varphi_1(x)$ and its derivative $\varphi_1'(x)$. Taking into account the value of $K_0$ in Eq. (83), we have that $-\bar{\sigma}''(K_0 + \bar{\sigma}E^2) = \bar{D}^2\pi^2$ so that Eq. (84) reduces to $\varphi_1'(x)^2 + \pi^2\varphi_1(x)^2 = 2(q_2 q_c + K_2\bar{\sigma})/\bar{D}^2$. A solution to this equation is clearly

$$
\varphi_1(x) = \sin(\pi x), \tag{85}
$$

where we have trivially chosen the amplitude to be just one (otherwise this amplitude would renormalize the values of both constants $q_2$ and $K_2$). This first correction was of course expected from the perturbative analysis of previous section. The equation relating the coefficients $q_2$ and $K_2$ is thus $2(q_2 q_c + K_2\bar{\sigma}) = \bar{D}^2\pi^2$, and hence

$$
K_2 = \frac{\bar{D}^2\pi^2 - 2q_2 q_c}{2\bar{\sigma}}. \tag{86}
$$

At order $\delta m^3$, Eq. (79) implies $\bar{D}^2\varphi_1'(x)\varphi_2'(x) = \bar{\sigma}''(K_0 + \bar{\sigma}E)\varphi_1(x)\varphi_2(x)$, which using Eqs. (83) and (85) for $K_0$ and $\varphi_1(x)$, respectively, reduces to

$$
\cos(\pi x)\varphi_2'(x) + \pi\sin(\pi x)\varphi_2(x) = 0. \tag{87}
$$

The only solution for this equation compatible with the boundary conditions $\varphi_2(0) = 0 = \varphi_2(1)$ is just

$$
\varphi_2(x) = 0, \tag{88}
$$

$\forall x \in [0, 1]$. This was again anticipated as the excess optimal density $\rho_q(x) - \bar{\rho}$ is expected to change sign under the transformation $\delta m \to -\delta m$ due to the underlying PH-symmetry, so all even powers $\delta m^{2n}$ in the expansion (77) should vanish.

Next we proceed to analyze order $\delta m^4$. Eq. (79) implies at this order the following equation for the unknown function $\varphi_3(x)$

$$2\bar{D}^2\pi\left[\cos(\pi x)\varphi_3'(x)+\sin(\pi x)\varphi_3(x)\right]=q_2^2+2q_4q_c+2\bar{\sigma}K_4+K_2\bar{\sigma}''\sin^2(\pi x)$$

$$-\bar{D}\bar{D}''\pi^2\sin^2(\pi x)\cos^2(\pi x)+\frac{1}{12}\left[3\bar{\sigma}''^2E^2-\frac{\bar{D}^2\pi^2}{\bar{\sigma}''}\bar{\sigma}^{(4)}\right]\sin^4(\pi x),\qquad(89)$$

where we have already used that $\varphi_1(x)=\sin(\pi x)$ and $\varphi_2(x)=0$, together with $(K_0+\bar{\sigma}E^2)=-\bar{D}^2\pi^2/\bar{\sigma}''$. Postulating a solution of the form $\varphi_3(x)=B\sin(3\pi x)$ and using the following trigonometric identities

$$\begin{aligned}
\sin^2(\pi x) &= \frac{1}{2}[1-\cos(2\pi x)],\\
\sin^4(\pi x) &= \frac{1}{8}[3-4\cos(2\pi x)+\cos(4\pi x)],\\
\sin(\pi x)\sin(3\pi x) &= \frac{1}{2}[\cos(2\pi x)-\cos(4\pi x)],\\
\cos(\pi x)\cos(3\pi x) &= \frac{1}{2}[\cos(2\pi x)+\cos(4\pi x)],\qquad(90)
\end{aligned}$$

we obtain an equation of the form

$$C_0(q_2,q_4,K_4)+C_2(q_2,B)\cos(2\pi x)+C_4(B)\cos(4\pi x)=0,\qquad(91)$$

where $C_0(q_2,q_4,K_4)$, $C_2(q_2,B)$ and $C_4(B)$ are certain constants. In order to fulfill this equation, each of these constants must be zero. Setting $C_0(q_2,q_4,K_4)=0$ yields a relation between constants $K_4$ and $q_4$ in terms of $q_2$, which is not relevant for our discussion here as $K_4$ and $q_4$ do not appear in the first two terms of the LDF $G(\delta m|q)$, see Eq. (78) and below. On the other hand, the constants $C_2(q_2,B)$ and $C_4(B)$ turn out to be

$$\begin{aligned}
C_2(q_2,B) &= -4\left(96\pi^2B\bar{D}^2\bar{\sigma}\bar{\sigma}''-12q_2q_c\bar{\sigma}''^2-\pi^2\bar{D}^2\bar{\sigma}\bar{\sigma}^{(4)}+6\pi^2\bar{D}^2\bar{\sigma}''^2+3E^2\bar{\sigma}\bar{\sigma}''^3\right),\\
C_4(B) &= -\bar{\sigma}\left(192\pi^2B\bar{D}^2\bar{\sigma}''-12\pi^2\bar{D}\bar{D}''\bar{\sigma}''+\pi^2\bar{D}^2\bar{\sigma}^{(4)}-3E^2\bar{\sigma}''^3\right).
\end{aligned}$$

Setting now $C_2(q_2,B)=0=C_4(B)$ yields two linear equations for $q_2$ and $B$ which can be easily solved to obtain

$$q_2 = \frac{4\pi^2\bar{D}\bar{\sigma}\bar{D}''\bar{\sigma}''+\pi^2\bar{D}^2\left(4\bar{\sigma}''^2-\bar{\sigma}\bar{\sigma}^{(4)}\right)+3E^2\bar{\sigma}\bar{\sigma}''^3}{8|q_c|\bar{\sigma}''^2},\qquad(92)$$

$$B = \frac{1}{192}\left(\frac{12\bar{D}''}{\bar{D}}+\frac{3E^2\bar{\sigma}''^2}{\pi^2\bar{D}^2}-\frac{\bar{\sigma}^{(4)}}{\bar{\sigma}''}\right),\qquad(93)$$

so that the last unknown function is

$$\varphi_3(x)=\frac{1}{192}\left(\frac{12\bar{D}''}{\bar{D}}+\frac{3E^2\bar{\sigma}''^2}{\pi^2\bar{D}^2}-\frac{\bar{\sigma}^{(4)}}{\bar{\sigma}''}\right)\sin(3\pi x).\qquad(94)$$

In this way, using the expansion (77) in the variational problem (55) for the current LDF, see also (78), together with the explicit expressions obtained for the functions $\varphi_l(x)$ above to perform the resulting integrals, we obtain

$$G(\delta m|q)=-\frac{\bar{\sigma}''}{8\bar{\sigma}^2}(q^2-q_c^2)\delta m^2+\left[\frac{\pi^2\bar{D}\bar{D}''}{16\bar{\sigma}}+\frac{\pi^2\bar{D}^2\bar{\sigma}''}{16\bar{\sigma}^2}-\frac{\pi^2\bar{D}^2\bar{\sigma}^{(4)}}{64\bar{\sigma}\bar{\sigma}''}+\frac{3E^2\bar{\sigma}''^2}{64\bar{\sigma}}\right]\delta m^4,\quad(95)$$

or equivalently $G(\delta m|q)=-g_2(q)\delta m^2+g_4\delta m^4$ [71]. Hence, for a system with $\bar{\sigma}''>0$ and $g_4>0$, we recover the typical double-well, Landau-like shape for the function $G(\delta m|q)$ for

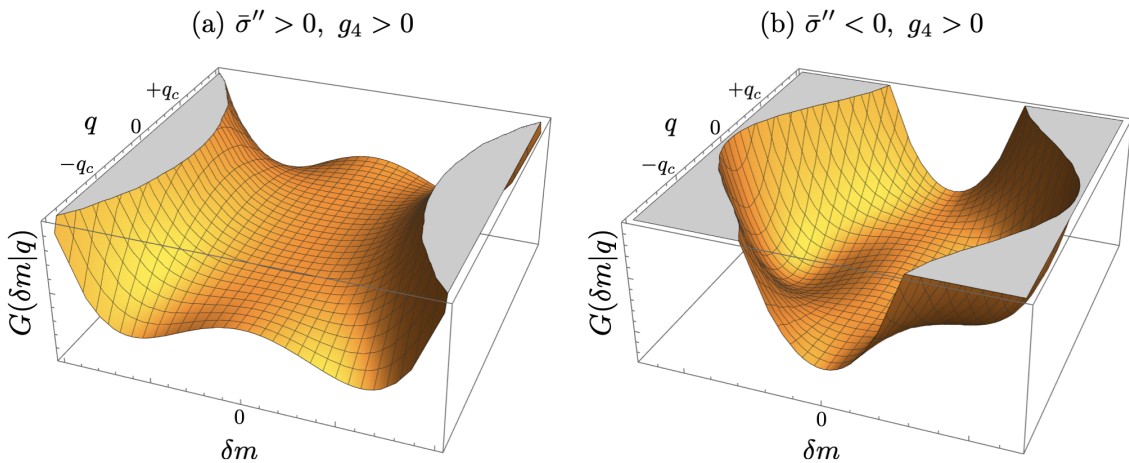

Figure 3: **Landau-like theory for the DPT in open systems.** Typical shape of the function $G(\delta m|q)$ of Eq. (95) as a function of the excess mass $\delta m$ and the current $q$ for the case (a) when $\bar{\sigma}'' > 0$ and $g_4 > 0$ and (b) when $\bar{\sigma}'' < 0$ and $g_4 > 0$. In case (a) two equivalent minima appear in $G(\delta m|q)$ at excess masses $\pm\delta m_q \neq 0$ for $|q| > q_c$, see Eq. (96), while in (b) these two equivalent minima appear for $|q| < q_c$.

$q^2 > q_c^2$ (SSB phase), while in the regime $q^2 \leq q_c^2$ the function $G(\delta m|q)$ exhibits a single minimum as a function of $\delta m$ for $\delta m = 0$ (flat phase). Conversely, for systems with $\bar{\sigma}'' < 0$ and $g_4 > 0$, the typical double-well structure appears for $q^2 < q_c^2$ (SSB phase), as in the WASEP example mentioned before, while the flat phase appears in this case for $q^2 > q_c^2$. We refer the interested reader to [52, 71] for a complete classification of possible dynamical symmetry-breaking scenarios in open systems. Whenever the double-well structure emerges (depending on the sign of $\bar{\sigma}''$ and $g_4$ for the model at hand), the excess mass $\delta m_q$ solving the variational problem (78), obtained by solving the equation $\partial_{\delta m} G(\delta m|q) = [4g_4\delta m^2 - 2g_2(q)]\delta m = 0$, is

$$\delta m_q = \pm\sqrt{\frac{g_2(q)}{2g_4}} = \pm\sqrt{\frac{4\bar{\sigma}''^2|q^2 - q_c^2|}{\left[4\pi^2\bar{D}\bar{D}''\bar{\sigma}\bar{\sigma}'' + \pi^2\bar{D}^2\left(4\bar{\sigma}''^2 - \bar{\sigma}\bar{\sigma}^{(4)}\right) + 3E^2\bar{\sigma}\bar{\sigma}''^3\right]}}, \qquad (96)$$

and therefore the current LDF reads

$$G(q) = G_{\text{flat}}(q) - \min_{\delta m} G(\delta m|q) = \begin{cases} -\dfrac{(q - \bar{\sigma}E)^2}{2\bar{\sigma}} & \text{(flat phase)} \\[4mm] -\dfrac{(q - \bar{\sigma}E)^2}{2\bar{\sigma}} + \dfrac{g_2(q)^2}{4g_4} & \text{(SSB phase)} \end{cases} \qquad (97)$$

Fig. 3.a shows a sketch of the function $G(\delta m|q)$ for the case $\bar{\sigma}'' > 0$ and $g_4 > 0$, while Fig. 3.b displays a typical $G(\delta m|q)$ when $\bar{\sigma}'' < 0$ and $g_4 > 0$.

## 4.3   Order parameter fluctuations across the dynamical phase transition

The perturbative analysis of previous section suggests to investigate the full range of joint fluctuations of the current and the order parameter for the transition, which is the total mass $m$ or equivalently the excess mass $\delta m = m - \bar{\rho}$. Indeed, as explained in the previous section, $m$ exhibits a behavior with $q$ strongly reminiscent of a standard $\mathbb{Z}_2$ phase transition, capturing the dynamical breaking of the PH symmetry at the DPT.

We are hence interested here in the joint statistics of the space&time-averaged current $q$ and mass $m$

$$q = \frac{1}{\tau} \int_0^\tau dt \int_0^1 dx \, j(x,t), \qquad m = \frac{1}{\tau} \int_0^\tau dt \int_0^1 dx \, \rho(x,t). \tag{98}$$

The probability $P_\tau(m,q)$ of observing a given $q$ and $m$ can now be written as a path integral over all possible trajectories $\{\rho, j\}_0^\tau$, weighted by its probability measure $P(\{j, \rho\}_0^\tau)$, and restricted to those trajectories compatible with the values of $q$ and $m$ in Eq. (98), the continuity equation $\partial_t \rho + \partial_x j = 0$ at every point of space and time, and the fixed boundary conditions for the density field. This path integral is similar to that in Eq. (9) but with the additional constraint on the mass. For long times and large system sizes, this sum over trajectories is dominated by the saddle point and scales as $P_\tau(m,q) \sim \exp[+\tau L G(m,q)]$, where $G(m,q)$ is the mass-current large deviation function (LDF), with $G(m,q) = \lim_{\tau \to \infty} \frac{1}{\tau} \max^*_{\{\rho,j\}_0^\tau} \mathcal{I}_\tau(\rho,j)$ and the $*$ representing all the mentioned constraints, such that $G(q) = \max_m G(m,q) = G(m_q,q)$. Under the additivity principle, assuming a (mostly) time-independent optimal trajectory [27, 40, 44], we can write a variational expression for the mass-current LDF $G(m,q)$ similar to Eq. (55) but subject to the additional constraint $m = \int_0^1 \rho_{m,q}(x)dx$, as well as to fixed boundary conditions. Indeed, writing $P_\tau(m,q)$ as a functional integral over density fields (assuming additivity)

$$P_\tau(m,q) = \int \mathcal{D}\rho \exp\left(-\tau L \int_0^1 dx \frac{[q + D(\rho)\partial_x \rho - \sigma(\rho)E]^2}{2\sigma(\rho)}\right) \delta\left(m - \int_0^1 \rho(x)\,dx\right) \tag{99}$$

and using now an integral representation of the Dirac $\delta$-function similar to Eq. (31)

$$\delta\left(m - \int_0^1 \rho(x)\,dx\right) = \int d\lambda \, e^{+\tau L \lambda \left[m - \int_0^1 \rho(x)\,dx\right]}, \tag{100}$$

we have that

$$P_\tau(m,q) = \int \mathcal{D}\rho \, d\lambda \exp\left[-\tau L \left(\int_0^1 dx \frac{[q + D(\rho)\partial_x \rho - \sigma(\rho)E]^2}{2\sigma(\rho)} - \lambda(m - \rho)\right)\right] \asymp e^{+\tau L G(m,q)}, \tag{101}$$

where we have now

$$G(m,q) = -\min_{\rho(x),\lambda} \int_0^1 dx \left[\frac{(q + D(\rho)\partial_x \rho - \sigma(\rho)E)^2}{2\sigma(\rho)} - \lambda(m - \rho)\right], \tag{102}$$

with $\lambda$ playing the role of a Lagrange multiplier, fixed *a posteriori* to enforce the mass constraint. This is a variational problem similar to the standard additivity principle problem (55) but with an additional linear term in the density field. Proceeding as in section §2.1 we obtain the ordinary differential equation (ODE) for the optimal density profile $\rho_{m,q}(x)$

$$D^2(\rho)\left(\frac{d\rho(x)}{dx}\right)^2 = q^2 + 2\sigma(\rho)[K + \lambda\rho] + E^2\sigma^2(\rho), \tag{103}$$

where we have integrated once the equation solving the variational problem (102), resulting in an integration constant $K = K(q^2, E^2)$. Note that the only difference with the ODE for the optimal profile $\rho_q(x)$ for current fluctuations, Eq. (56), is the term $2\sigma(\rho)\lambda\rho$. The value of the Lagrange multiplier $\lambda(m,q)$ can be now fixed by imposing that the total mass associated to the solution $\rho_{\lambda,q}(x)$ of the above differential equation is just $m$, i.e. $m = \int_0^1 \rho_{\lambda,q}(x)dx$. Using these results, it is now possible to study analytically the dynamical phase transition described

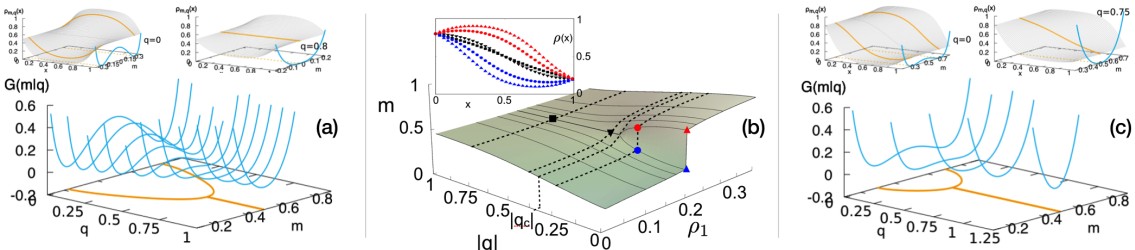

Figure 4: **Joint mass-current fluctuations for the** $1d$ **open WASEP.** (a) Total mass LDF conditioned on a given current, $G(m|q) = G(q) - G(m,q)$, as a function of the mass $m$ for different currents $q$, for equal and PH-symmetric boundary densities, $\rho_0 = 0.5 = \rho_1$, and external field $E = 4 > E_c$. The line projected in the $m-q$ plane corresponds to the local minima of the LDF $G(m|q)$, which define the mass $m_q$ associated to a current fluctuation $q$. In the symmetry-broken regime $|q| \leq q_c$ this defines the low- and high-mass branches $m_q^{\pm}$. The top panels show the optimal density profiles $\rho_{m,q}(x)$ obtained for $q = 0$ (left) and $q = 0.8$ (right). The thick lines are the optimal profiles associated to the local minima $m_q^{\pm}$ of $G(m|q)$, which is also shown for completeness. Panel (b) displays the mass $m_q$ of the optimal trajectory responsible for a current fluctuation $q$ for different boundary drivings, with $\rho_0 = 0.8$, $\rho_1 \in [0, 0.4]$, and external field $E = 4$. The inset shows the measured optimal density profiles for the case $\rho_1 = 0.2$ and the $q$'s signaled in the main plot with color points. Panel (c) shows an information equivalent to that of panel (a) but for a PH-symmetric boundary driving $\rho_0 = 0.8$ and $\rho_1 = 0.2$. Image reproduced from [109] with permission from the American Physical Society.

in sections §4.1 and §4.2 for arbitrary boundary gradients (symmetric or asymmetric), well beyond the perturbative nonequilibrium linear regime.

To gain some further insight, we now discuss briefly the results obtained by solving the joint mass-current variational problem (102)-(103) for the $1d$ open WASEP, the model of particle diffusion under exclusion interactions commented previously for which the diffusivity and mobility are simply $D(\rho) = 1/2$ and $\sigma(\rho) = \rho(1 - \rho)$, respectively. Note that this model exhibits PH symmetry for appropriate boundary conditions. Moreover, since $\sigma''(\rho) = -2 < 0$ for WASEP, we expect a dynamical phase transition in this model for currents $|q| \leq q_c$ and external field $E > E_c$, see sections §4.1 and §4.2 and Fig. 3.b. We refer the interested reader to Ref. [109] for details on the analytical and/or numerical solution of this variational problem for this particular model.

We studied the conditional mass-current LDF $G(m|q)$, defined as $G(m|q) \equiv G(q) - G(m,q)$, which captures the large deviations scaling of the conditional probability for observing a given total mass $m$ in the system given a fixed current $q$. For equal and PH-symmetric boundary densities, $\rho_0 = 0.5 = \rho_1$, $G(m|q)$ exhibits a peculiar change of behavior at a critical current $|q_c|$, see main panel in Fig. 4.a, as anticipated by the perturbative Landau-like expansion of section §4.2. In particular, $G(m|q)$ displays a single minimum at $m_q = 1/2$ for $|q| > |q_c|$, with an associated PH-symmetric optimal profile (see example in the top-right panel in Fig. 4.a). However, for currents $|q| < |q_c|$ two equivalent minima $m_q^{\pm}$ appear in $G(m|q)$, each one associated with a PH-symmetry-broken optimal profile $\rho_q^{\pm}(x)$, see top-left panel in Fig. 4.a, such that $\rho_q^{\pm}(x) \to 1 - \rho_q^{\mp}(1 - x)$ under the PH transformation. The emergence of this non-convex regime is the fingerprint of a second-order DPT to a dynamical phase with broken PH-symmetry. A similar symmetry-breaking scenario is observed in the presence of boundary gradients (with $\rho_0 \neq \rho_1$), provided that condition (59) holds, i.e. $\rho_0 = 1 - \rho_1$, see Fig. 4.c for $\rho_0 = 0.8$ and $\rho_1 = 0.2$. In particular, $G(m|q)$ also crosses over from unimodal for $|q| > |q_c|$ to

bimodal for $|q| < |q_c|$, with two equivalent minima in this regime characterized by two different PH-symmetry-broken optimal profiles $\rho_q^\pm(x)$, see top-left panel in Fig. 4.c. On the other hand, for *PH-asymmetric* boundaries, $\rho_0 \neq 1 - \rho_1$, the action (54) governing the system fluctuations is no longer PH-symmetric: the asymmetry favors one of the mass branches and $G(m|q)$ exhibits a single *global* minimum $\forall q$ and an unique optimal density profile. Still, $G(m|q)$ may become non-convex for low enough currents [109], and for weak gradient asymmetry the LDF $G(m|q)$ may develop metastable-like local minima, signaling the emergence of a sort of metastable dynamical coexistence. Indeed, the typical mass $m_q$ during a current fluctuation, where the (local or global) minima of $G(m|q)$ appear for a fixed $q$ and which can be simply evaluated by demanding $\partial_m G(m|q) = 0$, exhibits a behavior strongly reminiscent of a standard $\mathbb{Z}_2$ phase transition, see Fig. 4.b, defining the phase diagram for this DPT.

## 4.4   Instantons, Maxwell-like construction and violation of additivity principle

A natural question is whether time-dependent optimal trajectories exist which improve the additivity principle minimizers of the MFT action (54). The emergence of a non-convex regime in $G(m|q)$ –or equivalently in $G(m, q)$– for $|q| < |q_c|$ described in the previous sections suggests a Maxwell-like solution in this region [71, 153]. In this section we conjecture a time-dependent, instanton-like solution for the optimal density and current fields responsible of a joint fluctuation of the empirical current and mass, showing that it improves the additivity principle prediction in the regime where $G(m, q)$ becomes non-convex. This corresponds to a *dynamical coexistence* of the different symmetry-broken phases for $|q| < |q_c|$.

   We start from the most general variational expression for the joint mass-current LDF $G(m, q)$ in terms of the trajectory-level MFT action (54), i.e. $G(m, q) = \lim_{\tau \to \infty} \frac{1}{\tau} \max^*_{\{\rho, j\}_0^\tau} \mathcal{I}_\tau[\rho, j]$. The asterisk $^*$ in this expression represents as usual all the necessary constraints, including the continuity equation coupling the fields $\rho(x, t)$ and $j(x, t)$ at every point of space and time, $\partial_t \rho + \partial_x j = 0$, the constraints (98) on the empirical current $q$ and mass $m$, and the boundary conditions for the density field, $\rho(0, t) = \rho_0$ and $\rho(1, t) = \rho_1 \ \forall t$. For the time being, let us denote as $G_{\text{AP}}(m, q)$ the mass-current LDF obtained from the additivity principle (AP) [27], see Eq. (102). Similarly, we denote as $\rho_{m,q}^{\text{AP}}(x)$ the optimal density profile responsible of a joint mass and current fluctuation under the additivity hypothesis. To search for violations of the additivity principle, we focus now our attention in systems driven by a PH-symmetric density gradient ($\rho_0 = 1 - \rho_1$), including the equal boundary density case, in the regime of current fluctuations beyond the critical threshold where the joint LDF $G(m, q)$ exhibits a non-convex region (i.e. $|q| \geq |q_c|$ for systems with $\bar{\sigma}'' > 0$ or $|q| \leq |q_c|$ and $|E| > E_c$ for systems with $\bar{\sigma}'' < 0$). In this regime we conjecture a solution for the optimal *trajectory* responsible of a given mass-current fluctuation which is time-dependent for masses where $G(m, q)$ is non-convex. In particular, our ansatz in this current regime is

$$\rho_{m,q}(x, t) = \begin{cases} \rho_{m,q}^{\text{AP}}(x) & \text{if } m < m_q^- \text{ or } m > m_q^+ \\ \\ \rho_{m_q^+,q}^{\text{AP}}(x) \, \phi(t - t_{m,q}) + \rho_{m_q^-,q}^{\text{AP}}(x) \big[1 - \phi(t - t_{m,q})\big] & \text{if } m_q^- \leq m \leq m_q^+ \end{cases}$$

$$(104)$$

where $m_q^\pm$ are the local minima of $G(m, q)$ as a function of $m$ in this regime, see Figs. 3 and 4, and correspond to the masses of the optimal density profiles $\rho_q^\pm(x) = \rho_{m_q^\pm,q}^{\text{AP}}(x)$ associated to a current fluctuation in the PH-symmetry broken regime along the high-mass (+) and low-mass (−) branches. The time-dependent function $\phi(t)$ is a sufficiently smooth localized crossover function such that $\phi(t) = 0 \ \forall t < -\frac{\delta t}{2}$ and $\phi(t) = 1 \ \forall t > \frac{\delta t}{2}$, with $\delta t$ a fixed (small) crossover timescale, such that $\delta t / \tau \to 0$ in the long-time limit. The crossover time $t_{m,q} \leq \tau$ in Eq (104) can be determined now by imposing the constraint on the empirical mass $m$, see Eq. (98). In

particular

$$
\begin{aligned}
m &= \frac{1}{\tau} \int_0^\tau dt \int_0^1 dx \, \rho_{m,q}(x,t) \\
&= \left( \frac{t_{m,q} - \frac{\delta t}{2}}{\tau} \right) m_q^- + \left( \frac{\tau - (t_{m,q} + \frac{\delta t}{2})}{\tau} \right) m_q^+ + \frac{1}{\tau} \int_{t_{m,q} - \frac{\delta t}{2}}^{t_{m,q} + \frac{\delta t}{2}} dt \int_0^1 dx \rho_{m,q}(x,t) \\
&= \frac{t_{m,q}}{\tau} m_q^- + \left( 1 - \frac{t_{m,q}}{\tau} \right) m_q^+ + \frac{1}{\tau} \left[ -\delta t + \int_{t_{m,q} - \frac{\delta t}{2}}^{t_{m,q} + \frac{\delta t}{2}} dt \int_0^1 dx \rho_{m,q}(x,t) \right].
\end{aligned} \tag{105}
$$

The third term in the last line of the previous equation is of order $\sim \mathcal{O}(\delta t / \tau)$, so in the long-time limit ($\tau \to \infty$) and for a fixed crosscover time $\delta t$ this term tends to zero, and hence we find $t_{m,q} = p \, \tau$ with the definition

$$
p = \frac{m_q^+ - m}{m_q^+ - m_q^-}, \qquad (0 \le p \le 1). \tag{106}
$$

As mentioned above, the time-dependent optimal density field $\rho_{m,q}(x,t)$ must obey at all points of space and time a continuity equation $\partial_t \rho_{m,q}(x,t) + \partial_x j_{m,q}(x,t) = 0$. To obtain the optimal current field $j_{m,q}(x,t)$ for $m_q^- \le m \le m_q^+$, we first note that in this case

$$
\partial_t \rho_{m,q}(x,t) = \begin{cases} 0 & \text{if } t \notin [t_{m,q} - \frac{\delta t}{2}, t_{m,q} + \frac{\delta t}{2}] \\[2mm] \left[ \rho_{m_q^+,q}^{\mathrm{AP}}(x) - \rho_{m_q^-,q}^{\mathrm{AP}}(x) \right] \phi'(t - t_{m,q}) & \text{if } t \in [t_{m,q} - \frac{\delta t}{2}, t_{m,q} + \frac{\delta t}{2}] \end{cases} \tag{107}
$$

where $\phi'(t)$ is the time derivative of the crossover function. Therefore the continuity constraint in the non-convex mass regime $m_q^- \le m \le m_q^+$ leads to the following optimal current field

$$
j_{m,q}(x,t) = \begin{cases} q & \text{if } t \notin [t_{m,q} - \frac{\delta t}{2}, t_{m,q} + \frac{\delta t}{2}] \\[2mm] \chi(x) \, \phi'(t - t_{m,q}) & \text{if } t \in [t_{m,q} - \frac{\delta t}{2}, t_{m,q} + \frac{\delta t}{2}] \end{cases} \tag{108}
$$

where we have already taken into account the constraint on the empirical current $q$, see Eq. (98). The function $\chi(x)$ is such that $\chi'(x) = \rho_{m_q^+,q}^{\mathrm{AP}}(x) - \rho_{m_q^-,q}^{\mathrm{AP}}(x)$, and we note that the transient regime where $j_{m,q}(x,t)$ is different from $q$ does not contribute to the final value of the empirical current, see Eq. (98), as this transient is negligible against the long-time limit for $\tau$.

Using this ansatz for the optimal trajectory $\{\rho_{m,q}(x,t), j_{m,q}(x,t)\}_0^\tau$ responsible of a mass and current fluctuation, we obtain for the associated joint LDF

$$
\begin{aligned}
G(m,q) &= - \lim_{\tau \to \infty} \frac{1}{\tau} \int_0^\tau dt \int_0^1 dx \frac{[j_{m,q}(x,t) + D(\rho_{m,q}) \partial_x \rho_{m,q}(x,t) - E\sigma(\rho_{m,q})]^2}{2\sigma(\rho_{m,q})} \\
&= \lim_{\tau \to \infty} \left[ \left( \frac{t_{m,q} - \frac{\delta t}{2}}{\tau} \right) G_{\mathrm{AP}}(m_q^-, q) + \left( \frac{\tau - (t_{m,q} + \frac{\delta t}{2})}{\tau} \right) G_{\mathrm{AP}}(m_q^+, q) + \frac{1}{\tau} \mathcal{I}_{\delta t} \right],
\end{aligned}
$$

with the definition

$$
\mathcal{I}_{\delta t} \equiv - \int_{t_{m,q} - \frac{\delta t}{2}}^{t_{m,q} + \frac{\delta t}{2}} dt \int_0^1 dx \frac{[j_{m,q}(x,t) + D(\rho_{m,q}) \partial_x \rho_{m,q}(x,t) - E\sigma(\rho_{m,q})]^2}{2\sigma(\rho_{m,q})}.
$$

Noting that $\mathcal{I}_{\delta t} \sim \mathcal{O}(\delta t)$ and using the same arguments as above, we find in the long-time limit $\tau \to \infty$ that

$$G(m, q) = p\, G_{\text{AP}}(m_q^-, q) + (1-p)G_{\text{AP}}(m_q^+, q), \qquad (109)$$

which corresponds to the Maxwell construction (convex envelope) obtained from $G_{\text{AP}}(m, q)$ in the mass regime $m_q^- \le m \le m_q^+$ where this joint LDF is non-convex. Note that an equivalent argument can be formulated for the conditional mass-current LDF $G(m|q) = G(q) - G(m, q)$. This instanton solution corresponds to the dynamical coexistence of the different PH symmetry-broken phases which appear for currents *beyond* the critical threshold ($|q| > q_c$ or $|q| < q_c$ depending on the sign of $\bar{\sigma}''$). The previous time-dependent solution can be also generalized to PH-asymmetric boundaries in regimes where $G(m, q)$ is non-convex. Finally, we would like to mention that some subtleties of the instanton solution appear for $|q| \approx q_c$ related to the order of the $L \to \infty$ and $\tau \to \infty$ limits, see Ref. [71] for a discussion of this issue.

# 5 Time-translation symmetry breaking and traveling waves in periodic system

The previous sections have demonstrated the validity of the additivity principle [27] (and its weak variant [50, 53]) as a powerful conjecture to understand the current statistics in different nonequilibrium diffusive systems, both in $1d$ and in higher dimensions. The physical picture behind this hypothesis corresponds to a system that, in order to sustain a rare current fluctuation, and maybe after a short time transient (microscopic in the diffusive timescale), settles into a time-independent state with a structured density field (possibly different from the stationary one) and a current field constant and equal to $q$ in $1d$ or structured in $d > 1$.

Interestingly, this additivity conjecture has helped us in understanding intriguing dynamic phase transitions in some open diffusive media, involving a discrete $\mathbb{Z}_2$ symmetry breaking phenomenon [52, 71, 109]. However, we have also witnessed how the additivity principle can be violated in cases where the (joint mass-current) LDF becomes non-convex, see §4.4. In this section we explore a different scenario for additivity violations. In particular, we will show how in some cases the additivity picture eventually breaks down for large current fluctuations via a novel type of dynamical phase transition at the fluctuating level, where time-dependent optimal trajectories in the form of *traveling waves* emerge as dominant solution to the MFT variational problem [2, 14, 29, 43, 44, 56, 108]. Remarkably, this DPT involves the spontaneous breaking of a *continuous* symmetry, the time-translation invariance. This will allow us to connect later on in these lecture notes the emerging traveling-wave phases at the fluctuating level with the concept of *time crystals* [97–99, 101, 103, 154–156], a compelling new phase of nonequilibrium matter.

In order to settle ideas and simplify the discussion, we focus on $1d$ diffusive systems in the unit line, described at the mesoscopic level by a density $\rho(x, t)$ and current $j(x, t)$ fields obeying a Fick-type fluctuating hydrodynamic equation (3)-(4), and subject to an external driving field $E$ and, most crucially, to *periodic boundary conditions*, so that $\rho(0, t) = \rho(1, t)$ and $j(0, t) = j(1, t)\ \forall t$. Periodic boundary conditions, together with the local conservation law associated to diffusive dynamics, imply that the total mass in the system is a globally conserved magnitude,

$$\int_0^1 \rho(x, t)\, dx = \bar{\rho} \qquad \forall t. \qquad (110)$$

Due to the system periodicity, the system's stationary density profile is just homogeneous and uniformly equal to $\bar{\rho}$, so $\langle \rho(x) \rangle = \bar{\rho}$, and the average current is hence $\langle q \rangle = \bar{\sigma} E$. According to MFT, and as done previously in these lecture notes, the LDF for the space&time-averaged

current $q = \tau^{-1} \int_0^\tau dt \int_0^1 dx \, j(x, t)$ is given by a variational problem over trajectories for the MFT action functional, i.e. $G(q) = \lim_{\tau \to \infty} \frac{1}{\tau} \max^*_{\{\rho, j\}_0^\tau} \mathcal{I}_\tau[\rho, j]$, with $*$ representing the usual constraints on $q$, the continuity equation $\partial_t \rho + \partial_x j = 0$, and the boundary conditions, periodic in this case.

Finding the optimal trajectory solution of the above spatiotemporal variational problem is in general a complex task whose solution remains challenging in most cases. This problem becomes much simpler however in different limiting cases, as e.g. for small current fluctuations around the average, $q \approx \langle q \rangle$, which we expect to arise from the random superposition of weakly correlated local fluctuations of the microscopic dynamics. In this case it is plausible to assume the optimal density field to be just the homogeneous, steady-state one, so $\rho_q(x, t) = \bar{\rho}$ in this regime and hence $j_q(x, t) = q$, resulting in a simple quadratic form for the current LDF

$$G(q) = G_{\text{flat}}(q) = -\frac{(q - \bar{\sigma} E)^2}{2\bar{\sigma}}, \qquad \text{for } q \approx \langle q \rangle. \tag{111}$$

This results in Gaussian statistics for small (or *typical*) current fluctuations, as expected from the Central Limit Theorem [44]. This argument may break down for moderately strong current deviations where correlations may play a significant role, as we will show next.

## 5.1 Local stability of the flat profile against spatiotemporal perturbations

To analyze how far we can extend the Gaussian current statistics ansatz discussed in the previos paragraph, we will now study the local stability of the homogeneous density profile against small but otherwise arbitrary *spatiotemporal* perturbations, in the sense of checking whether the perturbed density and current profiles yield an improved minimizer of the MFT action for the current LDF $G(q)$. We follow in this section the calculation first performed in [29]. If $\{\rho_q(x, t), j_q(x, t)\}_0^\tau$ is the (unknown) optimal trajectory responsible of a given current fluctuation $q$, the current LDF can be written in general as

$$G(q) = -\int_0^\tau dt \int_0^1 dx \frac{\left(j_q + D(\rho_q)\partial_x \rho_q - \sigma(\rho_q)E\right)^2}{2\sigma(\rho_q)}. \tag{112}$$

To study the stability of the homogeneous (or flat) solution against small perturbations, we now assume an optimal current field of the form

$$j_q(x, t) = q + \varepsilon \left[ j_1(x)\cos(\nu t) + j_2(x)\sin(\nu t) \right], \tag{113}$$

where $\varepsilon \ll 1$ is a small parameter, $j_1(x)$ and $j_2(x)$ are some periodic functions of space with unit period, and $\nu$ is some temporal frequency. Due to the continuity constraint, $\partial_t \rho_q(x, t) + \partial_x j_q(x, t) = 0$, the associated optimal density profile is

$$\rho_q(x, t) = \bar{\rho} + \frac{\varepsilon}{\nu} \left[ -j_1'(x)\sin(\nu t) + j_2'(x)\cos(\nu t) \right]. \tag{114}$$

The question is thus whether $G(q)$ *increases* with respect to $G_{\text{flat}}(q)$ upon adding this small spatiotemporal perturbation to the optimal flat trajectory. Expanding the current LDF $G(q)$ in Eq. (112) to second order in $\varepsilon$ (and noting that now the odd derivatives of the transport coefficients $D(\rho)$ and $\sigma(\rho)$ do not vanish in general at the average density $\bar{\rho}$), we find

$$G(q) \underset{\varepsilon \ll 1}{\approx} G_{\text{flat}}(q) + \varepsilon^2 \int_0^1 dx \left[ -\frac{1}{4\bar{\sigma}}(j_1^2 + j_2^2) - \frac{\bar{D}^2}{4\nu^2 \bar{\sigma}}(j_1''^2 + j_2''^2) + \frac{q\bar{\sigma}'}{2\nu\bar{\sigma}^2}(j_1 j_2' - j_2 j_1') \right.$$
$$\left. + (j_1'^2 + j_2'^2)\left( \frac{q^2 \bar{\sigma}''}{8\nu^2 \bar{\sigma}^2} - \frac{E^2 \bar{\sigma}''}{8\nu^2} - \frac{q^2 \bar{\sigma}'^2}{4\nu^2 \bar{\sigma}^3} \right) \right]. \tag{115}$$

Next we would perform a Fourier series expansion of the currents fields $j_{1,2}(x)$ taking advantage of the system's periodic boundary conditions. Interestingly, since the dependence on the current fields $j_{1,2}(x)$ in Eq. (115) is quadratic, the different Fourier modes are decoupled, thus simplifying the problem. As in section §4.1, it turns out that the first spatial Fourier mode is the first to become unstable, triggering the dynamical phase transition. Hence, to simplify the calculation the calculation of the critical current, we assume now just a simple first-mode Fourier expression for $j_{1,2}(x)$

$$j_1(x) = a \cos(2\pi x) + b \sin(2\pi x), \qquad j_2(x) = c \cos(2\pi x) + d \sin(2\pi x), \qquad (116)$$

resulting in the following perturbed expresison for the current LDF after spatial integration

$$G(q) \underset{\varepsilon \ll 1}{\approx} G_{\text{flat}}(q) + \varepsilon^2 \left[ (a^2 + b^2 + c^2 + d^2) A_q(\nu) + (ad - bc) B_q(\nu) \right], \qquad (117)$$

where we have defined the amplitudes

$$A_q(\nu) \equiv \left( \frac{1}{8\bar{\sigma}} + \frac{2\pi^4 \bar{D}^2}{\nu^2 \bar{\sigma}} + \frac{\pi^2 \bar{\sigma}'^2 q^2}{2\nu^2 \bar{\sigma}^3} + \frac{E^2 \pi^2 \bar{\sigma}''}{4\nu^2} - \frac{\pi^2 \bar{\sigma}'' q^2}{4\nu^2 \bar{\sigma}^2} \right),$$

$$B_q(\nu) \equiv \frac{q \bar{\sigma}' \pi}{\nu \bar{\sigma}^2},$$

The perturbation in Eq. (117) is a quadratic form of $(a, b, c, d)$. Therefore the flat density profile will be *stable* against the conjectured perturbation (113)-(114), i.e. we will have that $G(q) < G_{\text{flat}}(q)$, whenever $A_q(\nu) > |B_q(\nu)|/2$, or equivalently

$$\left( \frac{1}{8\bar{\sigma}} + \frac{2\pi^4 \bar{D}^2}{\nu^2 \bar{\sigma}} + \frac{\pi^2 \bar{\sigma}'^2 q^2}{2\nu^2 \bar{\sigma}^3} + \frac{E^2 \pi^2 \bar{\sigma}''}{4\nu^2} - \frac{\pi^2 \bar{\sigma}'' q^2}{4\nu^2 \bar{\sigma}^2} \right) > \left| \frac{q \bar{\sigma}' \pi}{2\nu \bar{\sigma}^2} \right|.$$

Multiplying this expression by $\nu^2 > 0$ we obtain a second-order polynomial condition $P_q(\nu) = \frac{1}{8\bar{\sigma}} \nu^2 - \left| \frac{q \bar{\sigma}' \pi}{2\bar{\sigma}^2} \right| \nu + C_q > 0$. This polynomial has a minimum at a frequency $\nu_q$ such that $P'_q(\nu_q) = 0$, leading to

$$\nu_q = \frac{2\pi q \bar{\sigma}'}{\bar{\sigma}}. \qquad (118)$$

The flat density profile can become unstable if at least for one frequency the inequality $P(\nu) > 0$ is not satisfied, and this will happen first for the frequency of the minimum $\nu_q$. Hence the condition to deduce the critical current $q_c$ is $P_{q_c}(\nu_{q_c}) = 0$, or equivalently

$$8\pi^2 \bar{D}^2 \bar{\sigma} + (E^2 \bar{\sigma}^2 - q_c^2) \bar{\sigma}'' = 0, \qquad (119)$$

leading to a critical current

$$q_c = \pm \sqrt{\frac{8\pi^2 \bar{D}^2 \bar{\sigma}}{\bar{\sigma}''} + \bar{\sigma}^2 E^2}. \qquad (120)$$

The current regime where the flat density profile is unstable is defined by the condition $P_q(\nu_q) < 0$, or equivalently $8\pi^2 \bar{D}^2 \bar{\sigma} + (E^2 \bar{\sigma}^2 - q^2) \bar{\sigma}'' < 0$. Hence we conclude that for systems with $\bar{\sigma}'' > 0$ the flat profile will be unstable for currents $|q| \geq |q_c|$, i.e. for large enough currents. This is the case e.g. for the KMP model of heat conduction [14, 44, 119], where $\sigma(\rho) = \rho^2$. On the other hand, for systems with $\bar{\sigma}'' < 0$ the flat profile will be unstable for currents $|q| \leq |q_c|$, i.e. for low currents in a (symmetric) neighborhood of $q = 0$. An example of this type of behavior is the WASEP model of particle diffusion under exclusion interactions described in previous sections [43, 56, 120–122], where $\sigma(\rho) = \rho(1 - \rho)$. Moreover, when $\bar{\sigma}'' < 0$, the

instability can only exist if the strength of the driving field $|E|$ is large enough to guarantee a positive discriminant in Eq. (120) for $q_c$, namely, $|E| > E_c$ with

$$E_c = \text{Re}\left[ \sqrt{\frac{-8\pi^2 \bar{D}^2}{\bar{\sigma}\bar{\sigma}''}} \right]. \tag{121}$$

Imposing now that the perturbation over the flat current LDF in Eq. (117) has to be non-negative, i.e. $G(q) - G_{\text{flat}}(q) \geq 0$, we arrive at the relations $a = \pm d$ and $b = \mp c$ between the coefficients $(a, b, c, d)$ in Eq. (113). Using these relations we obtain that right beyond the instability, whenever the flat density and current profiles become unstable, and according to Eq. (113), the optimal current field takes the general form

$$j_q(x, t) = q + D \cos\left[ 2\pi \left( x - x_0 - \frac{q\bar{\sigma}'}{\bar{\sigma}}t \right) \right], \tag{122}$$

which is nothing but a *traveling wave* with an arbitrary amplitude $D$ and phase shift $x_0$ to be determined [29]. As mentioned at the beginning, a similar calculation can be performed to analyze the stability of the flat profile against higher-order Fourier modes (of the form $2\pi n x$, with $n \geq 1$). This calculation would allow us to conclude that the $n = 1$ mode is the first mode to become unstable, very much in the spirit of the local stability calculation of section §4.1.

The previous instability can be interpreted as a dynamical phase transition at the fluctuation level, and involves the spontaneous breaking of continuous time-translation symmetry. In fact, the formation of a traveling wave corresponds to the emergence of a macroscopic condensate which favors/hinders the transport of local density (depending on the sign of $\bar{\sigma}''$) to facilitate a current fluctuation beyond $q_c$.

## 5.2 Current fluctuations in the traveling-wave phase

Interestingly, the previous results show that the dominant perturbation immediately after the instability kicks in takes the form of a traveling wave. This corresponds to a collective re-arrangement of the density and current fields, which breaks the system continuous (spatio-)temporal translation symmetry by localizing the density field in a traveling condensate to facilitate current fluctuations beyond the instability threshold.

This perturbative solution can be now extended to all currents beyond the critical line. In particular, we assume now that the optimal density and current fields well beyond the instability conserve a traveling-wave structure moving at some constant velocity $v$, i.e. we conjecture

$$\rho_q(x, t) = \omega_q(x - vt), \tag{123}$$

with $\omega_q(z)$ some unknown function to be determined below from the variational problem for the current LDF. This ansatz implies, via the continuity equation (and the different constraints), an optimal current field

$$j_q(x, t) = q - v\bar{\rho} + v\omega_q(x - vt). \tag{124}$$

The particular shape of the traveling-wave function $\omega_q(z)$ for $q$'s beyond the critical current can be now obtained as the solution to the variational problem for the current LDF (112), or more generally Eq. (11), i.e.

$$G(q) = -\min_{\omega(x), v} \int_0^1 \frac{dx}{2\sigma[\omega(x)]} \left( q - v\bar{\rho} + v\omega(x) + D[\omega(x)]\omega'(x) - \sigma[\omega(x)]E \right)^2, \tag{125}$$

where the minimum is now taken over the traveling wave profile $\omega(x)$ and its velocity $v$. Note that we have dropped the time dependence in the previous expression due to the periodic

boundary conditions, using that $\int_0^\tau dt \int_0^1 F(x-vt)dx = \tau \int_0^1 F(x)dx$ for any periodic function $F(x)$ in the unit line; this can be shown using the Fourier series expansion for $F(x)$. Expanding now the square in Eq. (125), we notice that the terms linear in $\omega'$ give a null contribution when integrated due again to the system periodicity. Taking also into account the constraint $\int_0^1 \omega_q(x)dx = \bar\rho$, we obtain

$$G(q) = qE - \min_{\omega(x),v} \int_0^1 dx \left[ X_v(\omega) + Y(\omega)\omega'(x)^2 \right], \qquad (126)$$

where we have partially borrowed the notation of Ref. [29], see also [43], to define

$$X_v(\omega) \equiv \frac{[q - v(\bar\rho - \omega)]^2}{2\sigma(\omega)} + \frac{E^2\sigma(\omega)}{2}, \qquad Y(\omega) \equiv \frac{D(\omega)^2}{2\sigma(\omega)}. \qquad (127)$$

In order to obtain the differential equation for the optimal traveling wave profile, we perform now the functional derivative of the functional $H_v(\omega) \equiv \int_0^1 dx \left[ X_v(\omega) + Y(\omega)\omega'(x)^2 \right]$ in Eq. (126). Perturbing $\omega(x) \to \omega(x) + \delta\omega(x)$, we have

$$
\begin{aligned}
H_v(\omega + \delta\omega) &= H_v(\omega) + \int_0^1 dx\,\delta\omega(x)\left[ X_v'(\omega) + Y'(\omega)\omega'(x)^2 \right] + \int_0^1 dx\,\delta\omega'(x)2Y(\omega)\omega'(x) \\
&= H_v(\omega) + \int_0^1 dx\,\delta\omega(x)\left[ X_v'(\omega) - Y'(\omega)\omega'(x)^2 - 2Y(\omega)\omega''(x) \right], \qquad (128)
\end{aligned}
$$

where we have integrated by parts the integral with $\delta\omega'(x)$ in the first equality to obtain the second line (using that the boundary term is zero due to periodicity), and where we have defined $X_v'(\omega) = \delta X_v(\omega)/\delta\omega$ and $Y'(\omega) = \delta Y(\omega)/\delta\omega$. Imposing that the variation in Eq. (128) has to be zero for any (periodic) perturbation $\delta\omega(x)$ leads to the differential equation

$$X_v'(\omega) - Y'(\omega)\omega'(x)^2 - 2Y(\omega)\omega''(x) = C_2, \qquad (129)$$

where $C_2$ is an arbitrary constant (not necessarily 0), since $\int_0^1 dx\,C_2\delta\omega(x) = 0$ for any periodic $\delta\omega(x)$ with zero average. Finally, multiplying the previous equation by $\omega'(x)$ and noting that $Y'(\omega)\omega'(x)^3 + 2Y(\omega)\omega'(x)\omega''(x) = \partial_x[Y(\omega)\omega'(x)^2]$, we arrive at $\partial_x[X_v(\omega) - Y(\omega)\omega'(x)^2 - C_2\omega(x)] = 0$, or equivalently

$$X(\omega) - Y(\omega)\omega'(x)^2 = C_1 + C_2\omega(x), \qquad (130)$$

with $C_1$ an additional arbitrary constant. Using Eq. (127) we thus have the following ODE for the optimal traveling wave profile $\omega_q(x)$,

$$D(\omega)^2 \left( \frac{d\omega(x)}{dx} \right)^2 = [q - v\bar\rho + v\omega(x)]^2 + \sigma(\omega)^2 E^2 - 2\sigma(\omega)[C_1 + C_2\omega(x)]. \qquad (131)$$

The equation for the optimal velocity follows now from imposing $\frac{dH_v(\omega)}{dv} = \int_0^1 dx \frac{dX_v(\omega)}{dv} = 0$, and using that

$$\frac{dX_v(\omega)}{dv} = -(\bar\rho - \omega)\frac{[q - v(\bar\rho - \omega)]}{\sigma(\omega)} = v\frac{(\bar\rho - \omega)^2}{\sigma(\omega)} - q\frac{(\bar\rho - \omega)}{\sigma(\omega)},$$

see Eq. (127), we hence obtain

$$v = q\frac{\displaystyle\int_0^1 dx \frac{[\bar\rho - \omega_q(x)]}{\sigma[\omega_q(x)]}}{\displaystyle\int_0^1 dx \frac{[\bar\rho - \omega_q(x)]^2}{\sigma[\omega_q(x)]}}, \qquad (132)$$

with $\omega_q(x)$ the solution to Eq. (131). It is worth emphasizing that the optimal velocity is proportional to $q$. This implies that the optimal wave profile depends exclusively on $q^2$ and not on the current sign, see Eq. (131)), reflecting the Gallavotti-Cohen time-reversal symmetry [9–12,59]. This invariance of the optimal wave profile under the transformation $q \leftrightarrow -q$ can now be used in Eq. (126) to show explicitly the fluctuation theorem $G(q) - G(-q) = 2Eq$ for the symmetry-broken traveling wave phase.

Equation (131) for the optimal wave is also invariant under reflection transformations $x \to 1-x$, and hence it generically yields a *symmetric* profile $\omega_q(x) = \omega_q(1-x)$. The wave extrema $\omega_i = \omega_q(x_i)$ are obtained by setting $\omega'_q(x_i) = 0$ in Eq. (131). Periodicity then implies that wave extrema, if any, come in pairs (maximum and minimum). For many of the transport models studied in literature, as e.g. the WASEP model of particle diffusion or the KMP model of heat conduction, for which the mobility $\sigma(\rho)$ is quadratic in the density field, the wave profile turns out to have a single pair of extrema, i.e. a single minimum $\omega_0 = \omega_q(x_0)$ and a corresponding single maximum $\omega_1 = \omega_q(x_1)$. This, together with the wave reflection symmetry, implies that the distance between extrema is just half of the system, $|x_0 - x_1| = 1/2$. The constants $C_1$ and $C_2$ can be expressed in terms of these extrema, i.e.

$$X(\omega_i) = C_1 + C_2 \omega_i \qquad \text{for } i = 0, 1, \tag{133}$$

see Eq. (130). Moreover, the extrema locations are fixed by the constraints on the distance between them, $\frac{1}{2} = \int_{x_0}^{x_1} dx = \int_{\omega_0}^{\omega_1} d\omega_q / \omega'_q$, and the total density of the system, $\frac{\bar{\rho}}{2} = \int_{x_0}^{x_1} \omega_q(x) dx = \int_{\omega_0}^{\omega_1} d\omega_q \, \omega_q / \omega'_q$, or equivalently

$$\frac{1}{2} = \int_{\omega_0}^{\omega_1} \sqrt{\frac{Y(\omega_q)}{X(\omega_q) - C_1 - C_2 \omega_q}} d\omega_q, \qquad \frac{\bar{\rho}}{2} = \int_{\omega_0}^{\omega_1} \sqrt{\frac{\omega_q^2 Y(\omega_q)}{X(\omega_q) - C_1 - C_2 \omega_q}} d\omega_q, \tag{134}$$

where we have used the differential equation (131) to substitute $\omega'_q(x)$. In this way, for fixed values of the current $q$ and the density $\bar{\rho}$, we can use Eqs. (133)-(134) to determine the four constants $\omega_1, \omega_0, C_1, C_2$ which can be used in turn to obtain the shape of the optimal density profile $\omega_q(x)$ and its velocity from Eqs. (131)-(132) in an iterative manner [14,43,44, 53]. Notice that the unknown variables $\omega_0, \omega_1$ appear as the integration limits in Eq. (134), making this problem remarkably difficult to solve numerically. This challenge can be overcome however by a suitable change of variables, and we refer the interested reader to Refs. [14,43, 44,53] for details on this issue.

Compelling evidences of this continuous time-translation symmetry breaking phenomenon at the fluctuating level have been found in the current fluctuations of different models of transport [14,29,43,44,47,56,108,157]. This is the case for instance of the 1$d$ KMP model of heat conduction on a ring [14,44,119], see Fig. 5.a, for which the relevant transport coefficients are $D(\rho) = 1/2$ and $\sigma(\rho) = \rho^2$. In particular, small energy current fluctuations in this model result from the sum of weakly-correlated local random events in the energy density field, thus giving rise to Gaussian current statistics in this regime as dictated by the central limit theorem, and supported by a flat, structureless optimal energy field in average. Since $\sigma''(\rho) > 0$ in this model, this Gaussian homogeneous regime is expected for currents $|q| \le q_c$. The top-left panel in Fig. 5.a depicts a typical spatiotemporal trajectory of the energy field in this model for a current fluctuation $|q| < q_c$, showing no relevant structures whatsoever. However for large enough currents, $|q| > q_c$, the KMP system self-organizes into a coherent traveling wave which facilitates this rare fluctuation by packing energy in a localized condensate, see top-right panel in Fig. 5.a. The bottom panel in Fig. 5.a shows the shape of the energy traveling condensate predicted by MFT [14]. For the WASEP particle diffusion model the picture is somewhat similar, see Fig. 5.b, but with a different physical interpretation. In this case the transport

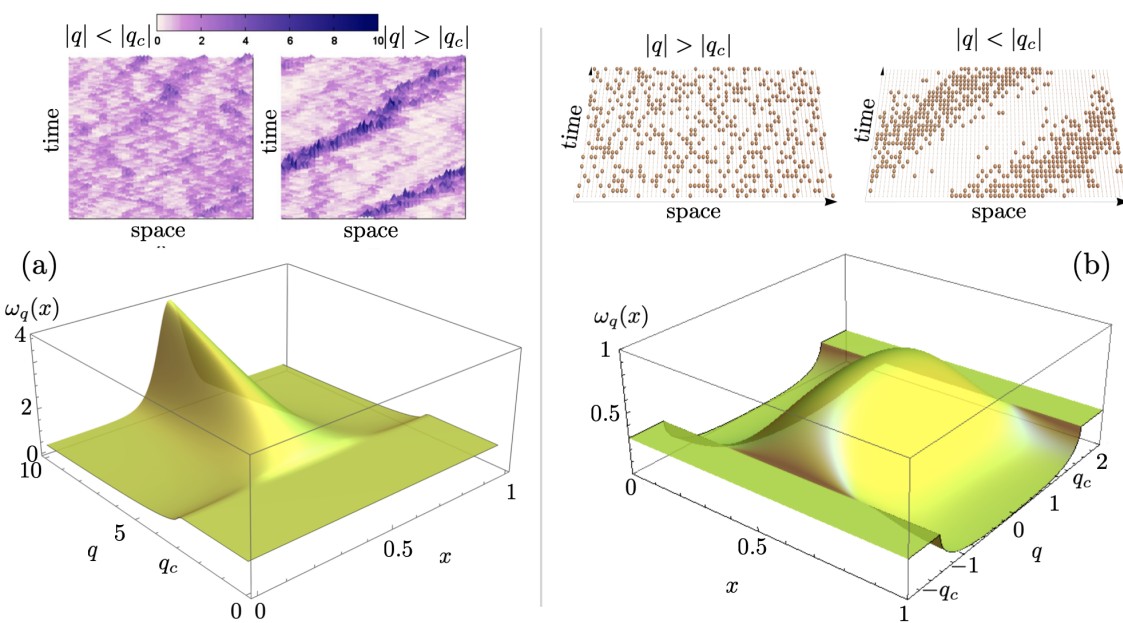

Figure 5: **Time-translation symmetry breaking in** $1d$ **periodic systems.** Typical evolution of microscopic configurations for current fluctuations $q$ above and below the critical threshold, and shape of the optimal density field as a function of $q$, for two different transport models on $1d$ periodic lattices, namely (a) the Kipnis-Marchioro-Presutti (KMP) model of heat conduction [44, 119], and (b) the weakly-asymmetric simple exclusion process (WASEP), a model of particle diffusion under exclusion interactions [44, 120–122]. Top-left panels in (a) and (b) correspond to typical trajectories for current fluctuations in the homogeneous phase, while top-right panels display typical trajectories in the time-translation symmetry-broken phase, where traveling waves of energy (a) or particles (b) emerge. This happens for $|q| > q_c$ in the KMP model and for $|q| < q_c$ in WASEP, depending on the sign of $\bar{\sigma}''$. Bottom panels in (a) and (b) show the MFT prediction for the optimal density profile $\omega_q(x)$ as a function of $x$ for different $q$'s. Density profiles are flat up to the critical current, beyond which a nonlinear wave pattern develops, moving at constant velocity. For the KMP panel (a) we take $E = 0$ and $\bar{\rho} = 1$ [14], while for the WASEP panel (b) we have $E = 10 > E_c$ and $\bar{\rho} = 0.3$ [43]. Image adapted from [108].

coefficients are $D(\rho) = 1/2$ and $\sigma(\rho) = \rho(1 - \rho)$, such that $\sigma''(\rho) < 0$. For small current fluctuations around the average $\langle q \rangle = \bar{\sigma} E$ the typical density profile is flat, while for $|q| \leq q_c$ (since $\sigma''(\rho) < 0$) a traveling wave dominates the typical trajectory, see top-right panel in Fig. 5.b, corresponding to the emergence of a macroscopic jammed state which hinders transport of particles to facilitate a current fluctuation well below the average. The predicted shape of this jammed condensate also reflects the particles' exclusion interaction, see bottom panel in Fig. 5.b.

The vector current statistics of periodic $d$-dimensional anisotropic driven diffusive systems under a driving field **E** has been also recently studied [56], using techniques similar to those described here. In this case the complex interplay among vector currents, the external field and anisotropy in high dimension leads to rich dynamical phase diagrams. A detailed local stability analysis of the resulting MFT equations for this broad family of systems [56], similar to the one developed in section §5.1, demonstrates the existence of a 2nd-order dynamic phase transition (DPT) at a critical current separating a homogeneous fluctuation phase with Gaussian current statistics and constant, structureless optimal fields, $\rho_{\mathbf{q}}(\mathbf{r}, t) = \bar{\rho}$ and

$\mathbf{j_q}(\mathbf{r}, t) = \mathbf{q}$, and a symmetry-broken non-Gaussian phase characterized by the emergence of coherent traveling waves with structure along a dominant direction, $\rho_\mathbf{q}(\mathbf{r}, t) = \omega_\mathbf{q}(x_\parallel - vt)$ and $\mathbf{j_q}(\mathbf{r}, t) = \mathbf{j_q}(x_\parallel - vt)$, with $v$ some velocity [56]. Interestingly, for mild or no anisotropy, different traveling-wave phases appear depending on the current separated by lines of 1st-order DPTs, a degeneracy which disappears beyond a critical anisotropy. This richness of the fluctuation phase diagram stems, as in section §3.1 above on the weak additivity principle, from the relevance of structured current fields at the fluctuating level in dimension $d > 1$. In particular, the continuity equation $\partial_t \rho_\mathbf{q} + \boldsymbol{\nabla} \cdot \mathbf{j_q} = 0$ applied to $1d$-like traveling-wave structures implies that $\partial_\parallel [j_{\parallel,\mathbf{q}}(z_\parallel) - v\omega(z_\parallel)] = 0$, where we have defined $z_\parallel \equiv x_\parallel - vt$, and this together with the constraint on the empirical vector current leads to $j_{\parallel,\mathbf{q}}(z_\parallel) = q_\parallel - v[\bar{\rho} - \omega_\mathbf{q}(z_\parallel)]$. On the other hand, all orthogonal current components follow directly from our theorem above as $\mathbf{j_{\perp,q}}(z_\parallel) = \mathbf{q}_\perp \sigma[\omega_\mathbf{q}(z_\parallel)]/\int_0^1 dy\, \sigma[\omega_\mathbf{q}(y)]$. This result, which is markedly different from the traveling-wave structure found in $1d$ models [14, 29, 43] described above, is a direct consequence of the general condition (45). Remarkably, these predictions and the resulting complex dynamical phase diagram have been corroborated in massive rare event simulations of the $2d$ WASEP model [56].

# 6 Spectral signatures of dynamical criticality at the fluctuation level

We have discussed in previous sections how to understand the current statistics of many nonequilibrium driven diffusive systems at the hydrodynamic level, using macroscopic fluctuation theory, the additivity principle conjecture (together with its weak version) and local stability analysis as main tools, and paying special attention to the typical or dominant path in mesoscopic phase space responsible of a given current fluctuation. We have shown at this mesoscopic level that, in some cases, when large enough current fluctuations come about, the associated optimal trajectory may change drastically, in a singular manner, reflecting an underlying dynamic phase transition at the fluctuating level [14,29,43,44,52,53,66–71]. The different dynamical phases in these DPTs correspond to different types of trajectories adopted by the system to sustain atypical values of the current. Interestingly, some of these dynamical phases may display emergent order and collective rearrangements in their trajectories, including symmetry-breaking phenomena and violations of the additivity hypothesis [14,29,43,44,108], while the LDFs controlling the statistics of these fluctuations exhibit non-analyticities at the DPT [72–79] reminiscent of standard critical behavior. At this mesoscopic level the existence of DPTs is governed by the MFT action functional (7) and its symmetries [28,29,52,129], which can be spontaneously broken at the DPT, in the sense that the optimal trajectory responsible of a given fluctuation may or may not inherit the MFT action symmetries.

A different, complementary path to investigate current statistics and the physics of DPTs consists in analyzing these phenomena in terms of microscopic dynamics, as described by the corresponding stochastic generator of the model of interest [107, 147]. We explore now this microscopic view on current fluctuations using spectral methods as a central tool [102]. In particular, we want to understand how the optimal path to a current fluctuation emerges from this microscopic dynamical perspective, providing at the same time a spectral point of view on the phenomenon of symmetry-breaking DPTs. In particular, we will describe in this section the generic spectral mechanism giving rise to symmetry-breaking DPTs in many body Markov systems, unveiling how the different dynamical phases emerge from the specific structure of the leading eigenvectors of the generator. This microscopic, spectral description provides a picture complementary to the field-theoretic MFT scenario introduced in previous sections. It also allows us to address a central problem in the theory of fluctuations, i.e. how to make rare

events *typical* with the use of Doob's transform to deduce the necessary external driving field [93, 94, 102, 158, 159]. This can be then used to exploit existing DPTs to engineer and control many-body systems with desired statistics *on demand* [96], for instance to build *programmable* time-crystal phases in nonequilibrium matter [97–103]. We will devote the final part of these lecture notes to this intriguing question. This section (including subsections 6.1 to 6.7) is adapted from Ref. [102] with permission from the American Physical Society.

## 6.1 Statistics of trajectories at the microscopic level and Doob dynamics

We thus start with the microscopic characterization of many-body interacting particle systems, described here as discrete-state stochastic jump processes on a lattice and evolving in continuous time. Using the quantum Hamiltonian formalism for the master equation [106], system states or configurations can be represented as vectors $|C\rangle$ of an orthonormal basis in a Hilbert space $\mathcal{H}$ satisfying $\langle C|C'\rangle = \delta_{CC'}$. This allows us to write the state of the system at time $t$ as a probability vector $|P_t\rangle = \sum_C P(C,t)|C\rangle$ whose real entries $P(C,t)$ correspond to the probability of finding the system in configuration $C$ at time $t$, such that $0 \le P(C,t) \le 1$. The time evolution of this probability vector $|P_t\rangle \in \mathcal{H}$ is controlled by a master equation $\partial_t |P_t\rangle = \hat{\mathbb{W}}|P_t\rangle$, where $\hat{\mathbb{W}}$ is the Markov generator of the dynamics [160],

$$\hat{\mathbb{W}} = \sum_{C,C'\neq C} W_{C\to C'}|C'\rangle\langle C| - \sum_C R_C|C\rangle\langle C|, \qquad (135)$$

where $W_{C\to C'}$ is the transition rate from configuration $C$ to $C'$, and $R_C \equiv \sum_{C'\neq C} W_{C\to C'}$ is the escape rate from configuration $C$. Since this generator is stochastic (i.e., it conserves probability), we have that $\langle -|\hat{\mathbb{W}} = 0$, where $\langle -|$ is the "flat" state defined as $\langle -| = \sum_C \langle C|$. This ensures that the probability vector remains normalized at all times, i.e., $\langle -|P_t\rangle = \sum_C P(C,t) = 1 \; \forall t$.

We are interested in the statistics of the time-averaged current in the ensemble of all possible system trajectories $\omega_\tau \equiv \{(C_i, t_i)\}_{i=0,1,\dots,m}$ of duration $\tau$. Each trajectory is completely specified by the sequence of configurations visited by the system, $\{C_i\}_{i=0,1,\dots,m}$, and the times at which the jumps between them occur, $\{t_i\}_{i=0,\dots m}$, with $m$ being the number of transitions throughout the trajectory,

$$\omega_\tau : \; C_0 \xrightarrow{t_1} C_1 \xrightarrow{t_2} C_2 \xrightarrow{t_3} (\cdots) \xrightarrow{t_m} C_m, \qquad (136)$$

with $t_0 = 0$ the time origin. The probability distribution of a trajectory (which is continuous on the transition times $\{t_i\}_{i=1,\dots,m}$ and discrete on the visited configurations $\{C_i\}_{i=0,1,\dots,m}$ and the number of transitions $m$) is then given by

$$P[\omega_\tau] = e^{-(\tau-t_m)R_{C_m}} W_{C_{m-1}\to C_m} \cdots e^{-(t_2-t_1)R_{C_1}} W_{C_0\to C_1} e^{-t_1 R_{C_0}} P(C_0, 0). \qquad (137)$$

The total current integrated along a given trajectory can be written as $Q(\omega_\tau) = \sum_{i=0}^{m-1} q_{C_i\to C_{i+1}}$, where $q_{C_i\to C_{i+1}}$ is the elementary current in the transition $C_i \to C_{i+1}$. Thus, the probability of having a given value of $q = Q/\tau$ after a time $\tau$ is simply $P_\tau(q) = \sum_{\omega_\tau} P[\omega_\tau]\delta[\tau q - Q(\omega_\tau)]$, and obeys a large deviation principle for long times [1, 2, 44], $P_\tau(q) \asymp e^{+\tau G(q)}$, where $G(q)$ is the current LDF.

To obtain a microscopic expression for the current LDF (or rather for its Legendre transform, see below), we now define the probability $P_\tau(C; Q)$ of being in configuration $C$ having a time-integrated current $Q$ up to time $\tau$. This probability evolves in time according to the following master equation

$$\partial_\tau P_\tau(C; Q) = \sum_{C'\neq C} W_{C'\to C} P_\tau(C', Q - q_{C'\to C}) - R_C P_\tau(C; Q). \qquad (138)$$

Note that in general $P_\tau(q) = \sum_C P_\tau(C; \tau q)$. Eq. (138) yields a complex hierarchy of equations for the configurational statistics for the different currents. It turns out now to be more convenient to work with the Laplace transform of $P_\tau(C; Q)$, i.e. $Z_\tau(C; \lambda) = \sum_Q e^{\lambda Q} P_\tau(C; Q)$, whose time evolution is given, by virtue of Eq. (138), by

$$\partial_\tau Z_\tau(C; \lambda) = \sum_{C' \neq C} W^\lambda_{C' \to C} Z_\tau(C'; \lambda) - R_C Z_\tau(C; \lambda), \tag{139}$$

where $W^\lambda_{C' \to C} = e^{\lambda q_{C' \to C}} W_{C' \to C}$ stands for the *biased* or *tilted* transition rates [1, 44, 102, 107, 108, 147]. We may write this *biased generator* in operational form as

$$\hat{\mathbb{W}}^\lambda = \sum_{C, C' \neq C} e^{\lambda q_{C \to C'}} W_{C \to C'} |C'\rangle\langle C| - \sum_C R_C |C\rangle\langle C|. \tag{140}$$

Summing now $Z_\tau(C; \lambda)$ over every configuration $C$ we obtain $Z_\tau(\lambda) = \sum_C Z_\tau(C; \lambda)$, which is the so-called *dynamical partition function*, or equivalently the moment generating function for the current distribution, $Z_\tau(\lambda) = \sum_Q e^{\lambda Q} P_\tau(Q)$. For long times this dynamical partition function follows a large deviation principle [8, 44, 107],

$$Z_\tau(\lambda) \asymp e^{\tau \theta(\lambda)} \tag{141}$$

with $\theta(\lambda)$ the scaled cumulant generating function, $\theta(\lambda) = \lim_{\tau \to \infty} \tau^{-1} \ln Z_\tau(\lambda)$, whose derivatives provide the cumulants of the time-averaged current in the biased ensemble. It is then easy to see using the Laplace-transform relation $Z_\tau(\lambda) = \sum_Q e^{\lambda Q} P_\tau(Q)$ and the large deviation scaling of $P_\tau(Q)$ that $\theta(\lambda)$, also known as the *dynamical free energy*, corresponds to Legendre-Fenchel transform [8] of the current LDF, i.e.

$$\theta(\lambda) = \max_q \left[ G(q) + \lambda q \right]. \tag{142}$$

Using the biased generator (140) one can show now that the dynamical partition function can be expressed in operational form as [107]

$$Z_\tau(\lambda) = \langle -|e^{\tau \hat{\mathbb{W}}^\lambda}|P_0\rangle, \tag{143}$$

where $|P_0\rangle \in \mathcal{H}$ is an arbitrary initial state. This operational expression hence allows us to relate the dynamical free energy $\theta(\lambda)$ with the spectrum of the (generally non-symmetric) tilted generator $\hat{\mathbb{W}}^\lambda$. In particular, let $|R^\lambda_j\rangle$ and $\langle L^\lambda_j|$ be the $j$-th right and left eigenvectors of $\hat{\mathbb{W}}^\lambda$, such that

$$\hat{\mathbb{W}}^\lambda |R^\lambda_j\rangle = \theta^\lambda_j |R^\lambda_j\rangle, \qquad \langle L^\lambda_j| \hat{\mathbb{W}}^\lambda = \theta^\lambda_j \langle L^\lambda_j|, \tag{144}$$

with $\theta^\lambda_j \in \mathbb{C}$ the corresponding eigenvalue, ordered in decreasing value of their real part. In most models of interest, the set of left and right eigenvectors form a complete biorthogonal basis of the Hilbert space, such that $\langle L^\lambda_i | R^\lambda_j \rangle = \delta_{ij}$. In this way, using now the spectral decomposition $\hat{\mathbb{W}}^\lambda = \sum_j \theta^\lambda_j |R^\lambda_j\rangle\langle L^\lambda_j|$, it is straightforward to show from Eqs. (143) and (141) that for long times $\tau$ the dynamical free energy is given by the eigenvalue of $\hat{\mathbb{W}}^\lambda$ with the largest real part, $\theta(\lambda) = \theta^\lambda_0$.

Before continuing our discussion, we note that this formalism can be extended to general trajectory-dependent observables, including e.g. configurational observables such as the energy, magnetization, etc., or jump-like observables (similar to the current) such as the dynamic activity; see e.g. [102] for a general discussion. We also want to highlight that a genetic-algorithm interpretation of the tilted or biased generator $\hat{\mathbb{W}}^\lambda$ in Eq. (140) is the basis of the successfull cloning Monte Carlo method to sample rare events in simulations of many-body systems [31, 44, 104, 105, 108, 161, 162].

Notice that, except for $\lambda = 0$ where the original (unbiased) dynamics is recovered, $\hat{\mathbb{W}}^\lambda$ does not conserve probability, i.e., $\langle -|\hat{\mathbb{W}}^\lambda \neq 0$, and therefore it is not a proper stochastic generator. This implies that it is not possible to directly retrieve from the tilted generator the physical trajectories leading to the current fluctuation $q(\lambda)$, since $\hat{\mathbb{W}}^\lambda$ does not represent an actual physical dynamics. To overcome this issue and obtain the physical dynamics making this rare event *typical* for each $\lambda$, we introduce an auxiliary or driven process built from $\hat{\mathbb{W}}^\lambda$, known as the Doob transformed generator [93, 94, 102, 158, 159]

$$\hat{\mathbb{W}}^\lambda_D = \hat{L}^\lambda_0 \hat{\mathbb{W}}^\lambda (\hat{L}^\lambda_0)^{-1} - \theta^\lambda_0 \hat{\mathbb{1}} \,, \tag{145}$$

where $\hat{L}^\lambda_0$ is a diagonal matrix built with the components of the leading left eigenvector, $(\hat{L}^\lambda_0)_{ij} = (\langle L^\lambda_0|)_i \delta_{ij}$, and $\hat{\mathbb{1}}$ is the identity matrix. The spectra of both generators, $\hat{\mathbb{W}}^\lambda$ and $\hat{\mathbb{W}}^\lambda_D$, are related by a shift in their eigenvalues, $\theta^\lambda_{j,D} = \theta^\lambda_j - \theta^\lambda_0$, and a simple transformation of their left and right eigenvectors, $\langle L^\lambda_{j,D}| = \langle L^\lambda_j|(\hat{L}^\lambda_0)^{-1}$ and $|R^\lambda_{j,D}\rangle = \hat{L}^\lambda_0 |R^\lambda_j\rangle$. As a consequence, the leading eigenvalue of $\hat{\mathbb{W}}^\lambda_D$ becomes zero and its associated leading right eigenvector, given by $|R^\lambda_{0,D}\rangle = \hat{L}^\lambda_0 |R^\lambda_0\rangle$, becomes the Doob stationary state. In addition, the leading left eigenvector is just the flat vector, $\langle L^\lambda_{0,D}| = \langle L^\lambda_0|(\hat{L}^\lambda_0)^{-1} = \langle -|$, confirming that the Doob generator does conserve probability, $\langle -|\hat{\mathbb{W}}^\lambda_D = 0$. In this way, the Doob generator(145) provides the physical trajectories distributed according to the $\lambda$-ensemble [95], revealing how particular current fluctuations are created in time. The left and right eigenvectors of the Doob generator also form a complete biorthogonal basis of the Hilbert space, and they are further normalized such that $\max_C |\langle L^\lambda_{j,D}|C\rangle| = 1$ and $\langle L^\lambda_{i,D}|R^\lambda_{j,D}\rangle = \delta_{ij}$ [163]. Note that this normalization specifies the eigenvectors with $j > 0$ up to an arbitrary complex phase. In addition, due to conservation of probability, $\langle -|\hat{\mathbb{W}}^\lambda_D = 0$, we have that $\langle -|R^\lambda_{j,D}\rangle = 0$ for all $j \neq 0$.

In the following subsections we want to analyze the general structure of the Doob eigenvectors across a generic dynamical phase transition, with the aim of unveiling how the different dynamical phases emerge from the microscopic dynamics when an underlying symmetry is broken. In many cases, DPTs involve the emergence of distinct $\mathbb{Z}_n$ symmetry-broken patterns [52, 71, 109, 134], which might be time dependent [14, 43, 56, 57, 101] as in the traveling-wave cases mentioned above, facilitating the corresponding fluctuation. In order to continue, we hence need first to specify what a $\mathbb{Z}_n$ symmetry is in this context.

## 6.2 $\mathbb{Z}_n$ symmetry

Our interest on symmetry-breaking DPTs hence calls for some general remarks about symmetry aspects of stochastic processes [164]. In particular, we are interested in symmetry properties under state space transformations of the original stochastic process, as defined by the generator $\hat{\mathbb{W}}$, and how these properties are inherited by the Doob auxiliary process $\hat{\mathbb{W}}^\lambda_D$ associated with the fluctuations of the time-averaged current. The set of transformations that leave a stochastic process *invariant* (as defined below) form a symmetry group. For discrete state space, any such symmetries correspond to permutations in configuration space [164] and hence are described by the $\mathbb{Z}_n$ group, a cyclic group of order $n$. Its elements can be thus built from the repeated application of a single operator $\hat{S} \in \mathbb{Z}_n$, which satisfies $\hat{S}^n = \hat{\mathbb{1}}$. This operator is then unitary, probability-conserving so $\langle -|\hat{S} = \langle -|$, invertible, and characterized by real and non-negative matrix elements. Moreover, being a symmetry, it commutes with the generator of the stochastic dynamics, $[\hat{\mathbb{W}}, \hat{S}] = 0$, or equivalently $\hat{\mathbb{W}} = \hat{S}\hat{\mathbb{W}}\hat{S}^{-1}$. The action of $\hat{S}$ on configurations produces a bijective transformation of state space, such that $\hat{S}|C\rangle = |C_S\rangle \in \mathcal{H}$. This transformation induces a corresponding mapping $\mathcal{S}$ in trajectory space

$$\omega_\tau : C_0 \xrightarrow{t_1} C_1 \xrightarrow{t_2} \dots \xrightarrow{t_m} C_m \qquad \xrightarrow{\mathcal{S}} \qquad \mathcal{S}\omega_\tau : C_{S0} \xrightarrow{t_1} C_{S1} \xrightarrow{t_2} \dots \xrightarrow{t_m} C_{Sm} \,, \tag{146}$$

transforming the configurations visited along the path, but leaving unchanged the transition times $\{t_i\}_{i=1,\dots,m}$ between them. For the symmetry to be inherited by the Doob auxiliary process $\hat{\mathbb{W}}_D^\lambda$, it is clearly necessary that the trajectory-dependent current observable remains invariant under any trajectory transformation, i.e. $Q(\mathcal{S}\omega_\tau) = Q(\omega_\tau) \; \forall \omega_\tau$. This condition, together with the invariance of the original dynamics under $\hat{S}$, i.e. $[\hat{\mathbb{W}}, \hat{S}] = 0$, crucially implies that both the tilted and the Doob generators are also invariant under $\hat{S}$ [102], $\hat{\mathbb{W}}^\lambda = \hat{S}\hat{\mathbb{W}}^\lambda\hat{S}^{-1}$ and $\hat{\mathbb{W}}_D^\lambda = \hat{S}\hat{\mathbb{W}}_D^\lambda\hat{S}^{-1}$. As a consequence, both $\hat{\mathbb{W}}_D^\lambda$ and $\hat{S}$ share a common eigenbasis and can be diagonalized together, so $|R_{j,D}^\lambda\rangle$ and $\langle L_{j,D}^\lambda|$ are also eigenvectors of $\hat{S}$ with eigenvalues $\phi_j$,

$$\hat{S}|R_{j,D}^\lambda\rangle = \phi_j|R_{j,D}^\lambda\rangle, \qquad \langle L_{j,D}^\lambda|\hat{S} = \phi_j\langle L_{j,D}^\lambda|.$$

Due to the unitarity and cyclic character of $\hat{S}$, the eigenvalues $\phi_j$ simply correspond to the $n$ roots of unity, i.e. $\phi_j = e^{i2\pi k_j/n}$ with $k_j = 0, 1, \dots, n-1$.

## 6.3 Dynamical phase transitions and spectrum degeneracy

The steady state corresponding to the Doob stochastic generator $\hat{\mathbb{W}}_D^\lambda$ characterizes the trajectory statistics during a large deviation event parameterized by $\lambda$. The formal solution to the Doob master equation at any time $t$, starting from an initial probability vector $|P_0\rangle$, is given by $|P_{t,P_0}^\lambda\rangle = \exp(+t\hat{\mathbb{W}}_D^\lambda)|P_0\rangle$. By introducing a spectral decomposition of this formal solution, we obtain

$$|P_{t,P_0}^\lambda\rangle = |R_{0,D}^\lambda\rangle + \sum_{j>0} e^{t\theta_{j,D}^\lambda}|R_{j,D}^\lambda\rangle\langle L_{j,D}^\lambda|P_0\rangle, \tag{147}$$

where we have already used that the Doob generator is stochastic and hence $\theta_{0,D}^\lambda = 0$. Additionally, since $\langle -|R_{j,D}^\lambda\rangle = \langle L_{0,D}^\lambda|R_{j,D}^\lambda\rangle = \delta_{0j}$, the entire probability of $|P_{t,P_0}^\lambda\rangle$ is concentrated in $|R_{0,D}^\lambda\rangle$, meaning that $\langle -|P_{t,P_0}^\lambda\rangle = \langle -|R_{0,D}^\lambda\rangle = 1$. Consequently, each term with $j > 0$ on the right-hand side of Eq. (147) represents a specific redistribution of the probability. Furthermore, since the symmetry operator $\hat{S}$ preserves probability, we obtain $1 = \langle -|\hat{S}|R_{0,D}^\lambda\rangle = \langle -|\phi_0|R_{0,D}^\lambda\rangle$, which indicates that $\phi_0 = 1$ for the symmetry eigenvalue of the leading eigenvector. This result shows that $|R_{0,D}^\lambda\rangle$ remains unchanged under the action of the symmetry operator.

To analyze the steady state of the Doob dynamics, defined as $|P_{ss,P_0}^\lambda\rangle \equiv \lim_{t\to\infty}|P_{t,P_0}^\lambda\rangle$, we introduce the spectral gaps $\Delta_j^\lambda = \text{Re}(\theta_0^\lambda - \theta_j^\lambda) = -\text{Re}(\theta_{j,D}^\lambda) \geq 0$, which determine the exponential decay of the corresponding eigenvectors, see Eq. (147). Note that the ordering of eigenvalues by their real part implies $0 \leq \Delta_1^\lambda \leq \Delta_2^\lambda \leq \dots$ When $\Delta_1^\lambda$ (commonly referred to as *the* spectral gap) is strictly positive, $\Delta_1^\lambda > 0$, the spectrum is gapped and the contribution of all subleading eigenvectors in Eq. (147) decays exponentially on timescales $t \gg 1/\Delta_1^\lambda$, resulting in a unique Doob steady state

$$|P_{ss,P_0}^\lambda\rangle = |R_{0,D}^\lambda\rangle. \tag{148}$$

This steady state preserves the symmetry of the Doob generator, $\hat{S}|P_{ss,P_0}^\lambda\rangle = |P_{ss,P_0}^\lambda\rangle$, meaning that symmetry-breaking cannot occur at the fluctuating level when the spectrum of $\hat{\mathbb{W}}_D^\lambda$ is gapped. This is the spectral scenario prior to any dynamical phase transition. Conversely, any symmetry-breaking DPT will require an emergent degeneracy in the leading eigenvectors of the corresponding Doob generator. This corresponds to the spectral signatures typically associated with symmetry-breaking phase transitions in stochastic systems [163–167]. Since the Doob auxiliary generator $\hat{\mathbb{W}}_D^\lambda$ is inherently stochastic, these spectral signatures [163] also characterize DPTs at the level of fluctuations. Specifically, for a many-body stochastic system undergoing a $\mathbb{Z}_n$ symmetry-breaking DPT, we anticipate that the difference between the real part of the first eigenvalue and those of the next $n-1$ eigenvalues $\theta_{j,D}^\lambda$ vanishes in the thermodynamic limit as the DPT sets in. In such a scenario, the *stationary* probability vector of the

Doob process is determined by the first $n$ eigenvectors, which form the degenerate eigenspace. However, according to the Perron-Frobenius theorem, the steady state remains non-degenerate for any finite system size, underscoring the importance of considering also the thermodynamic limit.

In general, the eigenvalues associated with these gap-closing eigenvectors may have non-zero imaginary parts, $\text{Im}(\theta_{j,\text{D}}^{\lambda}) \neq 0$, resulting in a time-dependent Doob *stationary* probability vector in the thermodynamic limit

$$|P_{\text{ss},P_0}^{\lambda}\rangle(t) = |R_{0,\text{D}}^{\lambda}\rangle + \sum_{j=1}^{n-1} \text{e}^{+it\text{Im}(\theta_{j,\text{D}}^{\lambda})}|R_{j,\text{D}}^{\lambda}\rangle\langle L_{j,\text{D}}^{\lambda}|P_0\rangle. \tag{149}$$

Furthermore, if these imaginary parts exhibit a band structure, the resulting Doob *stationary* state will display robust and stable periodic motion, a hallmark of time-crystalline phases [97–103]. This can be used in turn to build time crystals from the rare event statistics of some many-body systems, as we will see below [101]. However, in many scenarios, the gap-closing eigenvalues of the Doob eigenvectors within the degenerate subspace are purely real, meaning $\text{Im}(\theta_{j,\text{D}}^{\lambda}) = 0$. In this case, the resulting Doob steady state is genuinely stationary

$$|P_{\text{ss},P_0}^{\lambda}\rangle = |R_{0,\text{D}}^{\lambda}\rangle + \sum_{j=1}^{n-1} |R_{j,\text{D}}^{\lambda}\rangle\langle L_{j,\text{D}}^{\lambda}|P_0\rangle. \tag{150}$$

The number $n$ of vectors contributing to the Doob steady state corresponds to the number of distinct phases that emerge when the $\mathbb{Z}_n$ symmetry is broken. Specifically, an $n$-fold degeneracy of the leading eigenspace results in the appearance of $n$ distinct, linearly independent stationary distributions [164, 165], as we will demonstrate below. As in the general time-dependent solution (147), all the probability remains concentrated in the first eigenvector $|R_{0,\text{D}}^{\lambda}\rangle$, which preserves the symmetry, $\hat{S}|R_{0,\text{D}}^{\lambda}\rangle = |R_{0,\text{D}}^{\lambda}\rangle$. The remaining eigenvectors in the degenerate subspace capture the *redistribution* of this probability based on their projection $\langle L_{j,\text{D}}^{\lambda}|P_0\rangle$ onto the initial state, $j > 0$, thereby encoding all the information about the symmetry-breaking process.

It is important to note that even when the first $n$ eigenvalues are fully degenerate, $|R_{0,\text{D}}^{\lambda}\rangle$ can still be distinguished as the only eigenvector with eigenvalue $\phi_0 = 1$ under $\hat{S}$. The other gap-closing eigenvectors can be shown to have different eigenvalues under $\hat{S}$ [102]. Consequently, the steady state described by Eq.(150) does not generally preserve the symmetry of the generator, $\hat{S}|P_{\text{ss},P_0}^{\lambda}\rangle \neq |P_{\text{ss},P_0}^{\lambda}\rangle$, indicating that the symmetry is broken in the degenerate phase. The same applies to the time-dependent asymptotic Doob state given in Eq.(149).

## 6.4 Phase probability vectors

Our next goal is to identify the $n$ distinct and linearly independent stationary distributions $|\Pi_l^{\lambda}\rangle \in \mathcal{H}$, with $l = 0, 1, \ldots, n-1$, that arise at the DPT when degeneracy occurs [163–167]. Each of these phase probability vectors $|\Pi_l^{\lambda}\rangle$ is uniquely associated with a specific symmetry-broken phase $l \in [0 \, .. \, n-1]$, and the set spanned by these vectors and their left duals forms a new basis for the degenerate subspace. Thus, any phase probability vector $|\Pi_l^{\lambda}\rangle$ can always be expressed as a linear combination of the Doob eigenvectors within the degenerate subspace,

$$|\Pi_l^{\lambda}\rangle = \sum_{j=0}^{n-1} C_{l,j}|R_{j,\text{D}}^{\lambda}\rangle, \tag{151}$$

where the coefficients $C_{l,j}$ are complex numbers in general, $C_{l,j} \in \mathbb{C}$. These phase probability vectors must satisfy two key conditions. First, they must be normalized, such that $\langle -|\Pi_l^{\lambda}\rangle = 1$

for all $l \in [0 \dots n-1]$. Second, and most importantly, they must be connected through the symmetry operator,

$$|\Pi_{l+1}^\lambda\rangle = \hat{S}|\Pi_l^\lambda\rangle, \tag{152}$$

which leads to the relationship $|\Pi_l^\lambda\rangle = \hat{S}^l|\Pi_0^\lambda\rangle$ so that the coefficients $C_{l,j} = C_{0,j}(\phi_j)^l$, where $\phi_j$ are the eigenvalues of the symmetry operator. Requiring now that any steady state can be expressed as a statistical mixture (or convex combination) of the various phase probability vectors, it can be demonstrated that $C_{0,j} = 1$ for all $j \in [0 \dots n-1]$. Consequently, the coefficients become $C_{l,j} = (\phi_j)^l$, and the probability vectors associated with the symmetry-broken phases are given by

$$|\Pi_l^\lambda\rangle = \sum_{j=0}^{n-1}(\phi_j)^l|R_{j,\mathrm{D}}^\lambda\rangle = |R_{0,\mathrm{D}}^\lambda\rangle + \sum_{j=1}^{n-1}(\phi_j)^l|R_{j,\mathrm{D}}^\lambda\rangle. \tag{153}$$

It is important to note that the symmetry group broken during the DPT may extend beyond $\mathbb{Z}_n$. However, as long as it includes this cyclic symmetry, the above decomposition remains valid.

It is now helpful to define the left duals $\langle\pi_l^\lambda|$ of the phase probability vectors, which are row vectors that satisfy the biorthogonality condition $\langle\pi_{l'}^\lambda|\Pi_l^\lambda\rangle = \delta_{l',l}$. These left duals can be expressed as linear combinations of the left eigenvectors corresponding to the degenerate subspace, $\langle\pi_l^\lambda| = \sum_{j=0}^{n-1}\langle L_{j,\mathrm{D}}^\lambda|D_{l,j}$, where the coefficients $D_{l,j}$ are complex numbers, $D_{l,j} \in \mathbb{C}$. By enforcing biorthogonality, applying the spectral expansion (151), and noting that the first $n$ eigenvalues $\phi_j$ are precisely the $n$th roots of unity [102], we obtain that $D_{l,j} = \frac{1}{n}(\phi_j)^{-l}$, leading to

$$\langle\pi_l^\lambda| = \sum_{j=0}^{n-1}\frac{1}{n}(\phi_j)^{-l}\langle L_{j,\mathrm{D}}^\lambda|. \tag{154}$$

Using these left duals, we can express now the right eigenvectors $|R_{j,\mathrm{D}}^\lambda\rangle$ within the degenerate subspace, $\forall j \in [0 \dots n-1]$, in terms of the phase probability vectors as follows

$$|R_{j,\mathrm{D}}^\lambda\rangle = \sum_{l=0}^{n-1}|\Pi_l^\lambda\rangle\langle\pi_l^\lambda|R_{j,\mathrm{D}}^\lambda\rangle = \frac{1}{n}\sum_{l=0}^{n-1}(\phi_j)^{-l}|\Pi_l^\lambda\rangle, \tag{155}$$

where we have used the relation $\langle\pi_l^\lambda|R_{j,D}^\lambda\rangle = \frac{1}{n}(\phi_j)^{-l}$. Substituting this decomposition into Eq. (149), we can reconstruct the (degenerate) Doob *steady* state as a weighted sum of the phase probability vectors $|\Pi_l^\lambda\rangle$, each corresponding to one of the $n$ symmetry-broken phases,

$$|P_{\mathrm{ss},P_0}^\lambda\rangle(t) = \sum_{l=0}^{n-1}w_l(t)|\Pi_l^\lambda\rangle, \tag{156}$$

where the generically time-dependent weights $w_l(t) = \langle\pi_l^\lambda|P_{\mathrm{ss},P_0}^\lambda\rangle(t)$ are given by

$$w_l(t) = \frac{1}{n} + \frac{1}{n}\sum_{j=1}^{n-1}e^{+it\mathrm{Im}(\theta_{j,\mathrm{D}}^\lambda)}(\phi_j)^{-l}\langle L_{j,\mathrm{D}}^\lambda|P_0\rangle. \tag{157}$$

In many practical cases, the relevant eigenvalues are purely real, so that $\mathrm{Im}(\theta_{j,\mathrm{D}}^\lambda) = 0$ and the weights $w_l$ are time-independent. In such scenarios, the steady state becomes

$$|P_{\mathrm{ss},P_0}^\lambda\rangle = \sum_{l=0}^{n-1}w_l|\Pi_l^\lambda\rangle, \qquad \text{with } w_l = \frac{1}{n} + \frac{1}{n}\sum_{j=1}^{n-1}(\phi_j)^{-l}\langle L_{j,\mathrm{D}}^\lambda|P_0\rangle. \tag{158}$$

This demonstrates that the Doob steady state can be interpreted as a statistical mixture of the different symmetry-broken phases, represented by their unique phase probability vectors $|\Pi_l^\lambda\rangle$. The weights $w_l$ satisfy $\sum_{l=0}^{n-1} w_l = 1$ and $0 \leq w_l \leq 1$ for all $l \in [0 .. n-1]$. These weights are determined by the projection of the initial state onto the different phases, which depends in turn on the overlaps $\langle L_{j,D}^\lambda | P_0 \rangle$ of the degenerate left eigenvectors with the initial state as well as their associated symmetry eigenvalues.

Equation (158) outlines how to prepare the initial state $|P_0\rangle$ to isolate a specific symmetry-broken phase $|\Pi_{l'}^\lambda\rangle$ in the long-time limit. By comparing Eqs. (150) and (153), it becomes clear that setting $|P_0\rangle$ such that $\langle L_{j,D}^\lambda | P_0 \rangle = (\phi_j)^{l'}$ for all $j \in [1 .. n-1]$ results in a *pure* steady state $|P_{ss,P_0}^\lambda\rangle = |\Pi_{l'}^\lambda\rangle$, meaning $w_l = \delta_{l,l'}$. This approach provides a straightforward mechanism for phase selection through initial state preparation, resembling strategies previously discussed for open quantum systems with strong symmetries [89–91].

It is important to emphasize again that, in general, degeneracy of the leading eigenspace of a given stochastic generator can only occur in the thermodynamic limit. For finite-size systems, small but non-zero spectral gaps $\Delta_j^\lambda$, $j \in [1 .. n-1]$, are always present, meaning that the long-time Doob steady state is $|P_{ss,P_0}^\lambda\rangle = |R_{0,D}^\lambda\rangle$, as given in Eq. (148). This steady state, which can be expressed as $|R_{0,D}^\lambda\rangle = \frac{1}{n} \sum_{l=0}^{n-1} |\Pi_l^\lambda\rangle$, preserves the symmetry of the generator, preventing any symmetry-breaking DPT for finite systems. However, for large but finite system sizes, one can expect an emerging quasi-degeneracy [163, 166] in the parameter regime where the DPT would occur, characterized by $\Delta_j^\lambda / \Delta_n^\lambda \ll 1$ for all $j \in [1 .. n-1]$. Under such conditions, and on timescales $t \ll 1/\Delta_{n-1}^\lambda$ but $t \gg 1/\Delta_n^\lambda$, a sort of metastable form of symmetry breaking may be observed, described by the physical phase probability vectors $|\Pi_l^\lambda\rangle$, with punctuated jumps between different symmetry sectors during this metastable regime. In the long-time limit the original symmetry is effectively restored, however, as the finite system size prevents true symmetry breaking.

## 6.5   Structure of the degenerate manifold

A significant observation is that, once a symmetry-breaking phase transition occurs –whether configurational (i.e., standard) or dynamical– the associated *typical* (or most likely) configurations naturally fall into distinct symmetry classes. In other words, the symmetry is broken even at the level of individual configurations. For example, consider now the well-known $2d$ Ising model and its (standard) $\mathbb{Z}_2$ symmetry-breaking phase transition at the Onsager temperature $T_c$. This transition separates a disordered paramagnetic phase for $T > T_c$ from an ordered, symmetry-broken ferromagnetic phase for $T < T_c$ [152]. At temperatures well below the critical point, the stationary probability of random (symmetry-preserving) spin configurations is vanishingly small, whereas high-probability configurations exhibit a net non-zero magnetization, characteristic of symmetry breaking. This indicates that statistically significant configurations belong to a specific symmetry phase, meaning that they can be associated with the *basin of attraction* of a particular symmetry sector [166].

A similar scenario unfolds in $\mathbb{Z}_n$ symmetry-breaking DPTs. Specifically, once the DPT occurs and the symmetry is broken, statistically-relevant configurations $|C\rangle$ –those for which $\langle C|P_{ss,P_0}^\lambda\rangle = P_{ss,P_0}^\lambda(C)$ is significantly different from zero– belong to distinct symmetry classes labeled by an index $\ell_C \in [0 .. n-1]$. In terms of phase probability vectors, this implies

$$\frac{\langle C|\Pi_l^\lambda\rangle}{\langle C|\Pi_{\ell_C}^\lambda\rangle} \approx 0, \qquad \forall l \neq \ell_C. \tag{159}$$

In other words, the statistically-relevant configurations in the symmetry-broken Doob steady state can be grouped into disjoint symmetry classes. This simple yet fundamental observa-

tion reveals a hidden spectral structure within the degenerate subspace, associated with these configurations. For a configuration $|C\rangle$ belonging to symmetry sector $\ell_C$, Eq. (155) leads to

$$\langle C|R_{j,\text{D}}^\lambda\rangle = \frac{1}{n}\sum_{l=0}^{n-1}(\phi_j)^{-l}\langle C|\Pi_l^\lambda\rangle \approx \frac{1}{n}(\phi_j)^{-\ell_C}\langle C|\Pi_{\ell_C}^\lambda\rangle. \tag{160}$$

In particular, for $j = 0$, we find that $\langle C|R_{0,\text{D}}^\lambda\rangle \approx \frac{1}{n}\langle C|\Pi_{\ell_C}^\lambda\rangle$ since $\phi_0 = 1$, and thus

$$\langle C|R_{j,\text{D}}^\lambda\rangle \approx (\phi_j)^{-\ell_C}\langle C|R_{0,\text{D}}^\lambda\rangle, \tag{161}$$

for $j \in [1 .. n-1]$. This shows that the components $\langle C|R_{j,\text{D}}^\lambda\rangle$ of the subleading eigenvectors in the degenerate subspace associated with statistically-relevant configurations are approximately equal to those of the leading eigenvector $|R_{0,\text{D}}^\lambda\rangle$, but with a complex argument given by $(\phi_j)^{-\ell_C}$. This reveals how the $\mathbb{Z}_n$ symmetry-breaking process imposes a specific structure on the degenerate eigenvectors involved in a continuous DPT. This result relies on the assumption that statistically-relevant configurations can be partitioned into disjoint symmetry classes. This hypothesis is well-supported empirically in different models, see e.g. [102].

## 6.6  Order parameter space

Analyzing the eigenvectors of many-body stochastic systems is often impractical, as the size of configuration Hilbert space $\mathcal{H}$ grows exponentially with system size, and configurations are not naturally categorized by symmetry. To address this problem, we can partition $\mathcal{H}$ into equivalence classes based on a suitable order parameter for the DPT, grouping *similar* configurations by their symmetry properties. This reduces the effective dimensionality of the problem while introducing a natural parameter for analyzing spectral properties. We define the order parameter $\mu$ for the DPT as a map $\mu : \mathcal{H} \to \mathbb{C}$ that assigns each configuration $|C\rangle \in \mathcal{H}$ a complex number $\mu(C)$. The modulus of $\mu(C)$ measures the degree of order (how deep the system is in the symmetry-broken regime), while its argument identifies the phase. For $\mathbb{Z}_n$ symmetry-breaking DPTs, a single complex-valued order parameter suffices [102]. Using $\mu$, we introduce a reduced Hilbert space $\mathcal{H}_\mu = \{||v\rangle\rangle\}$ representing the possible values of the order parameter as vectors $||v\rangle\rangle$ of a biorthogonal basis satisfying $\langle\langle v'||v\rangle\rangle = \delta_{v,v'}$. The dimension of $\mathcal{H}_\mu$ is typically much smaller than $\mathcal{H}$, as the values of $\mu$ often scale linearly with system size.

To map probability vectors from the original Hilbert space $\mathcal{H}$ to the reduced space $\mathcal{H}_\mu$, we define a surjective map $\widetilde{\mathcal{T}} : \mathcal{H} \to \mathcal{H}_\mu$ that assigns *all* configurations $|C\rangle \in \mathcal{H}$ with the same order parameter $v$ to a single vector $||v\rangle\rangle \in \mathcal{H}_\mu$. Crucially, $\widetilde{\mathcal{T}}$ must conserve probability, meaning the total probability of all configurations with a given order parameter in $\mathcal{H}$ must equal the probability of the corresponding vector in $\mathcal{H}_\mu$. For a probability vector $|P\rangle \in \mathcal{H}$ and its reduced counterpart $||P\rangle\rangle = \widetilde{\mathcal{T}}|P\rangle$, this condition reads

$$P(v) \equiv \langle\langle v||P\rangle\rangle = \sum_{\substack{|C\rangle\in\mathcal{H}: \\ \mu(C)=v}}\langle C|P\rangle, \qquad \forall v. \tag{162}$$

This probability-conservation condition constrains the form of $\widetilde{\mathcal{T}}$. For any vector $|\psi\rangle \in \mathcal{H}$, its reduced version $||\psi\rangle\rangle \in \mathcal{H}_\mu$ is thus defined as

$$||\psi\rangle\rangle \equiv \sum_v \langle\langle v||\psi\rangle\rangle \, ||v\rangle\rangle = \sum_v \left[\sum_{|C\rangle\in\mathcal{H}:\,\mu(C)=v}\langle C|\psi\rangle\right]||v\rangle\rangle. \tag{163}$$

A *good* order parameter $\mu$ must effectively distinguish the different symmetry-broken phases and how the symmetry operator $\hat{S}$ relates these phases. Specifically, for each equivalence class

$\{|C\rangle\}_\nu = \{|C\rangle \in \mathcal{H} : \mu(C) = \nu\}$, applying $\hat{S}$ to all configurations in $\{|C\rangle\}_\nu$ spans a new set $\hat{S}(\{|C\rangle\}_\nu)$ which should correspond to the entire equivalence class $\{|C\rangle\}_{\nu'}$ associated with the order parameter vector $||\nu'\rangle\rangle \in \mathcal{H}_\mu$ for $\mu$ to be a good order parameter. In addition, we require that $\mu$ can distinguish any symmetry-broken configuration from its symmetry-transformed counterpart. This introduces a reduced symmetry operator $\hat{S}_\mu$ acting on $\mathcal{H}_\mu$ defining a bijective mapping $\hat{S}_\mu||\nu\rangle\rangle = ||\nu'\rangle\rangle$ between equivalence classes. Mathematically, this mapping can be defined from the relation $\widetilde{\mathcal{T}}\hat{S}|C\rangle = \hat{S}_\mu\widetilde{\mathcal{T}}|C\rangle \ \forall |C\rangle \in \mathcal{H}$.

Consider for instance the $2d$ Ising spin model undergoing the paramagnetic-ferromagnetic phase transition mentioned earlier [152]. Below the critical temperature, the system spontaneously breaks the $\mathbb{Z}_2$ symmetry between spin-up and spin-down configurations. This phase transition is naturally characterized by the total magnetization $m$, which serves as an effective order parameter. The symmetry operation in this case flips the sign of all spins in a configuration, inducing a bijective mapping between configurations with opposite magnetizations. An alternative choice for order parameter could be $m^2$. While $m^2$ can distinguish between the ordered phase ($m^2 \neq 0$) and the disordered one ($m^2 \approx 0$), it cannot differentiate between the two symmetry-broken phases, making it unsuitable as an order parameter according to the criteria defined above.

Interestingly, the reduced eigenvectors $||R_{j,\mathrm{D}}^\lambda\rangle\rangle = \widetilde{\mathcal{T}}|R_{j,\mathrm{D}}^\lambda\rangle$ obtained from the spectrum of the Doob generator in the original configuration space can be readily analyzed, and they encode key information on the underlying DPT, since most of the results obtained in the previous subsections also apply in the reduced order parameter space. In particular, before the DPT happens, the reduced Doob steady state is unique, $||P_{\mathrm{ss},P_0}^\lambda\rangle\rangle = ||R_{0,\mathrm{D}}^\lambda\rangle\rangle$, see Eq. (148), while once the DPT kicks in and the symmetry is broken, degeneracy develops and

$$||P_{\mathrm{ss},P_0}^\lambda\rangle\rangle = ||R_{0,\mathrm{D}}^\lambda\rangle\rangle + \sum_{j=1}^{n-1} ||R_{j,\mathrm{D}}^\lambda\rangle\rangle \langle L_{j,\mathrm{D}}^\lambda|P_0\rangle, \tag{164}$$

see Eq. (150) for purely real eigenvalues, while something similar happens for eigenvalues with non-zero imaginary parts, see Eq. (149). Note that the brackets $\langle L_{j,\mathrm{D}}^\lambda|P_0\rangle$ do not change under $\widetilde{\mathcal{T}}$ as they are just scalar coefficients. Reduced phase probability vectors can be defined in terms of the reduced eigenvectors in the degenerate subspace, see Eq. (153),

$$||\Pi_l^\lambda\rangle\rangle = ||R_{0,\mathrm{D}}^\lambda\rangle\rangle + \sum_{j=1}^{n-1} (\phi_j)^l ||R_{j,\mathrm{D}}^\lambda\rangle\rangle, \tag{165}$$

and the reduced Doob steady state can be written as $||P_{\mathrm{ss},P_0}^\lambda\rangle\rangle = \sum_{l=0}^{n-1} w_l ||\Pi_l^\lambda\rangle\rangle$, see Eq. (158). Finally, the structural relation between the degenerate Doob eigenvectors, Eq. (161), also appears in the order-parameter space. In particular, for statistically-relevant values of $\mu$

$$\langle\langle\mu||R_{j,\mathrm{D}}^\lambda\rangle\rangle \approx \phi_j^{-\ell_\mu} \langle\langle\mu||R_{0,\mathrm{D}}^\lambda\rangle\rangle, \qquad j \in [1 .. n-1], , \tag{166}$$

where $\ell_\mu = [0 .. n-1]$ is an indicator function which connects the different values of the order parameter $\mu$ to their corresponding phase index $\ell_\mu$. Remarkably, this implies that if the steady-state distribution of $\mu$ follows a large-deviation principle, $\langle\langle\mu||R_{0,\mathrm{D}}^\lambda\rangle\rangle \asymp e^{+\tau F(\mu)}$, then the rest of gap closing reduced eigenvectors obey the following *generalized* large-deviation principle $\langle\langle\mu||R_{j,\mathrm{D}}^\lambda\rangle\rangle \asymp \phi_j^{-\ell_\mu} e^{+\tau F(\mu)}$ for the statistically-relevant values of $\mu$.

## 6.7 Dynamical criticality in the open WASEP: A spectral perspective

To end this section, we will briefly illustrate the spectral signatures of dynamical phase transitions described above for the DPT observed in the current fluctuations of the weakly assymetric

simple exclusion process (WASEP) in contact with boundary reservoirs [52, 71, 102, 109]. In section §4 we have studied this DPT for generic transport models from a hydrodynamic point of view using macroscopic fluctuation theory [2].

The WASEP [120, 121, 168] is a stochastic lattice model of particle diffusion under exclusion interactions. It consists of $N$ particles on a $1d$ lattice with $L$ sites, which can be either empty or occupied by at most one particle. The system state $C$ is defined by the occupation numbers $C = \{n_k\}_{k=1,\dots,L}$, with $n_k = 0, 1$. This state is represented as a column vector $|C\rangle = \bigotimes_{k=1}^{L}(n_k, 1 - n_k)^{\mathrm{T}}$ in a Hilbert space $\mathcal{H}$ of dimension $2^L$. Particles jump stochastically to adjacent empty sites with asymmetric rates $p_{\pm} = \frac{1}{2}\mathrm{e}^{\pm E/L}$, driven by an external field $E$ (see Fig. 6.a). The lattice ends are connected to reservoirs which inject and remove particles at rates $\alpha$, $\gamma$ (leftmost site) and $\delta$, $\beta$ (rightmost site). These rates correspond to reservoir densities $\rho_0 = \alpha/(\alpha+\gamma)$ and $\rho_1 = \delta/(\delta+\beta)$. Overall, the combined effect of the external field and the boundary gradient typically drives the system to a nonequilibrium steady state with a net particle current [1]. Macroscopically, the WASEP is described by a fluctuating hydrodynamic equation (3)-(5) with a diffusivity $D(\rho) = 1/2$ and a mobility $\sigma(\rho) = \rho(1-\rho)$, as mentioned earlier in these lecture notes. At the microscopic level, dynamics is controlled by a stochastic generator $\hat{\mathbb{W}}$ which can be written as a $2^L \times 2^L$ matrix operator acting on $\mathcal{H}$, see below.

Considering now the statistics of an ensemble of trajectories conditioned on a given time-averaged current $q$ during a long time $\tau$, we have already seen at the beginning of this section that the current statistics is fully characterized by the dynamical free energy $\theta(\lambda)$, the Legendre transform of the current LDF $G(q)$. This dynamical free energy corresponds to the eigenvalue with largest real part of a biased or tilted generator $\hat{\mathbb{W}}^{\lambda}$, which for the $1d$ open WASEP reads

$$
\begin{aligned}
\hat{\mathbb{W}}^{\lambda} &= \sum_{k=1}^{L-1}\Big[p_+\big(\mathrm{e}^{\lambda/(L-1)}\hat{\sigma}_{k+1}^+\hat{\sigma}_k^- - \hat{n}_k(\hat{\mathbb{1}} - \hat{n}_{k+1})\big) + p_-\big(\mathrm{e}^{-\lambda/(L-1)}\hat{\sigma}_k^+\hat{\sigma}_{k+1}^- - \hat{n}_{k+1}(\hat{\mathbb{1}} - \hat{n}_k)\big)\Big] \\
&\quad + \alpha[\hat{\sigma}_1^+ - (\hat{\mathbb{1}} - \hat{n}_1)] + \gamma[\hat{\sigma}_1^- - \hat{n}_1] + \delta[\hat{\sigma}_L^+ - (\hat{\mathbb{1}} - \hat{n}_L)] + \beta[\hat{\sigma}_L^- - \hat{n}_L]. \qquad (167)
\end{aligned}
$$

Here $\hat{\sigma}_k^{\pm} = (\hat{\sigma}_k^x \pm i\hat{\sigma}_k^y)/2$ are the creation and annihilation operators, with $\hat{\sigma}_k^{x,y}$ the standard $x, y$-Pauli matrices, while $\hat{n}_k = \hat{\sigma}_k^+\hat{\sigma}_k^-$ and $\hat{\mathbb{1}}_k$ are the occupation and identity operators acting on site $k$, respectively. The first row in the rhs of the previous equation describes (biased) particle jumps to right/left neighboring empty sites with rates $p_{\pm}$, while the second row corresponds to the boundary injection and removal of particles. Note that the original Markov generator of the dynamics is just $\hat{\mathbb{W}} = \hat{\mathbb{W}}^{\lambda=0}$, while $\hat{\mathbb{W}}^{\lambda\neq 0}$ does not conserve probability.

Interestingly, when the boundary rates satisfy $\alpha = \beta$ and $\gamma = \delta$, resulting in $\rho_1 = 1 - \rho_0$, the WASEP dynamics is invariant under a particle-hole (PH) transformation, represented by a unitary operator $\hat{S}_{\mathrm{PH}}$ which thus commutes with the generator of the dynamics, $[\hat{S}_{\mathrm{PH}}, \hat{\mathbb{W}}] = 0$ [102, 109]. This transformation involves flipping the occupation of each site, $n_k \to 1 - n_k$, while reversing the spatial order, $k \to L - k + 1$; see Ref. [102] for an explicit expresion for the $\hat{S}_{\mathrm{PH}}$ operator. Invariance againts $\hat{S}_{\mathrm{PH}}$ is a $\mathbb{Z}_2$ symmetry, since $(\hat{S}_{\mathrm{PH}})^2 = \hat{\mathbb{1}}$. At the microscopic level, the empirical current $Q$ is defined as the net number of rightward jumps minus leftward jumps per bond (in the bulk) during a trajectory $\omega_{\tau}$ of duration $\tau$. This observable remains invariant under the PH transformation, $Q(\mathcal{S}_{\mathrm{PH}}\omega\tau) = Q(\omega_{\tau})$, because the changes in occupation and spatial inversion result in a double sign change in the flux, leaving the total current unchanged. Consequently, the symmetry under $\hat{S}_{\mathrm{PH}}$ is inherited by the biased generator $\hat{\mathbb{W}}^{\lambda}$ and the Doob driven process $\hat{\mathbb{W}}_{\mathrm{D}}^{\lambda}$ governing the fluctuations of the current, so the results of previous sections on the spectral fingerprints of spontaneous symmetry breaking apply for the DPT observed in the open WASEP.

We discuss now a particular example with $E = 4 > E_c$ and $\alpha = \beta = \gamma = \delta = 0.5$, corresponding to equal densities $\rho_0 = \rho_1 = 0.5$, though the spectral results shown also apply for arbitrary strong drivings and (PH-symmetric) boundary gradients. As explained in previous subsections, in a $\mathbb{Z}_2$-symmetry breaking DPT we expect an emergent degeneracy for the

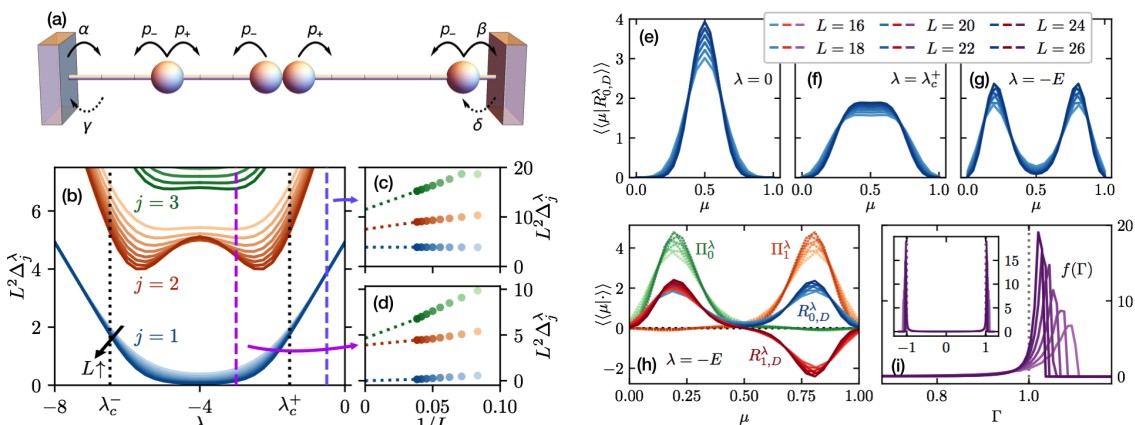

Figure 6: **A spectral view on dynamical criticality for the** $1d$ **open WASEP.**
(a) Sketch of the $1d$ boundary-driven WASEP, where $N$ particles in a lattice of $L$
sites jump randomly to empty neighboring sites with asymmetric rates $p_\pm$. (b)
Scaled spectral gaps, $L^2\Delta_j^\lambda$, as a function of $\lambda$ for $j = 1, 2, 3$ and different $L =$
$12, 14, 16, 18, 20, 22, 24, 26$. Colors codify different valuea of $j$, with darker color
meaning larger $L$. Panels (c) and (d) show $L^2\Delta_j^\lambda$ as a function of $1/L$ for (c)
$\lambda = -0.5 > \lambda_c^+$ and (d) $\lambda = -3.5 < \lambda_c^+$. Panels (e)-(g) show the structure of
$\langle\langle\mu||R_{0,\mathrm{D}}^\lambda\rangle\rangle$ for different $\lambda$ and varying $L$. Panel (h) compares the structure of the
degenerate reduced Doob eigenvertors $\langle\langle\mu||R_{j,\mathrm{D}}^\lambda\rangle\rangle$ with $j = 0, 1$, in the symmetry-
broken phase ($\lambda = -4$, full lines). The structure of the reduced phase probability
vectors $\langle\langle\mu||\Pi_l^\lambda\rangle\rangle$, $l = 0, 1$, is also displayed (dotted lines). Panel (i) shows the his-
togram for $\Gamma = \langle C|R_{1,\mathrm{D}}^\lambda\rangle/\langle C|R_{0,\mathrm{D}}^\lambda\rangle$ obtained for a large set of configurations $|C\rangle$ sam-
pled from the Doob steady-state distribution for $\lambda = -4$ and increasing $L$. Image
reproduced from [102] with permission from the American Physical Society.

first two leading eigenvectors of the Doob driven process. We hence analyze the scaled spec-
tral gaps $L^2\Delta_j^\lambda$ [1, 102, 109] for $j = 1, 2, 3, \ldots$ obtained by numerical diagonalization of the
Doob generator $\hat{\mathbb{W}}_\mathrm{D}^\lambda$ for the boundary-driven WASEP, see Figs. 6.b-d. We observe that, while
$L^2\Delta_{2,3}^\lambda(L) > 0$ for all $\lambda$ and $L$ (so their associated eigenvectors do not contribute to the degen-
erate Doob stationary subspace), $L^2\Delta_1^\lambda(L)$ vanishes as $L$ increases for $\lambda_c^- < \lambda < \lambda_c^+$, with $\lambda_c^\pm$
some critical $\lambda$-values directly related to the critical current $q_c$ in Eq. (71) [102, 109], signal-
ing that a $\mathbb{Z}_2$-symmetry breaking DPT has already kicked in this $\lambda$-regime. These two different
tendencies are more clearly appreciated in Figs. 6.c-d, which display the decay of $L^2\Delta_j^\lambda$ as a
function of $1/L$ for $j = 1, 2, 3$ and two different values of $\lambda$. In this way, outside the critical
region ($\lambda > \lambda_c^+$ or $\lambda < \lambda_c^-$), the Doob steady state is unique and preserves the PH symmetry of
the original dynamics, $|P_{\mathrm{ss},P_0}^\lambda\rangle = |R_{0,\mathrm{D}}^\lambda\rangle$. On the other hand, for $\lambda_c^- < \lambda < \lambda_c^+$, the spectral gap
$L^2\Delta_1^\lambda(L)$ vanishes as $L \to \infty$ and $|R_{1,\mathrm{D}}^\lambda\rangle$ enters the degenerate Doob stationary subspace. In
this way the Doob stationary state in the macroscopic limit is $|P_{\mathrm{ss},P_0}^\lambda\rangle = |R_{0,\mathrm{D}}^\lambda\rangle + |R_{1,\mathrm{D}}^\lambda\rangle\langle L_{1,\mathrm{D}}^\lambda|P_0\rangle$,
breaking spontaneously the PH-symmetry of the original dynamics.

To analyze the spectral structure of the Doob steady state, we can turn to the reduced
order parameter space and introduce the mean occupation of the lattice $\mu = L^{-1}\sum_{k=1}^L n_k$ as
order parameter. This allows us to extract the relevant macroscopic information contained
in the leading Doob eigenvectors by grouping together in equivalence classes configurations
with the same total mass. Figs. 6.e-g show the order parameter structure of the leading re-
duced Doob eigenvector $||R_{0,\mathrm{D}}^\lambda\rangle\rangle$ before (e) and after (g) the DPT, as well as near the criti-
cal point (f), for different system sizes. Interestingly, before the DPT happens $\langle\langle\mu||R_{0,\mathrm{D}}^\lambda\rangle$ is

unimodal in $\mu$, see Fig. 6.e, as there is only a single phase contributing to the Doob steady state, $||R_{0,D}^\lambda\rangle\rangle = ||\Pi_0^\lambda\rangle\rangle$, which hence preserves the $\mathbb{Z}_2$ symmetry of the original dynamics. Indeed, $\langle\langle\mu||R_{0,D}^\lambda\rangle\rangle$ in this phase is just the steady state probability distribution for $\mu$. Near the critical point $\lambda = \lambda_c^+$ (Fig. 6.f) $\langle\langle\mu||R_{0,D}^\lambda\rangle\rangle$ is still unimodal but becomes flat around the peak, while deep into the symmetry-broken region the distribution $\langle\langle\mu||R_{0,D}^\lambda\rangle\rangle$ becomes bimodal, see Fig. 6.g, with two symmetric peaks. Note that $\langle\langle\mu||R_{0,D}^\lambda\rangle\rangle$ is still invariant under the symmetry operator, i.e. $\langle\langle\mu||R_{0,D}^\lambda\rangle\rangle = \langle\langle\mu||\hat{S}_\mu||R_{0,D}^\lambda\rangle\rangle$ so it has a symmetry eigenvalue $\phi_0 = 1$, but the degenerate subspace also includes now $||R_{1,D}^\lambda\rangle\rangle$ in the $L \to \infty$ limit. Fig. 6.h compares the $\mu$-structure of both $\langle\langle\mu||R_{0,D}^\lambda\rangle\rangle$ and $\langle\langle\mu||R_{1,D}^\lambda\rangle\rangle$, showing that $\langle\langle\mu||R_{1,D}^\lambda\rangle\rangle$ is antisymmetric as expected, $\hat{S}_\mu||R_{1,D}^\lambda\rangle\rangle = -||R_{1,D}^\lambda\rangle\rangle$, so $\phi_1 = e^{i\pi} = -1$. Reduced phase probability vectors now follow from Eq. (165) particularized to this $\mathbb{Z}_2$-symmetry breakin DPT, i.e. $||\Pi_l^\lambda\rangle\rangle = ||R_{0,D}^\lambda\rangle\rangle + (-1)^l||R_{1,D}^\lambda\rangle\rangle$, with $l = 0, 1$, and they define the two degenerate reduced Doob steady states in each of the symmetry sectors, see dotted lines in Fig. 6.h. These distributions correspond to each of the symmetry-broken density profiles described in section §4 using MFT, see also Fig. 2.c. The generic reduced Doob steady state $||P_{ss,P_0}^\lambda\rangle\rangle$ in the limit $L \to \infty$ will be a weighted superposition of these two degenerate branches,

$$||P_{ss,P_0}^\lambda\rangle\rangle = w_0||\Pi_0^\lambda\rangle\rangle + w_1||\Pi_1^\lambda\rangle\rangle, \qquad \text{with weights } w_l = \frac{1}{2}\left[1 + (-1)^l\langle L_{1,D}^\lambda|P_0\rangle\right].$$

This illustrates the phase selection mechanism via initial state preparation discussed in §6.4.

Finally, we may test in this example the spectral relation (161) between components of the degenerate-subspace eigenvectors associated with statistically-relevant configurations. For the WASEP case, this relation is $\langle C|R_{1,D}^\lambda\rangle \approx (-1)^{-\ell_C}\langle C|R_{0,D}^\lambda\rangle$, with $\ell_C = 0, 1$ depending whether configuration $|C\rangle$ belongs to the high-$\mu$ or low-$\mu$ symmetry basin, respectively. To validate this prediction, we sampled a large number of statistically-relevant configurations in the Doob steady state and analyzed the histogram $f(\Gamma)$ of the ratio $\Gamma(C) \equiv \langle C|R_{1,D}^\lambda\rangle/\langle C|R_{0,D}^\lambda\rangle$, see Fig. 6.i. As predicted, $f(\Gamma)$ exhibits sharp peaks around $(\phi_1)^0 = 1$ and $\phi_1 = -1$ and becomes increasingly concentrated around these values as $L$ grows. This confirms that the component structure of $|R_{1,D}^\lambda\rangle$ is determined by that of $|R_{0,D}^\lambda\rangle$, depending on the symmetry basin $\ell_C$ of each configuration $|C\rangle$, supporting also *a posteriori* the assumption that statistically-relevant configurations can be divided into disjoint symmetry classes.

The general spectral results of sections §6.1-§6.6 have been also corroborated in a number of distinct DPTs in other paradigmatic many-body systems, as e.g. in the study of energy fluctuations in $n$-state Potts models for spin dynamics [169], $n = 3, 4$, where a $\mathbb{Z}_n$-symmetry breaking DPT has been observed and characterized [93, 102]. The above spectral results have been also confirmed with high accuracy in the time-translation symmetry-breaking DPT in current statistics observed for the 1$d$ periodic WASEP, and studied at the macroscopic level with MFT in section §5. This case is even more compelling, as the broken symmetry is asymptotically continuous in the thermodynamic limit, with an emergent rotating particle condensate as key feature. We refer the interested reader to Ref. [102] for more details on this issue. We will explore in the next section some of the spectral fingerprints of this DPT, but with the aim of understanding the underlying physical mechanism. This knowledge will allow us to engineer programmable time-crystal phases of nonequilibrium matter [101, 103].

# 7 The rare event route to programmable time crystals

We have seen repeatedly in these lecture notes that many-body systems may exhibit spontaneous symmetry-breaking phenomena in their fluctuation behavior. Sometimes, when rare

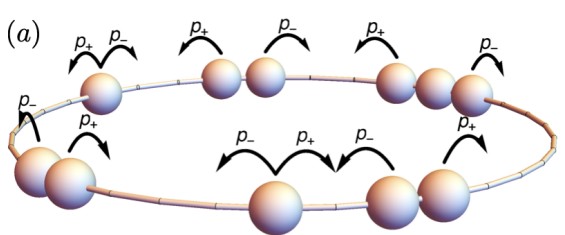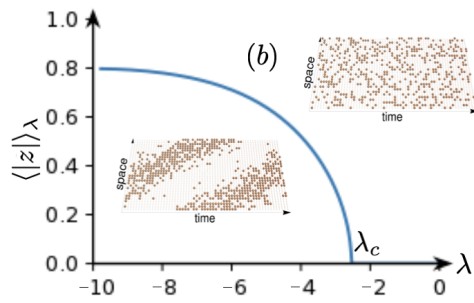

Figure 7: **Dynamical phase transition for the** $1d$ **periodic WASEP.** (a) Sketch of the WASEP model in a $1d$ periodic lattice, with stochastic particle jumps to neighboring empty sites at rates $p_\pm$. The total number of particles is conserved. (b) Magnitude of the packing order parameter $\langle|z|\rangle_\lambda$ as a function of the bias parameter $\lambda$. The inset shows typical spacetime trajectories for current fluctuations above (top) and below (bottom) the critical point for this DPT, located at $\lambda_c$. Image adapted from [102] with permission from the American Physical Society.

fluctuations of some time-extensive observable such as the empirical current come about, the system may exhibit a dynamical phase transition which manifests as a drastic, singular change in the optimal trajectory responsible of a given fluctuation, including typically changes in its symmetry properties. A main example of this behavior discussed in previous sections, particularly in §5, is the $1d$ WASEP model of particle diffusion on a periodic (ring) lattice, see Figs. 7 and 5.b. In order to sustain a fluctuation of the time-integrated current well below the steady-state average current $\langle q \rangle = \bar{\sigma}E$, this many-body system develops a jammed density wave or rotating particle condensate that hinders particle motion due to exclusion interactions, thus facilitating the observed current fluctuation [29, 43, 101–103]. This happens for flux fluctuations below a critical current. Indeed, as computed in section §5, this DPT occurs for external fields $|E| > E_c = \pi/\sqrt{\bar{\rho}(1-\bar{\rho})}$ and currents $|q| \leq q_c = \bar{\rho}(1-\bar{\rho})\sqrt{E^2 - E_c^2}$, where we have already used that $D(\rho) = 1/2$ and $\sigma(\rho) = \rho(1-\rho)$ for WASEP. This current regime corresponds to bias parameter in the range $\lambda_c^- < \lambda < \lambda_c^+$, where $\lambda_c^\pm = -E \pm \sqrt{E^2 - E_c^2}$ [29, 43, 44, 101, 102]. The emergence of a traveling particle condensate at the DPT moving at constant velocity, see bottom inset in Fig. 7.b, breaks spontaneously the original spatiotemporal translation symmetry of the $1d$ periodic WASEP. This is a key feature of the recently discovered time-crystal phase of matter [97–99, 101, 154–156, 170].

## 7.1 A spectral view on time-translation symmetry breaking

We now explore the spectral fingerprints of this second-order DPT and the associated time-crystal phases at the fluctuating level. Note that the (broken) translation symmetry is captured by a unitary operator $\hat{S}_T$ (see [102] for an explicit expression; this section is adapted from Ref. [102] with permission from the American Physical Society), which generates the cyclic group $\mathbb{Z}_L$ and commutes with the stochastic generator of the dynamics. The Legendre transform $\theta(\lambda)$ of the current LDF $G(q)$, corresponding to the scaled cumulant generating function for the empirical current distribution, follows as the eigenvalue with largest real part of the biased generator $\hat{\mathbb{W}}^\lambda$, as described in section §6.1. For the $1d$ WASEP model on a periodic lattice, this biased generator reads

$$\hat{\mathbb{W}}^\lambda = \sum_{k=1}^{L} \Big[ p_+\big(e^{+\lambda/L}\hat{\sigma}_{k+1}^+\hat{\sigma}_k^- - \hat{n}_k(\hat{\mathbb{1}} - \hat{n}_{k+1})\big) + p_-\big(e^{-\lambda/L}\hat{\sigma}_k^+\hat{\sigma}_{k+1}^- - \hat{n}_{k+1}(\hat{\mathbb{1}} - \hat{n}_k)\big) \Big].$$

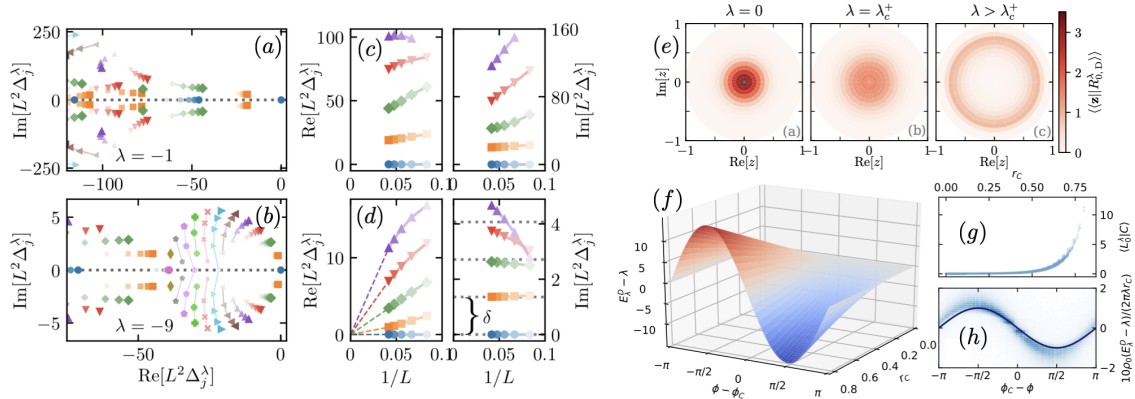

Figure 8: **Spectral signatures of the time-crystal phase for the** $1d$ **periodic WASEP and packing field.** (a)-(d) Structure in the complex plane of the leading scaled spectral gaps $L^2\Delta_j^\lambda$ with $\bar\rho = 1/3$, $E = 10 > E_c$, and lattice sizes $L = 9, 12, 15, 18, 21, 24$ for (a) the homogeneous phase at $\lambda = -1$ and (b) the condensate phase at $\lambda = -9$. Larger marker size corresponds to larger $L$, illustrating the spectrum's evolution as $L$ increases. Panels (c) and (d) analyze the finite-size scaling behavior of the real and imaginary parts of the leading scaled spectral gaps for the homogeneous (c) and condensate (d) phases. In the condensate phase, the real parts decay to zero as a power law of $1/L$, while the imaginary parts form a distinct band structure with a constant frequency spacing $\delta$, which is proportional to the condensate velocity [101, 102]. (e) Structure of the reduced leading eigenvector $\langle\langle\mathbf{z}||R_{0,D}^\lambda\rangle\rangle$ in the complex $\mathbf{z}$-plane for different values of $\lambda$ across the DPT for $L = 24$ and the same values of $\bar\rho = 1/3$ and $E = 10$. Note the change from unimodal $\langle\langle\mathbf{z}||R_{0,D}^\lambda\rangle\rangle$ peaked around $|\mathbf{z}| \approx 0$ for $\lambda > \lambda_c^+$ (leftmost panel) to the inverted Mexican-hat structure with a steep ridge around $|\mathbf{z}| \approx 0.7$ for $\lambda_c^- < \lambda < \lambda_c^+$ (rightmost panel). (f) Packing field as a function of $r_C \equiv |\mathbf{z}_C|$ and the angular distance from the center-of-mass position, $\phi - \phi_C$, for $\bar\rho = 1/3$ and $\lambda = -9$. Panel (g) shows $\langle L_0^\lambda|C\rangle$ vs $r_C$ for a large sample of configurations in the condensate phase ($\lambda = -9$) and the same parameters, while panel (h) displays the angular dependence of the Doob's smart field relative to the center-of-mass angular location for this sample, along with the $\sin(\phi - \phi_C)$ prediction (line). Image adapted from [101, 102] with permission from the American Physical Society.

This should be compared with the tilted generator of the boundary-driven WASEP, see Eq. (167). The spectrum of $\hat{\mathbb{W}}^\lambda$, or equivalently the spectrum of the associated Doob generator $\hat{\mathbb{W}}_D^\lambda$, see Eq. (145), encodes all the information on this DPT. Figs. 8.a-b shows the complex structure of the scaled spectral gaps $L^2\Delta_j^\lambda$ associated to $\hat{\mathbb{W}}_D^\lambda$ as obtained numerically for $L = 24$, $\bar\rho = N/L = 1/3$, $E = 10 > E_c$, and two biasing parameter values: one subcritical (Fig.8.a) and one within the DPT regime (Fig.8.b). The spectral structure in the complex plane undergoes a drastic transformation between the two phases. Specifically, the spectrum remains gapped (i.e., $\text{Re}[L^2\Delta_j^\lambda] < 0$ for $j > 0$) for $\lambda < \lambda_c^-$ or $\lambda > \lambda_c^+$. This is hinted at the $L$-evolution of the leading eigenvalues in Fig.8.a, and confirmed in their non-decaying evolution as a function of $1/L$ in Fig.8.c. However, in the condensate phase ($\lambda_c^- < \lambda < \lambda_c^+$), the real part of a macroscopic (i.e. proportional to $L$) fraction of eigenvalues develops a vanishing gap as $L \to \infty$, see Fig.8.b, decaying to zero as a power law with $1/L$ as confirmed in Fig. 8.d. Additionally, the imaginary parts of the gap-closing eigenvalues exhibit a distinct band structure with constant frequency spacing $\delta$, see dashed horizontal lines in Fig. 8.d, which is related with the velocity $v$ of the moving condensate, $\delta = 2\pi v/L$. These are hallmark spectral features of a time-crystal phase [98, 99, 155, 156, 170]. The emergence of an $\mathcal{O}(L)$-fold degeneracy for $\lambda_c^- < \lambda < \lambda_c^+$ as

$L$ increases further signals the presence of competing symmetry-broken states, stemming from the condensate's invariance under integer lattice translations. Thus, this DPT at the fluctuating level exhibits clear signatures of a time-crystal phase, offering a pathway to engineer such nonequilibrium phases of matter in driven diffusive fluids.

Given the above spectral properties for $\hat{\mathbb{W}}_{\mathrm{D}}^{\lambda}$, we expect a unique Doob steady state in the gapped regime ($\lambda > \lambda_c^+$ or $\lambda < \lambda_c^-$), captured by the leading Doob eigenvector $|P_{\mathrm{ss},P_0}^{\lambda}\rangle = |R_{0,\mathrm{D}}^{\lambda}\rangle$. This steady state is invariant under the symmetry operator $\hat{S}_{\mathrm{T}}$. However, in the gapless regime ($\lambda_c^- < \lambda < \lambda_c^+$), the Doob *stationary* subspace becomes $\mathcal{O}(L)$-fold degenerate, and the resulting Doob *steady state* is inherently time-dependent. Taking into account the band structure observed in the imaginary parts of the gap-closing eigenvalues, see Fig. 8.d, with constant spacing $\delta$, this time-dependent *steady state* can be shown to be asymptotically [102]

$$|P_{\mathrm{ss},P_0}^{\lambda}\rangle(t) = |R_{0,\mathrm{D}}^{\lambda}\rangle + 2 \sum_{\substack{j=2 \\ j\,\mathrm{even}}}^{L-1} \mathrm{Re}\left[ e^{+it\frac{j\delta}{2}} |R_{j,\mathrm{D}}^{\lambda}\rangle\langle L_{j,\mathrm{D}}^{\lambda}|P_0\rangle \right] = \sum_{l=0}^{L-1} w_l(t)|\Pi_l^{\lambda}\rangle, \qquad (168)$$

with phase probability vectors $|\Pi_l^{\lambda}\rangle$ and time-dependent weights $w_l(t)$ given by

$$|\Pi_l^{\lambda}\rangle = |R_{0,\mathrm{D}}^{\lambda}\rangle + 2 \sum_{\substack{j=2 \\ j\,\mathrm{even}}}^{L-1} \mathrm{Re}\left[ e^{+i\frac{\pi l}{L}j}|R_{j,\mathrm{D}}^{\lambda}\rangle \right], \qquad w_l(t) = \frac{1}{L} + \frac{2}{L} \sum_{\substack{j=2 \\ j\,\mathrm{even}}}^{L-1} \mathrm{Re}\left[ e^{+i(\frac{t\delta}{2} - \frac{\pi l}{L})j}\langle L_{j,\mathrm{D}}^{\lambda}|P_0\rangle \right].$$

From this structure it can be then shown that, in the quasi-degenerate condensate phase $\lambda_c^- < \lambda < \lambda_c^+$, the time-dependent *steady state* (168) is such that $\hat{S}_{\mathrm{T}}|P_{\mathrm{ss},P_0}^{\lambda}\rangle(t) = |P_{\mathrm{ss},P_0}^{\lambda}\rangle(t + \frac{2\pi}{L\delta})$. This demonstrates that indeed the $\hat{S}_{\mathrm{T}}$ symmetry is spontaneously broken in the condensate phase, $\hat{S}_{\mathrm{T}}|P_{\mathrm{ss},P_0}^{\lambda}\rangle(t) \neq |P_{\mathrm{ss},P_0}^{\lambda}\rangle(t)$, but also that spatial translation and time evolution are two sides of the same coin in this regime, leading to a time-periodic motion of period $2\pi/\delta$ or equivalently a density wave of velocity $v = L\delta/2\pi$.

Following the ideas of section §6.6, a suitable order parameter to characterize this DPT is the amount of *packing* of particles in the $1d$ ring. For a configuration $C = \{n_k\}_{k=1,\dots,L}$ with $N = \sum_{k=1}^{L} n_k$ particles, the *packing order parameter* $\mathbf{z}_C \in \mathbb{C}$ is defined as

$$\mathbf{z}_C = \frac{1}{N} \sum_{k=1}^{L} n_k\, e^{i2\pi k/L} = |\mathbf{z}_C| e^{i\phi_C}. \qquad (169)$$

This parameter measures the position of the particles' center of mass in the two-dimensional plane embedding the $1d$ ring. The magnitude $|\mathbf{z}_C|$ is near zero for any homogeneous particle distribution but increases significantly for condensed configurations, while the phase $\phi_C \in [0, 2\pi)$ indicates the angular position of the condensate's center of mass in the ring. Thus, we expect the average $\langle|\mathbf{z}|\rangle_{\lambda}$ to grow from zero when the condensate emerges at the DPT, as confirmed in Fig. 7.b, which shows the average magnitude of the packing order parameter as a function of the bias parameter $\lambda$ across the DPT as obtained from MFT [101, 102]. Similar behavior is observed in cloning Monte Carlo simulations [31, 44, 104, 105, 108, 161, 162] of this DPT at the microscopic particle level [103].

The Doob stationary subspace is more effectively analyzed in the reduced Hilbert space associated with the packing order parameter $\mathbf{z}$, see section §6.6. Before the DPT, in the gapped regime for $\lambda > \lambda_c^+$ or $\lambda < \lambda_c^-$, the distribution $\langle\langle\mathbf{z}\|R_{0,\mathrm{D}}^{\lambda}\rangle\rangle$ of the leading reduced Doob eigenvector in the complex $\mathbf{z}$-plane is unimodal and peaks around $|\mathbf{z}| \approx 0$, see left panel in Fig. 8.e, reflecting the absence of order in this symmetry-preserving phase. As $\lambda$ approaches $\lambda_c^+$, $\langle\langle\mathbf{z}\|R_{0,\mathrm{D}}^{\lambda}\rangle\rangle$ flattens and spreads over the unit complex circle, as shown in the central panel of Fig. 8.e. Deep within the condensate regime ($\lambda_c^- < \lambda < \lambda_c^+$), the distribution develops

an inverted Mexican-hat shape, with a steep ridge around $|\mathbf{z}| \approx 0.7$ and a uniform angular distribution, see right panel in Fig. 8.e, indicating that typical configurations contributing to $|R_{0,D}^\lambda\rangle$ correspond to symmetry-broken condensate states ($|\mathbf{z}| \neq 0$), which are localized but have a homogeneous angular distribution for their center of mass. $||R_{0,D}^\lambda\rangle\rangle$ remains invariant under the (reduced) symmetry operator $\hat{S}_\mathbf{z}$, representing a rotation of $2\pi/L$ radians in the complex $\mathbf{z}$-plane. Moreover, the subleading reduced Doob eigenvectors $||R_{j,D}^\lambda\rangle\rangle$ in the (quasi-)degenerate subspace ($j > 0$, not shown; see [102]) exhibit a multipolar structure in the $\mathbf{z}$-plane, cooperating to break the symmetry by localizing the condensate at a specific position in the lattice [102]. These results show how symmetry imposes a specific spectral structure across the DPT, explaining the microscopic origin of the structure of the optimal path to a current fluctuation.

## 7.2 The Doob's smart field and its underlying physics

To understand the physical origin of the observed time-crystal phase in the current fluctuations of the $1d$ periodic WASEP model, we now express Doob's dynamics $\hat{\mathbb{W}}_D^\lambda$ in terms of the original WASEP dynamics $\mathbb{W}$ modified by some *smart* driving field $E_\lambda^D$, with the aim of analyzing the structure of this field. Specifically, in terms of matrix components, we define this smart driving field from the following identity

$$\langle C_i | \mathbb{W}_D^\lambda | C_j \rangle = \langle C_i | \mathbb{W} | C_j \rangle e^{q_{C_j \to C_i}(E_\lambda^D)_{ij}/L}, \qquad (170)$$

where $q_{C_j \to C_i} = \pm 1$ is the direction of the particle jump in the transition $C_j \to C_i$. Using now the definition of $\hat{\mathbb{W}}_D^\lambda$ in Eq. (145), we find

$$(E_\lambda^D)_{ij} = \lambda + q_{C_j \to C_i} L \ln\left(\frac{\langle L_0^\lambda | C_i \rangle}{\langle L_0^\lambda | C_j \rangle}\right). \qquad (171)$$

$E_\lambda^D$ is nothing but the external driving field needed to make typical a rare event of bias parameter $\lambda$, and it will be in general a complex, non-local function of the high-dimensional configurations.

In order to analyze the physics underlying the Doob's smart field, we now scrutinize its dependence on the magnitude of the packing order parameter $r_C \equiv |\mathbf{z}_C|$ of the configurations. Fig. 8.g shows the projections $\langle L_0^\lambda | C \rangle$ entering the defition (171) of $E_\lambda^D$ plotted against $r_C$ for a large sample of microscopic configurations $|C\rangle$, as obtained numerically for $L = 24$, $\bar{\rho} = 1/3$, and $\lambda = -9$ (in the condensate phase). Interestingly, this plot indicates that $\langle L_0^\lambda | C \rangle$ is a sole function of $r_C$ to a high degree of accuracy, i.e. $\langle L_0^\lambda | C \rangle \simeq f_{\lambda,L}(r_C)$, where $f_{\lambda,L}(r)$ is some unknown function of the packing parameter that may depend on $\lambda$ and $L$. This is a huge simplification, as $|C\rangle$ is a high-dimensional object ($2^L$ for WASEP) while $r_C$ is just a scalar quantity. This also implies that the Doob's smart field $(E_\lambda^D)_{ij}$ is primarily determined by the packing parameters of configurations $C_i$ and $C_j$. Furthermore, since elementary transitions $C_j \to C_i$ between configurations involve only a local particle jump, the resulting change in the packing parameter is perturbatively small for sufficiently large $L$. Specifically, taking into account the definition of $|\mathbf{z}_C|$ in Eq. (169), if $C_k'$ is the configuration obtained from $C$ after a particle jump at site $k \in [1, L]$, then

$$r_{C_k'} \simeq r_C + \frac{2\pi}{\bar{\rho} L^2} q_{C \to C_k'} \sin(\phi_C - \phi_k) \equiv r_C + \delta r_C, \qquad (172)$$

with $\phi_k \equiv 2\pi k/L$. The second term in the rhs is perturbatively small for large system sizes ($\delta r_C \sim L^{-2}$), and hence we can Taylor-expand to first order $f_{\lambda,L}(r_{C_k'}) \approx f_{\lambda,L}(r_C) + \delta r_C \, f'_{\lambda,L}(r_C)$,

the prime $'$ meaning derivative with respect to the argument. Therefore, using that $\ln(1+\delta) \approx \delta$ to first order, the Doob's smart field (171) for this transition is simply

$$(E_\lambda^{\mathrm{D}})_{C_k', C} \simeq \lambda + \frac{2\pi}{\bar\rho L} \, g_{\lambda, L}(r_C) \sin(\phi_C - \phi_k), \qquad (173)$$

where we have defined $g_{\lambda, L}(r) \equiv f_{\lambda, L}'(r)/f_{\lambda, L}(r)$. The function $g_{\lambda, L}(r)$ has been empirically found to depend linearly as $g_{\lambda, L}(r) \approx -\lambda L r/10$ near the critical point $\lambda_c^+$ [101], as corroborated in Fig. 8.h which shows $10\bar\rho[(E_\lambda^{\mathrm{D}})_{C_k', C} - \lambda]/(2\pi\lambda r_C)$ for a large sample of connected configurations $C \to C_k'$ as a function of $\phi_C - \phi_k$. This excellent agreement thus confirms that the Doob's smart field can be written quite generically as

$$(E_\lambda^{\mathrm{D}})_{C_k', C} \simeq \lambda + \frac{2\pi}{10\bar\rho} \, \lambda r_C \, \sin(\phi_k - \phi_C). \qquad (174)$$

Hence $(E_\lambda^{\mathrm{D}} - \lambda) \propto \lambda r_C \, \sin(\phi_k - \phi_C)$ functions as a *packing field* for a given configuration $C$, driving forward particles lagging behind the center of mass at $\phi_C$ while slowing down those moving ahead (see Fig. 8.f), with an amplitude that is proportional to both the bias parameter $\lambda$ and the packing order parameter $r_C$. The Doob's smart field thus induces a nonlinear feedback mechanism which enhances naturally-occurring fluctuations in the packing parameter $r_C$, counteracting the diffusive tendency to smooth particle clumps, and ultimately giving rise to a time-crystal phase for $\lambda_c^- < \lambda < \lambda_c^+$. Note that effective potentials for atypical fluctuations similar to the packing field here described have been found in other driven systems [159, 171].

## 7.3 The packing field mechanism

Inspired by the results of the previous analysis, we now distill the key features of the Doob's smart field and introduce a new model, the time-crystal lattice gas (TCLG), where these ideas can be better understood. The TCLG is a variant of the $1d$ WASEP where a particle at site $k$ hops stochastically under an inhomogeneous and configuration-dependent field $E_k(C)$. Note that, mathematically, the action of the packing term in Doob's smart field, $(E_\lambda^{\mathrm{D}} - \lambda) \propto \lambda r_C \, \sin(\phi_k - \phi_C)$, can be seen as a controlled excitation of the first Fourier mode of the particle configuration around the instantaneous center of mass position at $\phi_C$ [101]. A natural generalization of this idea then consists in exciting higher, $m$th-order Fourier modes (with $m > 1$). We will show in this and subsequent sections how this excitation mechanism opens the door to fully programmable continuous time-crystal phases in driven diffusive fluids, characterized by an arbitrary number $m$ of rotating condensates [103].

In particular, and motivated by the previous discussion, we now choose the configuration-dependent field to be $E_k(C) = \epsilon + \eta \, \mathcal{E}_k^{(m)}(C)$. Here $\epsilon$ is a constant field driving particles in a homogeneous way along a given direction, and $\eta$ is a coupling constant to a generalized, $m$th-order packing field

$$\mathcal{E}_k^{(m)}(C) = |\mathbf{z}_m(C)| \sin\left(\phi_m(C) - \frac{2\pi m k}{L}\right), \qquad (175)$$

with $\mathbf{z}_m(C)$ the complex $m$th-order packing order parameter, similar to the Kuramoto-Daido parameter used in synchronization literature [172–176],

$$\mathbf{z}_m(C) = \frac{1}{N} \sum_{k=1}^{L} n_k e^{i2\pi m k/L} \equiv |\mathbf{z}_m(C)| e^{i\phi_m(C)}, \qquad (176)$$

with magnitude $|\mathbf{z}_m(C)|$ and argument $\phi_m(C)$. The constant driving $\epsilon$ gives rise to a net particle current in the desired direction, and controls the velocity of the resulting particle

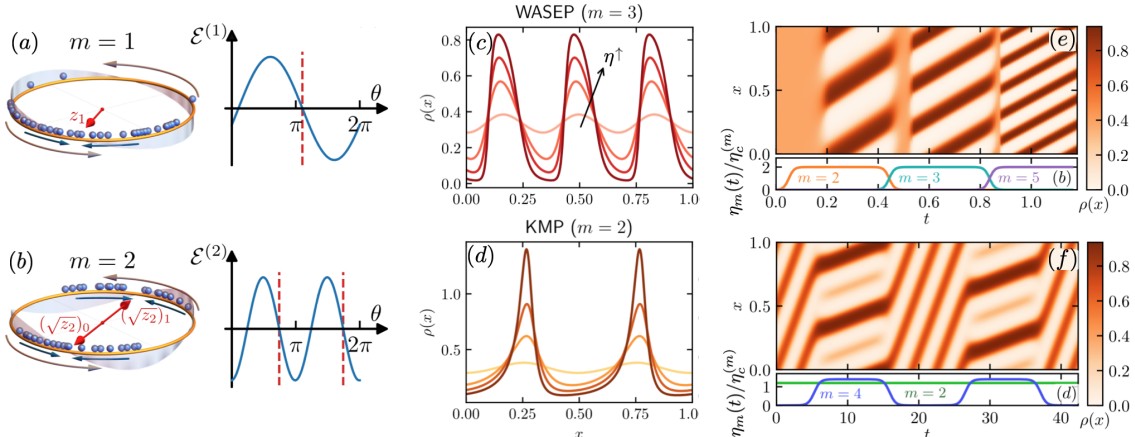

Figure 9: **Programmable time crystals from higher-order packing fields.** (a)-(b) Coupling a many-body particle system to an $m$th-order packing field $\mathcal{E}^{(m)}(C)$ with strength $\eta$ beyond a critical threshold can trigger an instability to a time-crystal phase characterized by the formation of $m$ rotating particle condensates. The magnitude $|\mathbf{z}_m|$ of the complex packing order parameter reflects the degree of particle packing around $m$ emergent localization centers, located at the complex arguments of $(\sqrt[m]{\mathbf{z}_m})_j$, with $j \in [0, m-1]$, represented by red arrows inside the ring in (a) and (b). Panels (c) and (d) show the condensates density profiles for two different models and increasing couplings $\eta$, namely (c) the WASEP model of particle diffusion with order $m = 3$, and (d) the KMP model of heat conduction with order $m = 2$. Panels (e) and (f) display raster plots of the spatiotemporal evolution of the density field in the WASEP subject to different time-modulated external fields $E_{x,t}[\rho]$ for $\bar{\rho} = 1/3$, see main text. Note the emergence of complex time-crystal phases enhanced with higher-order matter waves. Image adapted from [103].

condensates and the asymmetry of the associated density waves [103]. On the other hand, the packing field $\mathcal{E}^{(m)}_k(C)$ in Eq. (175) drives particles locally towards $m$ *emergent localization centers* where particles are most clustered (e.g. the center of mass for $m = 1$), see Fig. 9.a-b, with an amplitude proportional to $|\mathbf{z}_m|(C)$ and the coupling constant $\eta$. The angular position of these $m$ emergent localization centers is given by the angles

$$\phi_m^{(j)}(C) = \frac{\phi_m(C) + 2\pi j}{m} , \qquad (177)$$

with $j \in [0, m-1]$, i.e. the arguments of the $m$ roots of the complex $m$th-order packing order parameter, $(\sqrt[m]{\mathbf{z}_m})_j = \sqrt[m]{\mathbf{z}_m}e^{i2\pi j/m}$. In the same spirit as before, this generalized packing field mechanism operates by slowing the motion of particles ahead of the nearest localization center, i.e. at lattice sites $k$ where $\mathcal{E}^{(m)}_k(C) < 0$ (see Eq.(175)), while pushing particles that lag behind this point, where $\mathcal{E}^{(m)}_k(C) > 0$, as illustrated in Fig. 9.a-b. The strength of this mechanism is proportional to the coupling constant $\eta$ and the magnitude $|\mathbf{z}_m|$ of the packing order parameter, which quantifies the concentration of particles around the emergent localization centers. The packing field then creates a nonlinear feedback loop that amplifies the system's natural packing fluctuations, ultimately driving a phase transition to a time-crystal phase for sufficiently large coupling $\eta$. This phase exhibits signatures of spontaneous time-translation symmetry breaking [101, 103].

Interestingly, the packing field (175) can be interpreted as a long-range, all-to-all pairwise interaction. Indeed, noting that $\sin\alpha = (e^{+i\alpha} - e^{-i\alpha})/2i$ to expand Eq. (175) as $\mathcal{E}^{(m)}_k(C) =$

$(\mathbf{z}_m e^{-i2\pi mk/L} - \mathbf{z}_m^* e^{+i2\pi mk/L})/2i$, and using the definition (176) for $\mathbf{z}_m$, we can easily show that [101, 103]

$$\mathcal{E}_k^{(m)}(C) = \frac{1}{N} \sum_{k'=1}^{L} n_{k'} \sin\left(\frac{2\pi m(k'-k)}{L}\right). \tag{178}$$

The packing field is thus reminiscent of a generalized Kuramoto-like long-range interaction term, highlighting the mathematical connection between the time crystal lattice gas and the Kuramoto model of oscillator synchronization [172, 173, 175, 176]. This connection is only formal, however, since Kuramoto oscillator model lacks both real-space particle transport and exclusion interactions. Note also that the resulting non-local functional form (178) highlights the need of long-range interactions for the emergence of these time-crystal phases [177].

## 7.4 Hydrodynamic theory and time-crystal instability

At the macroscopic level, the packing field mechanism described above and the resulting phase transition to a time-crystal phase can be understood using hydrodynamic theory, not only for the TCLG model but also for general driven diffusive systems characterized by a diffusivity $D(\rho)$ and a mobility $\sigma(\rho)$. The starting point is a hydrodynamic evolution equation for the density field $\rho(x, t)$ in a $1d$ periodic diffusive system driven by a (non-local) external field $E_x[\rho]$ [178],

$$\partial_t \rho = -\partial_x \left[ -D(\rho)\partial_x \rho + \sigma(\rho)E_x[\rho] \right], \tag{179}$$

with $x \in [0, 1]$. Motivated by the discussion in previous section, the external field now takes the form $E_x[\rho] = \epsilon + \eta \mathcal{E}_x^{(m)}[\rho]$, where $\epsilon$ (constant driving field) and $\eta$ (coupling constant) have the same interpretation as above, while the packing field $\mathcal{E}_x^{(m)}[\rho]$ excites the $m$-th Fourier mode of the density field [101],

$$\mathcal{E}_x^{(m)}[\rho] = \frac{1}{\bar{\rho}} \int_0^1 dy \, \rho(y, t) \sin(2\pi m(y-x)) = |\mathbf{z}_m[\rho]| \sin(\phi_m[\rho] - 2\pi xm), \tag{180}$$

where $\bar{\rho} = \int_0^1 \rho(x, t)dx$ is the conserved average density, and we have used the complex $m$th-order packing order parameter [101, 103],

$$\mathbf{z}_m[\rho] = \frac{1}{\bar{\rho}} \int_0^1 dx \rho(x, t) e^{i2\pi mx} \equiv |\mathbf{z}_m[\rho]| e^{i\phi_m[\rho]}, \tag{181}$$

to rewrite the packing field in the second equality in Eq. (180).

For values of the coupling constant $\eta$ beyond some critical threshold, the hydrodynamic theory (179)-(180) exhibits an instability to a time-crystal phase characterized by the emergence of $m$ rotating condensates, spontaneously breaking spatiotemporal translation symmetry. To show this, a first observation is that the homogeneous density profile $\rho(x, t) = \bar{\rho}$ is a solution of Eq. (179) for any value of $\eta$, so performing a linear stability analysis of this solution will allow us to determine the critical value $\eta_c^{(m)}$ for the instability to happen. We hence introduce a small perturbation around the flat profile, $\rho(x, t) = \bar{\rho} + \delta\rho(x, t)$, ensuring that $\int_0^1 dx \delta\rho(x, t) = 0$ to conserve the system's global density $\bar{\rho}$. Substituting this perturbed profile into Eq. (179) and linearizing to first order in $\delta\rho(x, t)$, we obtain

$$\partial_t \delta\rho = -\partial_x \left[ -\bar{D}\partial_x \delta\rho + \epsilon\bar{\sigma}'\delta\rho + \eta\bar{\sigma}|\mathbf{z}_m[\delta\rho]| \sin(\phi_m[\delta\rho] - 2\pi xm) \right], \tag{182}$$

where $\bar{D} = D(\bar{\rho})$ and $\bar{\sigma} = \sigma(\bar{\rho})$, and we have used that $|\mathbf{z}_m|$ is already first-order in $\delta\rho$, see Eq. (181). The system periodicity suggests to expand the density field perturbation in Fourier

modes,

$$\delta\rho(x,t) = \sum_{j=-\infty}^{\infty} C_j(t)e^{i2\pi x j},$$

where the $j$-th Fourier coefficient in this expansion is given by $C_j(t) = \int_0^1 dx\,\delta\rho(x,t)e^{-i2\pi x j}$. Noticing now that the packing order parameter of the density field perturbation can be written in terms of the $(-m)$-th Fourier coefficient, i.e. $\mathbf{z}_m[\delta\rho] = C_{-m}(t)/\bar{\rho}$, and using the above Fourier expansion in Eq. (182), we arrive at

$$\sum_{j=-\infty}^{\infty} \left(\partial_t C_j(t) + \zeta_j C_j(t)\right)e^{i2\pi x j} = 0, \tag{183}$$

where we have defined the coefficients

$$\zeta_j \equiv (2\pi j)^2 \bar{D} + i2\pi j\bar{\sigma}'\epsilon - \eta\frac{\bar{\sigma}}{2\bar{\rho}}2\pi m(\delta_{j,m} + \delta_{j,-m}), \tag{184}$$

and $\delta_{j,m}$ and $\delta_{j,-m}$ are Kronecker deltas. As the different complex exponentials in Eq. (183) are linearly independent, each term in the sum must be zero independently, leading to a set of simple uncoupled differential equations for the different Fourier coefficients, $\partial_t C_j(t) + \zeta_j C_j(t) = 0\ \forall j$. The solutions are just exponentials, $C_j(t) = C_j(0)e^{-\zeta_j t}$, with $C_j(0)$ the $j$th-coefficients of the initial perturbation $\delta\rho(x,0)$. The stability of the various Fourier modes is then governed by the real part of $\zeta_j$, which requires us to examine two separate cases: $|j| \neq m$ and $|j| = m$. For the first case ($|j| \neq m$), we have $\mathrm{Re}(\zeta_j) = (2\pi j)^2 \bar{D} > 0\ \forall j$, indicating that these Fourier modes will always decay. Conversely, for $|j| = m$, the decay rate reflects a competition between the diffusion term and the packing field, i.e.,

$$\mathrm{Re}(\zeta_{\pm m}) = (2\pi m)^2\left(\bar{D} - \eta\frac{\bar{\sigma}}{4\pi m\bar{\rho}}\right). \tag{185}$$

The critical value of $\eta$ is attained when $\mathrm{Re}(\zeta_{\pm m}) = 0$, and is given by

$$\eta_{\mathrm{c}}^{(m)} = 4\pi m\frac{\bar{D}\bar{\rho}}{\bar{\sigma}}. \tag{186}$$

The homogeneous density solution $\rho(x,t) = \bar{\rho}$ will hence become unstable when $\mathrm{Re}(\zeta_{\pm m}) < 0$ or equivalently $\eta > \eta_{\mathrm{c}}^{(m)}$, resulting in a non-homogeneous density field solution with a more intricate spatiotemporal structure.

Based on our previous analysis of a similar transition (see section §5), we expect that, right beyond the instability, the main non-homogeneous contributions to the density profile will arise from the $\pm m$th-order (unstable) Fourier modes. Consequently, we anticipate in this regime ($\eta \gtrsim \eta_{\mathrm{c}}^{(m)}$) a traveling-wave profile approximately given by $\rho(x,t) = \bar{\rho} + A\cos(\omega t - 2\pi m x)$, where $A$ is a small amplitude, and the angular velocity $\omega$ is determined by the imaginary part of Eq. (184), $\omega = 2\pi m\bar{\sigma}'\epsilon$, at least close to the instability. In this way, the homogeneous density turns beyond the instability into a single ($m = 1$) or multiple ($m > 1$) condensates periodically moving at a constant velocity proportional to $\bar{\sigma}'$ and $\epsilon$. This instability hence breaks spontaneously the time-traslation symmetry of the homogeneous solution, thus giving rise to a continuous time crystal [97–102, 154–156]. This time-crystal phase is fully *programmable*, in the sense that we can control the number of emerging condensates with the order $m$ of the packing field applied, as well as their shape and velocity using $\epsilon$ [102, 103]. On the other hand, the average current $\langle q \rangle = \int_0^1 dx\,j(x,t)$ can be calculated from the local current in the linearized equation (182), resulting in $\langle q \rangle = \langle q \rangle_0 + A^2\bar{\sigma}''\epsilon/4$ right beyond the instability,

where $\langle q \rangle_0 = \bar{\sigma}\epsilon$ is the mean current in the homogeneous phase. While these equations hold true only close to the instability, $\eta \gtrsim \eta_c^{(m)}$, they highlight the relevance of the model transport coefficients for its response under a packing field. In particular, the slope and convexity of the mobility coefficient for each model will determine whether the packing field will enhance or lower the current and speed of the traveling wave.

Interestingly, the value of the critical coupling $\eta_c^{(m)}$ in Eq. (186) grows with $m$, reflecting the subtle interplay between diffusion and the packing field. Specifically, the effect of diffusion, which acts against the $m$ emergent condensates, scales as $m^2$ near the instability, see first term in the rhs of Eq. (185), whereas the influence of the packing field, which favors the formation of condensates, scales as $m$. Consequently, a higher coupling $\eta$ is required to destabilize the flat solution as the integer index $m$ increases.

A remarkable feature of these programmable time-crystal phases is that the $m$th-order ($m > 1$) traveling-wave solution of Eq. (179) in the symmetry-broken regime can be constructed by *gluing* together $m$ copies of the $m = 1$ solution with appropriately rescaled driving parameters. Specifically, it can be shown [102] that $\rho_m(\omega_m t - 2\pi m x) = \rho_1(m\omega_1 t - 2\pi m x)$, where $\rho_1(\omega_1 t - 2\pi x)$ is a traveling-wave solution of Eq. (179) with velocity $\omega_1$ for $m = 1$ under generic driving parameters $\epsilon_1$ and $\eta_1$. Meanwhile, $\rho_m(\omega_m t - 2\pi m x)$ represents the corresponding solution for $m > 1$, with velocity $\omega_m = m\omega_1$ and rescaled driving parameters $\epsilon_m = m\epsilon_1$ and $\eta_m = m\eta_1$. This scaling law is valid for generic nonlinear transport coefficients, and allows to collapse traveling-wave profiles across different orders $m$ and driving parameters, simplifying the range of possible solutions.

To illustrate these ideas, we particularize our results for two of the paradigmatic transport models already studied in these lecture notes and which admit a hydrodynamic description of the form of Eq. (179), but now under the action of a packing field (180). These systems are the WASEP model mentioned above, characterized by $D(\rho) = 1/2$ and $\sigma(\rho) = \rho(1-\rho)$ [120,168], and the KMP model of heat transport [119], with $D(\rho) = 1/2$ and $\sigma(\rho) = \rho^2$. Both models exhibit programmable time-crystal phases appearing for couplings above the critical threshold $\eta_c^{(m)}$, and to study the resulting traveling wave patterns we solved numerically Eq. (179) using the prescribed $D(\rho)$ and $\sigma(\rho)$ in each case [103]. Figs. 9.c,d display the condensate density profiles obtained for both models with $\bar{\rho} = 1/3$, for different orders $m = 2$, 3 and multiple supercritical couplings $\eta > \eta_c^{(m)}$. Notably, the condensate strusture in each case reveals the nonlinear transport features specific to each model. For WASEP, the emergence of condensates hinders the overall particle dynamics due to the particle exclusion interactions, leading to a current suppression compared to the homogeneous phase (note that $\bar{\sigma}'' < 0$ in this case). This results in a sharp density accumulation at the condensate's tail (Fig. 9.c), while the front exhibits a gradual decay due to the available free space. In contrast, the KMP model shows the opposite behavior: the excess current is positive ($\bar{\sigma}'' > 0$ now), indicating that dynamics in the time-crystal phase is faster than in the homogeneous phase. Consequently, condensates feature now a sharp front and a smoother tail, as shown in Fig 9.d.

These programmable time-crystal phases can be enhanced by incorporating higher-order matter waves resulting from competing packing fields modulated in time. To illustrate this idea, consider a generalized external field given by $E_{x,t}[\rho] = \epsilon + \sum_m \eta_m(t)\mathcal{E}x^{(m)}[\rho]$. Figs. 9.e,f display the spatiotemporal evolution of $\rho(x,t)$ obtained from the numerical solution of Eq. (179) under various modulated combinations of external fields. For example, we can alternate between different numbers of condensates over time (Fig. 9.e), by activating and shutting off specific orders $m$ via time-dependent modulation of $\eta_m(t)$, as shown in Fig. 9.e. Additionally, customized decorated time-crystal phases can be achieved by activating a higher-order $2m$ mode in time through $\eta^{(2m)}(t)$, as illustrated in Fig. 9.f. This occurs against a constant background matter wave created by setting $\eta^{(m)} > \eta_c^{(m)}$. Remarkably, a time-dependent decorated pattern emerges, oscillating in phase with $\eta_{2m}(t)$. This pattern switches between a symmetric

time-crystal phase with $m$ condensates when $\eta_{2m}(t) \approx 0$ and an asymmetric phase with $2m$ condensates when $\eta_{2m}(t) > \eta_c^{(2m)}$. These examples, among countless other intriguing possibilities, highlight the versatility of the packing-field approach for engineering and controlling programmable time-crystal phases in driven diffusive fluids, paving the way for exciting future research and technological advancements.

Before ending this section, we want to stress that these programmable time-crystal phases could be engineered in the lab using current technologies. A promising experimental route consists in confining assemblies of colloidal particles ring-shaped light traps, created for instance with infrared optical tweezers steered through an acousto-optic deflector [179–181]; see also [182, 183]. The necessary packing field could be created using a feedback loop from particle tracking in real time. This would allow to modulate the depth of the individual lattice traps to bias motion locally in a configuration-dependent way. This opens up a path to exploit the rare-event route to time crystals in this and other geometries.

## 8   Discussion and conclusion

In these lecture notes we have explored the statistics of current fluctuations (and their associated *thermodynamics*) in diffusive systems driven out of equilibrium by external fields and/or boundary gradients, paying special attention to the optimal path to a current fluctuation, and the symmetry properties of this trajectory. We have employed macroscopic fluctuation theory (MFT) as a primary framework [2], complemented by microscopic spectral methods and symmetry tools.

MFT provides a systematic way to analyze the probability distributions of trajectory dependent observables, as e.g. the empirical current, offering insights into the emergent behavior of many-body systems under sustained driving forces [2]. The starting point is a mesoscopic fluctuating hydrodynamic equation that governs the evolution of a density field, whose validity has been rigorously established for a broad class of stochastic microscopic models [178]. From this equation, and using a path integral formalism, we can derive the probability of rare events and identify the associated optimal paths in mesoscopic phase space. These paths reveal how the system navigates through unlikely states, shedding light on the interplay between microscopic dynamics and macroscopic fluctuations. The path action can be then used to obtain the large deviation functions of the relevant macroscopic observables through contraction principles [1, 2], and in particular the current LDF.

The resulting spatiotemporal variational problem is typically challenging, but we have shown how a neat and powerful additivity conjecture [27] simplifies the calculation in $1d$ driven diffusive systems, enabling explicit predictions for the current LDF and the associated optimal path. The additivity principle in $1d$ amounts to assume the time-independence of the optimal path, and its predictions have been confirmed with high precision in different models using numerical simulations of rare events [38, 40, 44, 114, 118]. We have extended the additivity conjecture to general $d$-dimensional driven diffusive media, demonstrating that the existence of a structured optimal current vector field, coupled to the local mobility, is essential to understand current statistics in $d > 1$ [50, 53]. This stems from a more general, fundamental relation which strongly constrains the architecture of optimal paths in $d$ dimensions. The predictions obtained from the so-called weak additivity principle have been confirmed against both rare-event simulations and microscopic exact calculations of different paradigmatic models of diffusive transport in $d > 1$ [41, 49, 50, 125]. These models are however somewhat oversimplified for real applications, and the challenge now is to extend the path integral formulation of MFT to more complex nonequilibrium scenarios, characterized e.g. by multiple local conservation laws with nonlinear transport in $d > 1$ [113]. This will allow to

investigate coupled fluctuations of multiple currents (e.g. energy and particle flows), looking for organizing principles that generalize the weak additivity conjecture to these more intricate situations. Furthermore, it seems also interesting to explore symmetries associated with the invariance of optimal paths in the MFT action functional to formulate new fluctuation theorems [61] for the coupled current statistics.

In this lecture notes we have also learned how rare fluctuations in a many-body system can be realized sometimes via symmetry-broken trajectories that maximize their probability [14, 28, 29, 43, 44, 52, 71, 108, 109, 129–132]. These changes appear in the form of dynamical phase transitions, accompanied by non-analyticities and Lee-Yang singularities in the associated large deviation functions. In particular, we have examined transport fluctuations in open channels coupled to boundary reservoirs, where we found a discrete particle-hole symmetry-breaking DPT for currents, for which we studied a Landau-like theory as well as the joint statistics of the current and a suitable order parameter. Interestingly, we also learned here that the additivity conjecture can be eventually violated in the form of time-dependent (instantonic) optimal paths associated to the non-convex regimes of the joint large deviation function [109]. The possibility of time-translation symmetry breaking DPTs in periodic systems has been also discussed in these lecture notes. In this case the system of interest develops a coherent traveling wave to facilitate current fluctuations beyond some critical threshold. In the same spirit than above, it would be interesting to study similar DPTs for coupled current fluctuations in more realistic systems characterized by several conservation laws, see e.g. [113] for a two-fields transport problem with particle density and energy. We anticipate that the competition between particle and energy currents can result in a unexplored types of DPTs with nontrivial cooperative effects. In addition, it would be desirable to investigate the role of dimensionality, see e.g. [53], to make contact with the physics of DPTs in hydrodynamic settings.

We have also studied the microscopic spectral mechanism underlying some of the DPTs studied in previous sections at the MFT level. Interestingly, the appearance of a DPT is associated with an emergent degeneracy of the ground state of the microscopic Markov generator, with all symmetry-breaking properties determined by the subleading eigenvectors of this degenerate subspace. By introducing a lower-dimensional order-parameter space, we were able to analyze the spectral signatures of these DPTs and understand the underlying mechanism in terms of a Doob's smart field that makes these rare DPTs typical. It would be interesting to apply this method for the DPT observed in open channels so as to uncover the external field driving this transition. This method has also proven useful to understand time-translation symmetry-breaking DPTs to traveling waves in ring geometry as a result of an instability triggered by a non-local packing field. This packing field drives forward particles lagging behind the instantaneous center of mass while slowing down those moving ahead, opening the door to engineering programmable time crystal phases in many-body systems. The challenge now is to tailor complex time-crystalline phases in systems with multiple conservation laws, both in $1d$ and $d > 1$. Moreover, the modern experimental control of trapped colloidal fluids [179–183] and the availability of feedback-control force protocols to implement the nonlinear packing field using optical tweezers opens the door to the lab characterization of these time-crystal phases.

As a final side note, many of the ideas here described can be also explored in the quantum realm, starting from the Lindblad master equation [90, 184, 185] describing the coherent evolution of an open quantum system's density matrix, punctuated with incoherent quantum jumps induced by the environment. This has paved the way to the understanding of the thermodynamics of quantum-jump trajectories, encompassing the statistics of current and activity fluctuations in open quantum systems, and uncovering along the way various dynamical phase transitions similar in spirit to those found in classical systems. Using these tools, the power of

symmetry as a resource to control quantum transport has been recently demonstrated, leading to the design of quantum devices with novel transport properties, as e.g. symmetry-controlled quantum thermal switches [89–91] and quantum engines [186, 187]. These results highlight the importance of symmetries not only as a fundamental principle in physics but also as a resource for controlling quantum systems.

We hope that the results described in these lecture notes will inspire young physicists to work in this field, as they show that the combined use of MFT and its extensions together with microscopic spectral methods (and rare-event simulation techniques for many-body systems) offer a robust theoretical framework to advance the frontiers of knowledge in nonequilibrium physics.

## Acknowledgements

I would like to thank my colleagues Abhishek Dhar, Joachin Krug, Satya N. Majumdar, Alberto Rosso and Gregory Schehr, for organizing a wonderful 2024 Les Houches Summer School on the Theory of Large Deviations and Applications, where I delivered the set of lectures summarized in these notes. I also thank my fellow lecturers, for creating a nice and exciting scientific environment that we all enjoyed, and students for promoting a special atmosphere in a unique environment such as the *Ecole de Physique des Les Houches*, in the french alps, with magnificent views on the Mont Blanc range. Having the opportunity to lecture at Les Houches, where so many distinguished scientists have lectured before me, it's a honor and a great pleasure.

I would like to thank also my main collaborators in the works described in these lecture notes, in particular profs. Pedro L. Garrido, Rubén Hurtado-Gutierrez, Carlos Pérez-Espigares, and Nicolás Tizón-Escamilla, as well as Federico Carollo and Juan P. Garrahan, for the many discussions and efforts that led to these results. I also want to acknowledge here all the colleagues with whom I've discussed about these topics during all these years. They are too many to list them here, but my gratitude goes to all of them.

The research leading to these results has received funding from the I+D+i grants PID2023-149365NB-I00, PID2020-113681GB-I00, PID2021-128970OA-I00, C-EXP-251-UGR23 and P20 _00173, funded by MICIU/AEI/10.13039/501100011033/, ERDF/EU, and Junta de Andalucía - Consejería de Economía y Conocimiento, as well as from fellowship FPU17/02191 financed by the Spanish Ministerio de Universidades. We are also grateful for the the computing resources and related technical support provided by PROTEUS, the supercomputing center of Institute Carlos I for Theoretical and Computational Physics in Granada, Spain.

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
