# Peer review of "Optimal paths and dynamical symmetry breaking in the current fluctuations of driven diffusive media"

_SciPost Physics Lecture Notes_

## Round 1 · Referee Report · Baruch Meerson (Referee 1) · 2025-9-15

Report
This is a timely and useful review paper on an important subject of large-deviation theory of macroscopic stochastic systems out of equilibrium.
I recommend that the author introduce the following minor changes:
-
Introduction: Rare events are not always dominated by optimal paths. A well-studied class of examples is provided by large deviations of long-time averages of fluctuating quanties, treated by the Donsker-Varadhan theory, reviewed e.g. in Ref. [8]. For such large deviations there are multiple paths which contribute in an important way to the large-deviation function in question.
-
Sec. 2, shortly after Eq. (4). The local thermodynamic equilibrium (LTE) assumption is too important for the MFT to mention it only in passing. The reader would benefit from an explanation of why the LTE assumption is valid in this class of problems.
-
Eqs. (6) and (7). Why not using a more standard sign convention here and in the following, which would have a minus sign in (6), so that the action functional in (7) is positive, as it is common in classical mechanics and classical field theory?
-
Shortly after Eq. (12). The terms "time-independent" and "stationary" are traditionally used as synonyms, so this sentence can cause confusion. I think the term "the stationary state observed in the absence of fluctuations" would make this statement more clear.
-
End of page 11. "Fokker-Planck description" should be replaced by "a path-integral description".
-
Page 32. When citing Refs. [2,14,29,43,44,56,108], the author should also cite Ref. [66].
-
The paper would benefit from a slight language polishing.
Recommendation
Ask for minor revision

---

## Editorial Decision

unknown